# Uncertainty Quantification for Physics-Informed Neural Networks with Extended Fiducial Inference

**Frank Shih**
Department of Epidemiology and Biostatistics
Memorial Sloan Kettering Cancer Center
New York, NY 10017
shihf@mskcc.org

**Zhenghao Jiang**
Department of Statistics
Purdue University
West Lafayette, IN 47907
jiang976@purdue.edu

**Faming Liang**
Department of Statistics
Purdue University
West Lafayette, IN 47907
fmliang@purdue.edu

## Abstract

Uncertainty quantification (UQ) in scientific machine learning is increasingly critical as neural networks are widely adopted to tackle complex problems across diverse scientific disciplines. For physics-informed neural networks (PINNs), a prominent model in scientific machine learning, uncertainty is typically quantified using Bayesian or dropout methods. However, both approaches suffer from a fundamental limitation: the prior distribution or dropout rate required to construct honest confidence sets cannot be determined without additional information. In this paper, we propose a novel method within the framework of extended fiducial inference (EFI) to provide rigorous uncertainty quantification for PINNs. The proposed method leverages a narrow-neck hyper-network to learn the parameters of the PINN and quantify their uncertainty based on imputed random errors in the observations. This approach overcomes the limitations of Bayesian and dropout methods, enabling the construction of honest confidence sets based solely on observed data. This advancement represents a significant breakthrough for PINNs, greatly enhancing their reliability, interpretability, and applicability to real-world scientific and engineering challenges. Moreover, it establishes a new theoretical framework for EFI, extending its application to large-scale models, eliminating the need for sparse hyper-networks, and significantly improving the automaticity and robustness of statistical inference.

**Keywords**: Adaptive Stochastic Gradient MCMC, Deep Learning, Black-Scholes Model, Partial Differential Equation, Porous-FKPP Model

## 1 Introduction

Physics-informed neural networks (PINNs) (Raissi et al., 2019) are a class of scientific machine learning models that integrate physical principles directly into the training process of the DNN models. They achieve this by incorporating terms derived from ordinary differential equations (ODEs) or partial differential equations (PDEs) into the DNN's loss function. PINNs take spatial-temporal coordinates as input and produce functions that approximate the solutions to the differential equations. Because they embed physics-based constraints, PINNs can typically address problems that are described by few data while ensuring adherence to the given physical laws (Zou et al., 2025; Cuomo

et al., 2022), paving the ways for the use of neural networks in out-of-distribution (OOD) prediction (see e.g., Yao et al. (2024)).

From a Bayesian viewpoint, PINNs can be seen as using "informative priors" drawn from mathematical models, thus requiring less data for system identification (Zhang et al., 2019). However, this naturally raises a challenge in uncertainty quantification: How should one balance the prior information and the observed data to ensure faithful inference about the underlying physical system? As emphasized in Zou et al. (2025), uncertainty quantification is becoming increasingly critical as neural networks are widely employed to tackle complex scientific problems, particularly in high-stake application scenarios. However, addressing this issue within the Bayesian framework is difficult, as the prior information and data information are essentially exchangeable under Bayesian formulations. The use of "informative priors" can conflict with the spirit of "posterior consistency" (see e.g., Ghosal et al. (2000)), a foundational principle in Bayesian inference. This conflict creates a dilemma: *using an informative prior risks overshadowing the data, while using a weak prior may lead to violations of the underlying physical law.* In practice, this makes it difficult, if not impossible, to properly calibrate the resulting credible intervals without additional information.

In addition to Bayesian methods (Yang et al., 2021), dropout (Srivastava et al., 2014) has also been employed to quantify the uncertainty of PINNs, as demonstrated by Zhang et al. (2019). Dropout is primarily used as a regularization technique to reduce overfitting during DNN training. Gal and Ghahramani (Gal and Ghahramani, 2016) showed that dropout training in DNNs can be interpreted as approximate Bayesian inference in deep Gaussian processes, allowing model uncertainty to be estimated from dropout-trained DNN models. However, this approach shares a limitation similar to Bayesian methods: *The dropout rate, which directly influences the magnitude of the estimated model uncertainty, cannot be determined without additional information, making it challenging to ensure consistent and reliable uncertainty quantification.*

This paper introduces an EFI (Liang et al., 2025) approach to quantify the uncertainty in PINNs. EFI provides a rigorous theoretical framework that addresses the limitations of Bayesian and dropout methods by formulating the problem as a structural equation-solving task. In this framework, each observation is expressed as a data-generating equation, with the random errors (contained in observations) and DNN parameters treated as unknowns (see Section 2). EFI jointly imputes the random errors and estimates the inverse function that maps the observations and imputed random errors to DNN parameters. Consequently, the imputed random errors are propagated to the DNN parameters through the estimated inverse function, allowing the model uncertainty to be accurately quantified without the need for additional information. Our contribution in this paper is two-fold:

- **A new theoretical framework for EFI:** We develop a new theoretical framework for EFI that significantly enhances the automaticity of statistical inference. Originally, EFI was developed in Liang et al. (2025) under a Bayesian framework, where a sparse prior is imposeStochasticd on the hyper-network (referred to as the $w$-network in Section 2) to ensure consistent estimation of the inverse function. However, due to the limitations of existing sparse deep learning theory (Sun et al., 2022), this Bayesian approach could only be applied to models with dimensions fixed or increasing at a very low rate with the sample size. In this paper, we propose learning the inverse function using a narrow-neck $w$-network, which ensures consistent estimation of the inverse function without relying on the use of sparse priors. Moreover, it enables EFI to work for large-scale models, such as PINNs, where the number of model parameters can far exceed the sample size. By avoiding the need for Bayesian sparse priors, our framework allows EFI to fulfil the original goal of fiducial inference: *Inferring the uncertainty of model parameters based solely on observations.*

- **Open-source software for uncertainty quantification in PINNs:** We provide an open source software package for uncertainty quantification in PINNs, which can be easily extended to conventional DNNs and other high-dimensional statistical models.

**Related Work**   The proposed method belongs to the class of imprecise probabilistic techniques (Augustin et al., 2014). However, compared to other methods in the class, such as credal Bayesian deep learning (Caprio et al., 2023a), imprecise Bayesian neural networks (Caprio et al., 2023b), and other Bayesian neural network-based methods, the key advantage of EFI is that it avoids the need for prior specification while ensuring accurate calibration of predictions.

Another related line of work concerns uncertainty quantification for machine learning models. Beyond the Bayesian and dropout methods noted above, this line includes conformal prediction

(Vovk et al., 2005), deep ensembles (Lakshminarayanan et al., 2016), and stochastic deep learning (Sun and Liang, 2022; Liang et al., 2022), among others. These methods primarily target predictive uncertainty and are often ineffective or inapplicable for quantifying uncertainty in model parameters. In contrast, EFI addresses both predictive uncertainty and parameter uncertainty, and further provides theoretical guarantees for the validity of the resulting prediction and confidence intervals. The ability to accurately quantify uncertainty in deep neural network parameters is a distinctive advantage of EFI.

## 2 A Brief Review of EFI

While fiducial inference was widely considered as a big blunder by R.A. Fisher, the goal he initially set —inferring the uncertainty of model parameters based solely on observations — has been continually pursued by many statisticians, see e.g. structural inference (Fraser, 1966, 1968), generalized fiducial inference (Hannig, 2009; Hannig et al., 2016; Murph et al., 2022), and inferential models (Martin and Liu, 2013, 2015; Martin, 2023). To this end, Liang et al. (2025) developed the EFI method based on the fundamental concept of structural inference.

Consider a regression model: $Y = f(\boldsymbol{X}, Z, \boldsymbol{\theta})$, where $Y \in \mathbb{R}$ and $\boldsymbol{X} \in \mathbb{R}^d$ represent the response and explanatory variables, respectively; $\boldsymbol{\theta} \in \mathbb{R}^p$ represents the vector of parameters; and $Z \in \mathbb{R}$ represents a scaled random error following a known distribution $\pi_0(\cdot)$. Suppose that a random sample of size $n$, denoted by $\{(y_1, \boldsymbol{x}_1), (y_2, \boldsymbol{x}_2), \ldots, (y_n, \boldsymbol{x}_n)\}$, has been collected from the model. In structural inference, the observations can be expressed in data-generating equations as follows:

$$y_i = f(\boldsymbol{x}_i, z_i, \boldsymbol{\theta}), \quad i = 1, 2, \ldots, n. \tag{1}$$

This system of equations consists of $n + p$ unknowns, namely, $\{\boldsymbol{\theta}, z_1, z_2, \ldots, z_n\}$, while there are only $n$ equations. Therefore, the values of $\boldsymbol{\theta}$ cannot be uniquely determined by the data-generating equations, and this lack of uniqueness of unknowns introduces uncertainty in $\boldsymbol{\theta}$.

Let $\boldsymbol{Z}_n = \{z_1, z_2, \ldots, z_n\}$ denote the unobservable random errors contained in the data, which are also called latent variables in EFI. Let $G(\cdot)$ denote an inverse function/mapping for $\boldsymbol{\theta}$, i.e.,

$$\boldsymbol{\theta} = G(\boldsymbol{Y}_n, \boldsymbol{X}_n, \boldsymbol{Z}_n). \tag{2}$$

It is worth noting that the inverse function is generally non-unique. For example, it can be constructed by solving any $p$ equations in (1) for $\boldsymbol{\theta}$. As noted by Liang et al. (2025), this non-uniqueness of inverse function mirrors the flexibility of frequentist methods, where different estimators of $\boldsymbol{\theta}$ can be constructed to achieve desired properties such as efficiency, unbiasedness, and robustness.

Since the inverse function $G(\cdot)$ is generally unknown, Liang et al. (2025) proposed to approximate it using a sparse DNN, see Figure A1 in the Appendix for illustration. They also introduced an adaptive stochastic gradient Langevin dynamics (SGLD) algorithm, which facilitates the simultaneous training of the sparse DNN and simulation of the latent variables $\boldsymbol{Z}_n$. See Algorithm 1 for the pseudo-code. Refer to Section A1 of the Appendix for the mathematical formulation of the method. Briefly, they let $\boldsymbol{w}_n$ denote the weights of $\boldsymbol{w}$-network and define an energy function $U_n(\boldsymbol{Y}_n, \boldsymbol{X}_n, \boldsymbol{Z}_n, \boldsymbol{w}_n)$. subsequently, they define a posterior distribution $\pi_\epsilon(\boldsymbol{w}_n | \boldsymbol{X}_n, \boldsymbol{Y}_n, \boldsymbol{Z}_n)$ for $\boldsymbol{w}_n$ and a predictive distribution $\pi_\epsilon(\boldsymbol{Z}_n | \boldsymbol{X}_n, \boldsymbol{Y}_n, \boldsymbol{w}_n)$ for $\boldsymbol{Z}_n$, where $\epsilon$ can be read as a temperature. They treat $\boldsymbol{Z}_n$ as missing data and learn $\boldsymbol{w}_n$ through solving the following equation:

$$\nabla_{\boldsymbol{w}_n} \log \pi_\epsilon(\boldsymbol{w}_n | \boldsymbol{X}_n, \boldsymbol{Y}_n) = \int \left[ \nabla_{\boldsymbol{w}_n} \log \pi_\epsilon(\boldsymbol{w}_n | \boldsymbol{X}_n, \boldsymbol{Y}_n, \boldsymbol{Z}_n) \pi_\epsilon(\boldsymbol{Z}_n | \boldsymbol{X}_n, \boldsymbol{Y}_n, \boldsymbol{w}_n) \right] d\boldsymbol{Z}_n = 0, \tag{3}$$

using Algorithm 1.

Under mild conditions for the adaptive SGLD algorithm, it can be shown that

$$\|\boldsymbol{w}_n^{(k)} - \boldsymbol{w}_n^*\| \xrightarrow{p} 0, \quad \text{as } k \to \infty, \tag{5}$$

where $\boldsymbol{w}_n^*$ denotes a solution to equation (3) and $\xrightarrow{p}$ denotes convergence in probability, and that

$$\boldsymbol{Z}_n^{(k)} \xrightarrow{d} \pi_\epsilon(\boldsymbol{Z}_n | \boldsymbol{X}_n, \boldsymbol{Y}_n, \boldsymbol{w}_n^*), \quad \text{as } k \to \infty, \tag{6}$$

in 2-Wasserstein distance, where $\xrightarrow{d}$ denotes weak convergence. To study the limit of (6) as $\epsilon$ decays to 0, i.e., $p_n^*(\boldsymbol{z} | \boldsymbol{Y}_n, \boldsymbol{X}_n, \boldsymbol{w}_n^*) = \lim_{\epsilon \downarrow 0} \pi_\epsilon(\boldsymbol{Z}_n | \boldsymbol{X}_n, \boldsymbol{Y}_n, \boldsymbol{w}_n^*)$, where $p_n^*(\boldsymbol{z} | \boldsymbol{Y}_n, \boldsymbol{X}_n, \boldsymbol{w}_n^*)$ is referred

---
**Algorithm 1:** Adaptive SGLD for EFI computation
---

**(i) (Initialization)** Initialize $\boldsymbol{w}_n^{(0)}$, $\boldsymbol{Z}_n^{(0)}$, $M$ (the number of fiducial samples to collect), and $\mathcal{K}$ (burn-in iterations).

**for** *k=1,2,...,$\mathcal{K}+M$* **do**

> **(ii) (Latent variable imputation)** Given $\boldsymbol{w}_n^{(k)}$, simulate $\boldsymbol{Z}_n^{(k+1)}$ using the SGLD algorithm:
>
> $$\boldsymbol{Z}_n^{(k+1)} = \boldsymbol{Z}_n^{(k)} + \upsilon_{k+1}\nabla_{\boldsymbol{Z}_n} \log \pi_\epsilon(\boldsymbol{Z}_n^{(k)}|\boldsymbol{X}_n, \boldsymbol{Y}_n, \boldsymbol{w}_n^{(k)}) + \sqrt{2\upsilon_{k+1}}\boldsymbol{e}^{(k+1)}$$
>
> where $\upsilon_{k+1}$ is the learning rate, and $\boldsymbol{e}^{(k+1)} \sim N(0, I_{d_{\boldsymbol{z}}})$.
>
> **(iii) (Parameter updating)** Draw a minibatch $\{(y_1, \boldsymbol{x}_1, z_1^{(k)}), \ldots, (y_m, \boldsymbol{x}_m, z_m^{(k)})\}$ and update the network weights by the SGD algorithm:
>
> $$\boldsymbol{w}_n^{(k+1)} = \boldsymbol{w}_n^{(k)} + \gamma_{k+1}\left[\frac{n}{m}\sum_{i=1}^{m} \nabla_{\boldsymbol{w}_n} \log \pi_\epsilon(y_i|\boldsymbol{x}_i, z_i^{(k)}, \boldsymbol{w}_n^{(k)}) + \nabla_{\boldsymbol{w}_n} \log \pi(\boldsymbol{w}_n^{(k)})\right], \quad (4)$$
>
> where $\gamma_{k+1}$ is the step size, and $\log \pi_\epsilon(y_i|\boldsymbol{x}_i, z_i^{(k)}, \boldsymbol{w}_n^{(k)})$ can be appropriately defined according to (A6).
>
> **(iv) (Fiducial sample collection)** If $k+1 > \mathcal{K}$, calculate $\hat{\boldsymbol{\theta}}_i^{(k+1)} = \hat{g}(y_i, \boldsymbol{x}_i, z_i^{(k+1)}, \boldsymbol{w}_n^{(k+1)})$ for each $i \in \{1, 2, \ldots, n\}$ and average them to get a fiducial $\bar{\boldsymbol{\theta}}$-sample as calculated in (A5).

**end**

**(v) (Statistical Inference)** Conducting statistical inference for the model based on the collected fiducial samples.

---

to as the extended fiducial density (EFD) of $\boldsymbol{Z}_n$, Liang et al. (2025) impose specific conditions on the structure of the $\boldsymbol{w}$-network, including that the $\boldsymbol{w}$-network is sparse and that the output layer width (i.e., the dimension of $\boldsymbol{\theta}$) is either fixed or grows very slowly with the sample size $n$. Under these assumptions, they prove the consistency of $\boldsymbol{w}_n^*$ based on the sparse deep learning theory developed in Sun et al. (2022). This consistency further implies that

$$G^*(\boldsymbol{Y}_n, \boldsymbol{X}_n, \boldsymbol{Z}_n) = \frac{1}{n}\sum_{i=1}^{n} \hat{g}(y_i, \boldsymbol{x}_i, z_i, \boldsymbol{w}_n^*), \quad (7)$$

serves as a consistent estimator for the inverse function/mapping $\boldsymbol{\theta} = G(\boldsymbol{Y}_n, \boldsymbol{X}_n, \boldsymbol{Z}_n)$, where $\hat{g}(\cdot)$ denotes the learned neural network function. Refer to Appendix A2 for the expression of $p_n^*(\boldsymbol{z}|\boldsymbol{Y}_n, \boldsymbol{X}_n, \boldsymbol{w}_n^*)$.

Let $\mathcal{Z}_n = \{\boldsymbol{z} \in \mathbb{R}^n : U_n(\boldsymbol{Y}_n, \boldsymbol{X}_n, \boldsymbol{Z}_n, \boldsymbol{w}_n^*) = 0\}$ denote the zero-energy set. Under some regularity conditions on the energy function, Liang et al. (2025) proved that $\mathcal{Z}_n$ is invariant to the choice of $G(\cdot)$. Let $\Theta := \{\boldsymbol{\theta} \in \mathbb{R}^p : \boldsymbol{\theta} = G^*(\boldsymbol{Y}_n, \boldsymbol{X}_n, \boldsymbol{z}), \boldsymbol{z} \in \mathcal{Z}_n\}$ denote the parameter space of the target model, which represents the set of all possible values of $\boldsymbol{\theta}$ that $G^*(\cdot)$ takes when $\boldsymbol{z}$ runs over $\mathcal{Z}_n$. Then, for any function $b(\boldsymbol{\theta})$ of interest, its EFD $\mu_n^*(\cdot|\boldsymbol{Y}_n, \boldsymbol{X}_n)$ associated with $G^*(\cdot)$ is given by

$$\mu_n^*(B|\boldsymbol{Y}_n, \boldsymbol{X}_n) = \int_{\mathcal{Z}_n(B)} dP_n^*(\boldsymbol{z}|\boldsymbol{Y}_n, \boldsymbol{X}_n, \boldsymbol{w}_n^*), \quad (8)$$

for any measurable set $B \subset \Theta$, where $\mathcal{Z}_n(B) = \{\boldsymbol{z} \in \mathcal{Z}_n : b(G^*(\boldsymbol{Y}_n, \boldsymbol{X}_n, \boldsymbol{z})) \in B\}$, and $P_n^*(\boldsymbol{z}|\boldsymbol{X}_n, \boldsymbol{Y}_n, \boldsymbol{w}_n^*)$ denote the cumulative distribution function (CDF) corresponding to $p_n^*(\boldsymbol{z}|\boldsymbol{X}_n, \boldsymbol{Y}_n, \boldsymbol{w}_n^*)$. The EFD provides an uncertainty measure for $b(\boldsymbol{\theta})$. Practically, it can be constructed based on the samples $\{b(\bar{\boldsymbol{\theta}}_1), b(\bar{\boldsymbol{\theta}}_2), \ldots, b(\bar{\boldsymbol{\theta}}_M)\}$, where $\{\bar{\boldsymbol{\theta}}_1, \bar{\boldsymbol{\theta}}_2, \ldots, \bar{\boldsymbol{\theta}}_M\}$ denotes the fiducial $\bar{\boldsymbol{\theta}}$-samples collected at step (iv) of Algorithm 1. As a practical application, Kim and Liang (2025) applied EFI to quantify the uncertainty of individual treatment effects in causal inference.

Finally, we note that for a neural network model, its parameters are only unique up to certain loss-invariant transformations, such as reordering hidden neurons within the same hidden layer or simultaneously altering the sign or scale of certain connection weights (Sun et al., 2022). Therefore, for the $\boldsymbol{w}$-network, the consistency of $\boldsymbol{w}_n^*$ refers to its consistency with respect to one of the equivalent solutions to (3), while mathematically $\boldsymbol{w}_n^*$ can still be treated as unique.

# 3 EFI for Uncertainty Quantification in PINNs

## 3.1 EFI Formulation for PDEs

Consider a multidimensional dynamic process, $u(\boldsymbol{x})$, defined on a domain $\Omega \subset \mathbb{R}^d$ through a PDE:

$$\mathcal{F}(u(\boldsymbol{x}); \boldsymbol{\beta}) = f(\boldsymbol{x}), \quad \boldsymbol{x} \in \Omega,$$
$$\mathcal{B}(u(\boldsymbol{x})) = b(\boldsymbol{x}), \quad \boldsymbol{x} \in \partial\Omega,$$

where $\boldsymbol{x} = (x_1, x_2, \ldots, x_{d-1}, t)^T \in \mathbb{R}^d$ indicates the space-time coordinate vector, $u(\cdot)$ represents the unknown solution, $\boldsymbol{\beta}$ are the parameters related to the physics, and $f$ and $b$ are called the physics term and initial/boundary term, respectively. The observations are given in the forms $\{\boldsymbol{x}_i^u, u_i\}_{i=1}^{n_u}$, $\{\boldsymbol{x}_i^f, f_i\}_{i=1}^{n_f}$, and $\{\boldsymbol{x}_i^b, b_i\}_{i=1}^{n_b}$. Let $u_{\boldsymbol{\vartheta}}(\boldsymbol{x})$ denote the DNN approximation to the solution $u(\boldsymbol{x})$, where $\boldsymbol{\vartheta}$ denotes the DNN parameters. In data-generating equations, the observations can be expressed as

$$u_i = u_{\boldsymbol{\vartheta}}(\boldsymbol{x}_i^u) + z_i^u, \quad i = 1, 2, \ldots, n_u,$$
$$f_i = \mathcal{F}(u_{\boldsymbol{\vartheta}}(\boldsymbol{x}_i^f); \boldsymbol{\beta}) + z_i^f, \quad i = 1, 2, \ldots, n_f, \tag{9}$$
$$b_i = \mathcal{B}(u_{\boldsymbol{\vartheta}}(\boldsymbol{x}_i^b)) + z_i^b, \quad i = 1, 2, \ldots, n_b,$$

where $z_i^u$, $z_i^f$, and $z_i^b$ are independent Gaussian random errors with zero mean. Our objective is to infer $u$ and/or $\boldsymbol{\beta}$, as well as to quantify their uncertainty, given the data and governing physical law. In physics, inferring $u$ with $\boldsymbol{\beta}$ known is termed the *forward problem*, while inferring $u$ when $\boldsymbol{\beta}$ is unknown is referred to as the *inverse problem*.

When applying EFI to address this problem, the EFI network comprises two DNNs. The first, referred to as the data modeling network, is used to approximate $u(\boldsymbol{x})$. The second, called the $\boldsymbol{w}$-network, is used to approximate the parameters of the data modeling network as well as other parameters in equation (9). For an illustration, see Figure 1. Specifically, for the inverse problem, the output of the $\boldsymbol{w}$-network corresponds to $\boldsymbol{\theta} = \{\boldsymbol{\vartheta}, \boldsymbol{\beta}\}$; for the forward problem, its output corresponds to $\boldsymbol{\theta} = \{\boldsymbol{\vartheta}\}$.

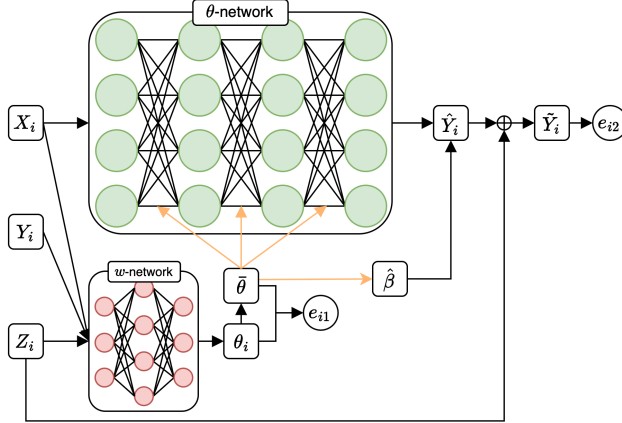

Figure 1: An EFI network with a double neural network (double-NN) structure.

Given the data-generating equations, the energy function for EFI can be defined as follows:

$$U_n(\boldsymbol{u}_n, \boldsymbol{f}_n, \boldsymbol{g}_n, \boldsymbol{z}_n, \boldsymbol{w}_n) = \eta_\theta \sum_{i=1}^{n} \|\hat{\boldsymbol{\theta}}_i - \bar{\boldsymbol{\theta}}\|^2 + \eta_u \sum_{i=1}^{n_u} \|u_i - u_{\boldsymbol{\vartheta}}(\boldsymbol{x}_i^u) - z_i^u\|^2$$
$$+ \eta_f \sum_{i=1}^{n_f} \|f_i - \mathcal{F}(u_{\boldsymbol{\vartheta}}(\boldsymbol{x}_i^f); \boldsymbol{\beta}) - z_i^f\|^2 + \eta_b \sum_{i=1}^{n_b} \|b_i - \mathcal{B}(u_{\boldsymbol{\vartheta}}(\boldsymbol{x}_i^b)) - z_i^b\|^2, \tag{10}$$

where $\boldsymbol{u}_n = (u_1, \ldots, u_n)^T$, $\boldsymbol{f} = (f_1, \ldots, f_n)^T$, $\boldsymbol{b}_n = (b_1, \ldots, b_n)^T$, $\boldsymbol{z}_n = (z_1^u, \ldots, z_n^u; z_1^f, \ldots, z_n^f; z_n^b, \ldots, z_n^b)^T$; $n = \#\{z_i^u \neq 0, z_i^f \neq 0, z_i^b \neq 0\}$ denotes the total number of noisy observations in the dataset; and $\eta_\theta$, $\eta_u$, $\eta_f$, and $\eta_b$ are belief weights for balancing

different terms. Fortunately, as shown in Liang et al. (2025), the choices for these terms will not affect much the performance of the algorithm as long as $\epsilon \to 0$.

Using EFI to solve PINNs, if we can correctly impute $\boldsymbol{Z}_n$ and consistently estimate the inverse function $G(\cdot)$, the uncertainty of $\boldsymbol{\theta}$ can be accurately quantified according to (8). However, consistent estimation of the inverse function is unattainable under the current EFI theoretical framework due to the high dimensionality of $\boldsymbol{\theta}$, which often far exceeds the sample size $n$. This limitation arises from the existing sparse deep learning theory (Sun et al., 2022), which constrains the dimension of $\boldsymbol{\theta}$ to remain fixed or grow very slowly with $n$. To address this challenge, we propose a new theoretical framework for EFI, as detailed in Section 3.2, which extends EFI to accommodate large-scale models and addresses the constraints of the current framework.

## 3.2  A New Theoretical Framework of EFI for Large-Scale Models

EFI treats the random errors in observations as latent variables. Consequently, training the $\boldsymbol{w}$-network is reduced to a problem of parameter estimation with missing data. Under a Bayesian setting, Liang et al. (2025) addressed the problem by solving equation (3) and employ Algorithm 1 for the solution. By imposing regularity conditions such as smoothness and dissipativity (Raginsky et al., 2017), Liang et al. (2025) established the following convergence results for Algorithm 1:

**Lemma 3.1.** *(Theorem 4.1 and Theorem 4.2, (Liang et al., 2025)) Suppose the regularity conditions in Liang et al. (2025) hold, and the learning rate sequence $\{v_k : k = 1, 2, \ldots\}$ and the step size sequence $\{\gamma_k : k = 1, 2, \ldots\}$ are set as: $v_k = \frac{C_v}{c_v + k^\alpha}$ and $\gamma_k = \frac{C_\gamma}{c_\gamma + k^\beta}$ for some constants $C_v > 0$, $c_v > 0$, $C_\gamma > 0$ and $c_\gamma > 0$, and $\alpha, \beta \in (0, 1]$ satisfying $\beta \leq \alpha \leq \min\{1, 2\beta\}$.*

*(i) (Root Consistency) There exists a root $\boldsymbol{w}_n^* \in \{\boldsymbol{w} : \nabla_{\boldsymbol{w}} \log \pi(\boldsymbol{w} | \boldsymbol{X}_n, \boldsymbol{Y}_n) = 0\}$ such that*

$$E\|\boldsymbol{w}_n^{(k)} - \boldsymbol{w}_n^*\| \leq \zeta \gamma_k, \quad k \geq k_0,$$

*for some constant $\zeta > 0$ and iteration number $k_0 > 0$.*

*(ii) (Weak Convergence of Latent Variables) Let $\pi_z^* = \pi(\boldsymbol{Z}_n | \boldsymbol{X}_n, \boldsymbol{Y}_n, \boldsymbol{w}_n^*)$ and $T_k = \sum_{i=0}^{k-1} v_{i+1}$, and let $\pi_z^{(T_k)}$ denote the probability law of $\boldsymbol{Z}_n^{(k)}$. Then,*

$$\mathbb{W}_2(\pi_z^{(T_k)}, \pi_z^*) \leq (c_0 \delta_g^{1/4} + c_1 \gamma_1^{1/4}) T_k + c_2 e^{-T_k/c_{LS}},$$

*for any $k \in \mathbb{N}$, where $\mathbb{W}_2(\cdot, \cdot)$ is the 2-Wasserstein distance, $c_0$, $c_1$, and $c_2$ are some positive constants, $c_{LS}$ is the logarithmic Sobolev constant of $\pi_z^*$, and $\delta_g$ is a coefficient reflecting the variation of the stochastic gradient used in latent variable imputation step.*

In our implementation, the latent variable imputation step is done for each observation separately, ensuring $\delta_g = 0$. We choose $\alpha \in (0, 1]$ and set $\gamma_1 \prec \frac{1}{T_k^4}$ for any $T_k$, which ensures $\mathbb{W}_2(\pi_z^{(T_k)}, \pi_z^*) \to 0$ as $k \to \infty$. It is worth noting that Lemma 3.1 holds regardless of the size of the $\boldsymbol{w}$-network. Thus, it remains valid for large-scale models. However, in this work, we do not impose any priors on $\boldsymbol{w}_n$, which corresponds to the non-informative prior setting $\pi(\boldsymbol{w}_n) \propto 1$.

Given the root consistency result, it is still necessary to establish that the resulting inverse function estimator (7) is consistent with respect to the true parameter $\boldsymbol{\theta}^*$ to ensure valid downstream inference. Liang et al. (2025) established this consistency using sparse deep learning theory (Sun et al., 2022), but their approach was limited to settings where the dimension of $\boldsymbol{\theta}$ is fixed or increases very slowly with the sample size $n$. In this work, we achieve the consistency by employing a narrow-neck $\boldsymbol{w}$-network, which overcomes the limitation of the current framework of EFI.

To motivate the development of the narrow-neck $\boldsymbol{w}$-network, we first note an important mathematical fact: As implied by (A6), each $\hat{\boldsymbol{\theta}}_i \in \mathbb{R}^p$ in the EFI network tends to converge to a constant vector as $\epsilon \to 0$. Consequently, different components of $\hat{\boldsymbol{\theta}}$ become highly correlated across $n$ observations and $\hat{\boldsymbol{\theta}}$ can be effectively represented in a much lower-dimensional space, even though the dimension of $\boldsymbol{\theta}$ may be very high. A straightforward solution to address this issue is to incorporate a restricted Boltzmann machine (RBM) (Hinton and Salakhutdinov, 2006) into the $\boldsymbol{w}$-network, as illustrated in Figure A2, where the neck layer is binary and the last two layers form a Gaussian-binary RBM (Gu et al., 2022; Chu et al., 2017). As discussed in Liang et al. (2025), $\hat{\boldsymbol{\theta}}$ can be treated as Gaussian

random variables in the EFI network. The binary layer of the RBM serves as a dimensionality reducer for $\hat{\boldsymbol{\theta}}$. Such a RBM-embedded $\boldsymbol{w}$-network can be trained using the imputation-regularized optimization (IRO) algorithm (Liang et al., 2018a) in a similar way to that used in Wu et al. (2019).

Since the primary role of the $\boldsymbol{w}$-network is forward learning of $\boldsymbol{\theta}$, it can also be formulated as a stochastic neural network (StoNet) (Liang et al., 2022; Sun and Liang, 2022), where the binary layer can be extended to be continuous. This StoNet-formulation leads to the following hierarchical model:

$$
\begin{aligned}
\hat{\theta}_i^{(j)} &= \boldsymbol{m}_i^T \boldsymbol{\xi}_j + e_{i,j}, \\
\boldsymbol{m}_i &= R(\boldsymbol{\mu}_i, \boldsymbol{v}_i),
\end{aligned}
\tag{11}
$$

where $i = 1, 2, \ldots, n$, $j = 1, 2, \ldots, p$, $e_{i,j} \sim N(0, \sigma_{\hat{\theta}}^2)$, $\boldsymbol{v}_i \in \mathbb{R}^{d_h}$ denotes a vector of random errors following a known distribution, $d_h$ denotes the width of the stochastic neck layer, $h$ denotes the number of hidden layers of the $\boldsymbol{w}$-network, $\boldsymbol{\mu}_i = g(X_i, Y_i, Z_i; \boldsymbol{w}_n^{(1)})$ is the mean of the feeding vector to the stochastic neck layer, and $R(\cdot)$ represents a transformation. With a slight abuse of notation, we assume that $\boldsymbol{m}_i$ has been augmented with a constant component to account for the intercept term in the regression model for $\hat{\theta}_i^{(j)}$. For $R(\cdot)$, we recommend the setting:

(*) *The neck layer is stochastic with:* $R(\boldsymbol{\mu}_i, \boldsymbol{v}_i) = \Psi(\boldsymbol{\mu}_i + \boldsymbol{v}_i)$, *where* $\boldsymbol{v}_i \sim N(0, \sigma_v^2)$ *with a pre-specified value of* $\sigma_v^2$, *and* $\Psi$ *is an (element-wise) activation function.*

Under this setting, the $\boldsymbol{w}^{(1)}$-network forms a nonlinear Gaussian regression with response $\boldsymbol{\mu}_i + \boldsymbol{v}_i$ for $i = 1, 2, \ldots, n$. Conceptually, the StoNet can be trained using the stochastic EM algorithm (Celeux et al., 1996) by iterating between the steps: (i) *Latent variable imputation*: Impute the latent variables $\{\boldsymbol{v}_i : i = 1, \ldots, n\}$ and $\boldsymbol{Z}_n = \{z_1, \ldots, z_n\}$ conditioned on the current estimates of $\boldsymbol{w}_n = \{\boldsymbol{w}_n^{(1)}, \boldsymbol{w}_n^{(2)}\}$, where $\boldsymbol{w}_n^{(2)} = \{\boldsymbol{\xi}_j : j = 1, 2, \ldots, p\}$. (ii) *Optimization*: Conditioned on the imputed latent variables, update the estimates of $\boldsymbol{w}_n^{(1)}$ and $\boldsymbol{w}_n^{(2)}$ separately. Specifically, $\boldsymbol{w}_n^{(1)}$ can be estimated by training the $\boldsymbol{w}_n^{(1)}$-network using SGD, and $\boldsymbol{w}_n^{(2)}$ can be estimated by performing $p$ linear regressions as specified in (11).

Following the standard theory of the stochastic EM algorithm (Nielsen, 2000; Liang et al., 2018a), $\{\boldsymbol{w}_n^{(1)}, \boldsymbol{w}_n^{(2)}\}$ will converge to a solution to the equation: $\nabla_{\boldsymbol{w}_n} \log \pi_\epsilon(\boldsymbol{w}_n | \boldsymbol{X}_n, \boldsymbol{Y}_n) = \int \left[ \nabla_{\boldsymbol{w}_n} \log \pi_\epsilon(\boldsymbol{w}_n | \boldsymbol{X}_n, \boldsymbol{Y}_n, \boldsymbol{Z}_n, \boldsymbol{V}_n) \pi_\epsilon(\boldsymbol{Z}_n, \boldsymbol{V}_n | \boldsymbol{X}_n, \boldsymbol{Y}_n, \boldsymbol{w}_n) \right] d\boldsymbol{Z}_n d\boldsymbol{V}_n = 0$, as the sample size $n$ and the number of iterations of the algorithm become large, where the prior $\pi(\boldsymbol{w}_n) \propto 1$ and $\boldsymbol{V}_n = (\boldsymbol{v}_1, \boldsymbol{v}_2, \ldots, \boldsymbol{v}_n)^T$. Denote the converged solution by $\boldsymbol{w}_n^* = \{\boldsymbol{w}_n^{*(1)}, \boldsymbol{w}_n^{*(2)}\}$. The consistency of the resulting inverse function estimator can be established by leveraging the sufficient dimension reduction property of the StoNet (Liang et al., 2022) and the prediction property of linear regressions. Specifically, $\{\boldsymbol{m}_i : i = 1, \ldots, n\}$ serves as a sufficient dimension reduction for $\{(X_i, Y_i, Z_i) : i = 1, \ldots, n\}$. The consistency of the inverse mapping can thus be ensured by the linear relationship $\hat{\boldsymbol{\theta}}_i \sim \boldsymbol{m}_i$, as described in (11), along with the prediction property of linear regression. Note that the linear relationship $\hat{\boldsymbol{\theta}}_i \sim \boldsymbol{m}_i$ can be generally ensured by the universal approximation ability of the DNN.

Toward a rigorous mathematical development, we impose the following conditions:

$$
(i)\ \lambda_{\min}(\mathbb{E}(\boldsymbol{m}_i \boldsymbol{m}_i^T)) \geq \rho_{\min} := c' \sigma_v^2 + \delta > 0, \quad (ii)\ \lambda_{\max}(\mathbb{E}(\boldsymbol{m}_i \boldsymbol{m}_i^T)) \leq \rho_{\max} < \infty, \tag{12}
$$

where $\mathbb{E}(\cdot)$ denotes expectation; $\lambda_{\min}(\cdot)$ and $\lambda_{\max}(\cdot)$ denote the minimum and maximum eigenvalue of a matrix, respectively; and $c' > 0$, $\delta > 0$, and $\rho_{\max} > 0$ are some constants. Condition (i) is generally satisfied by choosing a sufficiently narrow neck layer, in particular, we set $d_h \prec n$. Condition (ii) is justified in Appendix A4.2.

By the asymptotic equivalence between the StoNet and conventional DNN (Liang et al., 2022) (see also Appendix A4.1), the StoNet can be trained by directly training the DNN. Based on this asymptotic equivalence, Theorem 3.2 establishes the consistency of the inverse mapping learned by Algorithm 1 for large-scale models, see Appendix A4.3 for the proof.

**Theorem 3.2.** *Suppose that the narrow neck layer is set as in (*) with* $\sigma_v^2 \prec \frac{\epsilon}{\eta h d_h p}$, *the activation function* $\Psi(\cdot)$ *is c-Lipschitz continuous, and* $d_h \prec n$ *is sufficiently small such that the conditions in*

*([12](#)) are satisfied while admitting a non-empty zero energy set $\mathcal{Z}_n$. Additionally, assume that the other regularity conditions (Assumptions [A4.1](#)-[A4.2](#) in Appendix [A4](#)) hold. If $\epsilon \prec \min\{\frac{n}{p^2 d_h^2}, \frac{h}{p d_h}\}$, then the inverse mapping $\boldsymbol{\theta} = G(\boldsymbol{Y}_n, \boldsymbol{X}_n, \boldsymbol{Z}_n)$, learned by Algorithm [1](#) with a narrow neck $\boldsymbol{w}$-network, is consistent.*

This narrow neck setting for the $\boldsymbol{w}$-network eliminates the need to specify a prior for $\boldsymbol{w}_n^{(1)}$. Notably, under this setting, the resulting estimate $\boldsymbol{\theta}$ is not necessarily sparse, introducing a new research paradigm for high-dimensional problems; moreover, the $\boldsymbol{w}_n^{(1)}$-network does not need to be excessively large, keeping the overall size of the $\boldsymbol{w}$-network manageable. This facilitates the application of EFI to high-dimensional and complex models.

## 4 Simulation Studies

We compared EFI with Bayesian and dropout methods across multiple simulation studies, including the Poisson equation under various settings and the Black-Scholes model. Below, we present the results for the 1-D Poisson equation, with results from other studies provided in Appendix [A5](#). The experimental settings for all simulation studies are detailed in Appendix [A6](#).

### 4.1 1-D Poisson Equation

Consider a 1-D Poisson equation as in [Yang et al. (2021)](#):

$$\beta \frac{\partial^2 u}{\partial x^2} = f, \quad x \in \Omega, \tag{13}$$

where $\Omega = [-0.7, 0.7]$, $\beta = 0.01$, $u = \sin^3(6x)$, and $f$ can be derived from ([13](#)). Here, we assume the analytical expression of $f$ is unavailable; instead, 200 sensor measurements of $f$ are available with the sensors equidistantly distributed across $\Omega$. Additionally, there are two sensors at $x = -0.7$ and $x = 0.7$ to provide the left/right Dirichlet boundary conditions for $u$. We model the boundary noise as $z_i^u \sim N(0, 0.05^2)$ for $i = 1, 2, \ldots, 20$, with 10 observations drawn from each boundary sensor. For the interior domain, we assume noise-free measurements of $f$. This simulation is repeated over 100 independent datasets to evaluate the robustness and reliability of the proposed approach.

The Bayesian method was first applied to this example as in [Yang et al. (2021)](#) using a two-hidden-layer DNN, with 50 hidden units in each layer, to approximate $u(\boldsymbol{x})$. The same network architecture was also used for this example by other methods as shown in Table [1](#). The negative log-posterior is given as follows:

$$\ell(\boldsymbol{\vartheta}) = \sum_{i=1}^{n_u} \frac{\|u_i - u_{\boldsymbol{\vartheta}}(\boldsymbol{x}_i^u)\|^2}{2\sigma_u^2} + \sum_{i=1}^{n_f} \frac{\|f_i - \mathcal{F}(u_{\boldsymbol{\vartheta}}(\boldsymbol{x}_i^f); \boldsymbol{\beta})\|^2}{2\sigma_f^2} + \sum_{i=1}^{n_b} \frac{\|b_i - \mathcal{B}(u_{\boldsymbol{\vartheta}}(\boldsymbol{x}_i^b))\|^2}{2\sigma_b^2} \tag{14}$$
$$- \log(\pi(\boldsymbol{\vartheta})) + C,$$

where $C$ denotes the log-normalizing constant, $n_u = 20$, $n_f = 200$, $n_b = 0$, and $\sigma_u = 0.05$. The prior $\pi(\boldsymbol{\vartheta})$ is specified as in [Yang et al. (2021)](#), where each unknown parameter is assigned an independent standard Gaussian distribution. The same prior is used across all Bayesian simulations in this paper. We implement the method using the Python package *hamiltorch* ([Cobb and Jalaian, 2021](#)). Since $f$ is observed exactly, the corresponding variance $\sigma_f$ should theoretically be zero. However, the formulation in ([14](#)) does not allow $\sigma_f = 0$, necessitating experimentation with different non-zero values of $\sigma_f$. These variations resulted in different widths of confidence intervals, as illustrated in Figure [A4](#). Notably, there is no clear guideline for selecting $\sigma_f$ to achieve the appropriate interval width necessary for the desired coverage rate. This ambiguity underscores the dilemma (mentioned in Introduction) inherent in the Bayesian method. Specifically, the $f$-term in ([14](#)) functions as part of the prior for the parameter $\boldsymbol{\vartheta}$.

A similar issue arises when selecting the dropout rate for PINNs. As illustrated in Figure [A3](#), the uncertainty estimation in PINNs is highly sensitive to the choice of the dropout rate. When the dropout rate approaches zero, the confidence interval collapses into a point estimate, failing to capture uncertainty. Conversely, excessively large dropout rates introduce significant bias, leading to unreliable interval estimates. This highlights the challenge of determining an appropriate dropout rate to balance bias and variability in uncertainty quantification. As a possible way to address this issue,

Concrete Dropout (Gal et al., 2017) was implemented for the example with the code provided at `https://github.com/yaringal/ConcreteDropout`. Unlike the conventional dropout method using fixed dropout rate, Concrete Dropout treats the dropout rate as a hyperparameter and learns it simultaneously when training the model.

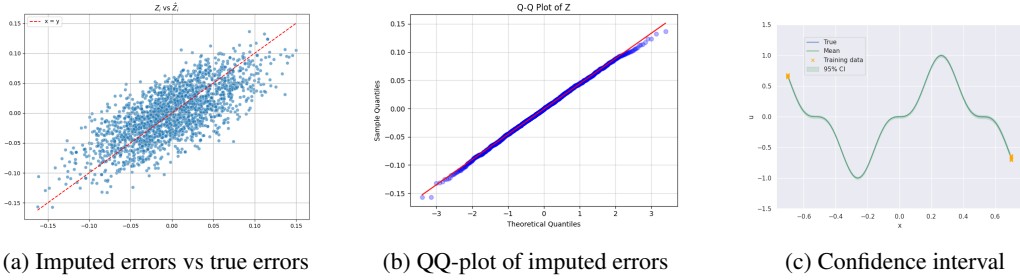

(a) Imputed errors vs true errors   (b) QQ-plot of imputed errors   (c) Confidence interval

Figure 2: EFI-PINN diagnostic for 1D-Poisson

We then applied EFI to this example, using the same DNN structure for the data-modeling network. Figure 2 summarizes the imputed random errors by EFI for 100 datasets, showing that EFI correctly imputes the realized random errors from the observations. Consequently, EFI achieves the correct recovery of the underlying physical law.

Table 1: Metrics for 1D-Poisson, averaged over 100 runs.

| Method | MSE | Coverage Rate | CI-Width |
|---|---|---|---|
| PINN (no dropout) | 0.000121 (0.000012) | 0.0757 (0.011795) | 0.002139 (0.000024) |
| Dropout (0.5%) | 0.000228 (0.000024) | 1.0000 (0.000000) | 0.191712 (0.004367) |
| Dropout (1%) | 0.000851 (0.000111) | 0.9999 (0.000104) | 0.274651 (0.006734) |
| Dropout (5%) | 0.006276 (0.000982) | 0.9893 (0.004024) | 0.660927 (0.030789) |
| Concrete Dropout | 0.000184 (0.000020) | 0.1185 (0.018002) | 0.003706 (0.000073) |
| Bayesian ($\sigma_f = 0.05$) | 0.000826 (0.000102) | 0.9997 (0.000302) | 0.372381 (0.002635) |
| Bayesian ($\sigma_f = 0.005$) | 0.000170 (0.000020) | 0.9914 (0.008600) | 0.081557 (0.000692) |
| Bayesian ($\sigma_f = 0.0005$) | 0.001460 (0.000219) | 0.5884 (0.032390) | 0.055191 (0.001123) |
| **EFI** | **0.000148 (0.000016)** | **0.9517 (0.014186)** | **0.050437 (0.000462)** |

We evaluated the accuracy and robustness of each method by recording three metrics, as summarized in Table 1, based on an average across 100 repeated experiments. The 'MSE' measures the mean squared distance between the network prediction and the true solution. The coverage rate represents the percentage of the true solution contained within the 95% confidence interval, while the CI-Width quantifies the corresponding interval width. As discussed above, the dropout method produces inflated coverage rates and excessively wide confidence intervals, with CI widths ranging from 0.1917 to 0.6609. Similarly, the Bayesian method is sensitive to the choice of $\sigma_f$, with larger $\sigma_f$ values yielding inflated coverage rates, while smaller $\sigma_f$ values result in underestimated coverage rates. In contrast, EFI is free of hyperparameter tuning, making inference on the model parameters based on the observations and the embedded physical law. It produces the most balanced results, with a mean squared error (MSE) of $1.48 \times 10^{-4}$ and a coverage rate of 95.17%, closely aligning with the nominal 95% target. Moreover, it provides the shortest CI-width 0.0504. Concrete Dropout achieves a comparable MSE as EFI, indicating similar model estimation accuracy. However, it performs significantly worse in terms of uncertainty quantification, with a coverage rate of only 11.85%, far below the nominal 95%. This substantial under-coverage is due to an underestimation of predictive uncertainty, as evidenced by the markedly narrower confidence interval width.

## 5   Real Data Examples

This section demonstrates the abiliy of the EFI-PINN framework to quantify uncertainty for models learned from real data. We considered two models: the Montroll growth model,

$$\frac{du}{dt} = k \cdot u \cdot \left(1 - \left(\frac{u}{C}\right)^{\theta}\right), \tag{15}$$

where $k$, $C$, and $\theta$ are unknown parameters; and the reaction-diffusion model governed by the generalized Porous-Fisher-Kolmogoriv-Petrovsky-Piskunov (P-FKPP) equation:

$$\frac{du}{dt} = D\frac{\partial}{\partial x}\left[\left(\frac{u}{K}\right)^m \frac{\partial u}{\partial x}\right] + ru\left(1 - \frac{u}{K}\right), \tag{16}$$

where $D$, $m$, and $r$ are unknown parameters. We modeled the Chinese hamster V79 fibroblast tumor cell growth data (Rodrigues, 2024) using equation (15), and the scratch assay data (Jin and Cai, 2006) using equation (16). Additional details about the datasets are provided in the Appendix. Figure 3 illustrates that the proposed EFI-PINN framework is not only capable of learning the PDE models from data but also effectively quantifying the uncertainty of the models. Further results and discussions can be found in the Appendix. The prediction uncertainty can also be quantified with the proposed method.

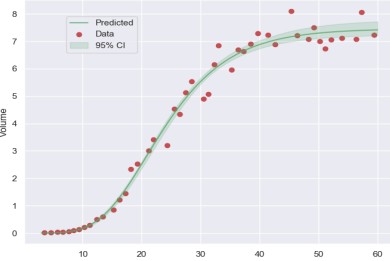 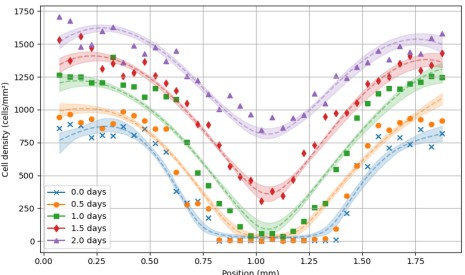

Figure 3: Confidence bands (shaded areas) for the learned models: (left): Montroll growth model; (right): generalized P-FKPP model.

## 6 Conclusion

This paper presents a novel theoretical framework for EFI, enabling effective uncertainty quantification for PINNs. EFI addresses the challenge of uncertainty quantification through a unique approach of "solving data-generating equations," transforming it into an objective process that facilitates the construction of honest confidence sets. In contrast, existing methods such as dropout and Bayesian approaches rely on subjective hyperparameters (e.g., dropout rates or priors), undermining the honesty of the resulting confidence sets. This establishes a new research paradigm for statistical inference of complex models, with the potential to significantly impact the advancement of modern data science.

Although Algorithm 1 performs well in our examples, its efficiency can be further improved through several enhancements. For instance, in the latent variable imputation step, the SGLD algorithm (Welling and Teh, 2011) can be replaced with the Stochastic Gradient Hamiltonian Monte Carlo (Chen et al., 2014), which offers better sampling efficiency. Similarly, in the parameter updating step, the SGD algorithm can be accelerated with momentum. Momentum-based algorithms facilitate faster convergence to a good local minimum during the early stages of training. As training progresses, the momentum can be gradually reduced to zero, ensuring alignment with the convergence theory of EFI. This approach balances computational efficiency with theoretical rigor, enhancing the overall performance of EFI.

While the examples presented focus on relatively small-scale models, this is not a limitation of the approach. In principle, EFI can be extended to large-scale models, such as ResNets and CNNs, through transfer learning—a direction we plan to explore in future work.

## Acknowledgments

Liang's research is supported in part by the NSF grant DMS-2210819 and the NIH grant R01-GM152717.
Shih's research is partially supported by MSK Cancer Center Support Grant/Core Grant (P30 CA008748).

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

# Appendix

This appendix is organized as follows. Section A1 provides a brief review of the EFI method and its self-diagnostic property. Section A2 presents the analytical expression of the extended fiducial density (EFD) function of $\boldsymbol{Z}_n$. Section A3 includes an illustrative plot of the proposed narrow-neck $\boldsymbol{w}$-network. Section A4 gives the proof of Theorem 3.2. Section A5 presents additional numerical results. Finally, Section A6 details the parameter settings used in our numerical experiments.

## A1    A Brief Review of the EFI Method and Its Self-Diagnosis

### A1.1    The EFI Method

To ensure a smooth presentation of the EFI method, this section partially overlaps with Section 2 of the main text. Consider a regression model:

$$Y = f(\boldsymbol{X}, Z, \boldsymbol{\theta}),$$

where $Y \in \mathbb{R}$ and $\boldsymbol{X} \in \mathbb{R}^d$ represent the response and explanatory variables, respectively; $\boldsymbol{\theta} \in \mathbb{R}^p$ represents the vector of parameters; and $Z \in \mathbb{R}$ represents a scaled random error following a known distribution $\pi_0(\cdot)$. Suppose that a random sample of size $n$, denoted by $\{(y_1, \boldsymbol{x}_1), (y_2, \boldsymbol{x}_2), \ldots, (y_n, \boldsymbol{x}_n)\}$, has been collected from the model. In structural inference, the observations can be expressed in data-generating equations as follows:

$$y_i = f(\boldsymbol{x}_i, z_i, \boldsymbol{\theta}), \quad i = 1, 2, \ldots, n. \tag{A1}$$

This system of equations consists of $n + p$ unknowns, namely, $\{\boldsymbol{\theta}, z_1, z_2, \ldots, z_n\}$, while there are only $n$ equations. Therefore, the values of $\boldsymbol{\theta}$ cannot be uniquely determined by the data-generating equations, and this lack of uniqueness of unknowns introduces uncertainty in $\boldsymbol{\theta}$.

Let $\boldsymbol{Z}_n = \{z_1, z_2, \ldots, z_n\}$ denote the unobservable random errors contained in the data, which are also called latent variables in EFI. Let $G(\cdot)$ denote an inverse function/mapping for $\boldsymbol{\theta}$, i.e.,

$$\boldsymbol{\theta} = G(\boldsymbol{Y}_n, \boldsymbol{X}_n, \boldsymbol{Z}_n). \tag{A2}$$

It is worth noting that the inverse function is generally non-unique. For example, it can be constructed by solving any $p$ equations in (1) for $\boldsymbol{\theta}$. As noted by Liang et al. (2025), this non-uniqueness of inverse function mirrors the flexibility of frequentist methods, where different estimators of $\boldsymbol{\theta}$ can be constructed to achieve desired properties such as efficiency, unbiasedness, and robustness.

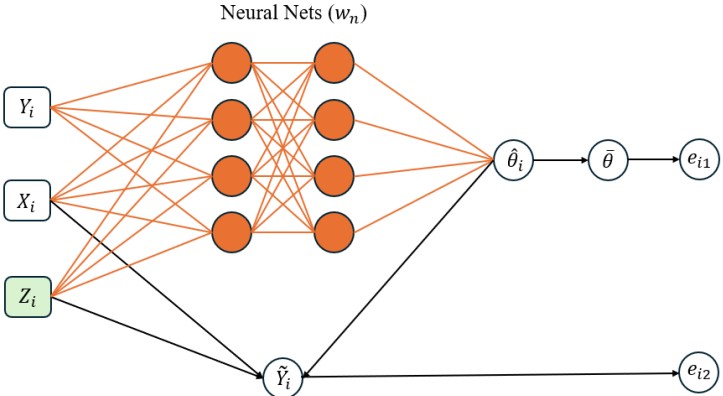

Figure A1: Diagram of EFI given in Liang et al. (2025): The orange nodes and orange links form a deep neural network (DNN), referred to as the $\boldsymbol{w}$-network, which is parameterized by $\boldsymbol{w}_n$ (with the subscript $n$ indicating its dependence on the training sample size $n$); the green node represents the latent variable to impute; and the black lines represent deterministic functions.

Since the inverse function $G(\cdot)$ is generally unknown, Liang et al. (2025) proposed to approximate it using a sparse DNN, see Figure A1 for illustration. The EFI network has two output nodes defined,

respectively, by

$$e_{i,1} := \|\hat{\boldsymbol{\theta}}_i - \bar{\boldsymbol{\theta}}\|^2,$$
$$\text{and} \quad e_{i,2} := d(y_i, \tilde{y}_i) := d(y_i, \boldsymbol{x}_i, z_i, \bar{\boldsymbol{\theta}}), \tag{A3}$$

where $\tilde{y}_i = f(\boldsymbol{x}_i, z_i, \bar{\boldsymbol{\theta}})$, $f(\cdot)$ is as specified in (1), and $d(\cdot)$ is a function measuring the difference between $y_i$ and $\tilde{y}_i$. That is, the node $e_{i,1}$ quantifies the variation of $\hat{\boldsymbol{\theta}}_i$, while the node $e_{i,2}$ represents the fitting error. For a normal linear/nonlinear regression, $d(\cdot)$ can be defined as

$$d(y_i, \boldsymbol{x}_i, z_i, \bar{\boldsymbol{\theta}}) = \|y_i - f(\boldsymbol{x}_i, z_i, \bar{\boldsymbol{\theta}})\|^2. \tag{A4}$$

For logistic regression, it is defined as a ReLU function, see Liang et al. (2025) for details.

Let $\hat{\boldsymbol{\theta}}_i := \hat{g}(y_i, \boldsymbol{x}_i, z_i, \boldsymbol{w}_n)$ denote the DNN prediction function parameterized by the weights $\boldsymbol{w}_n$ in the EFI network, and let

$$\bar{\boldsymbol{\theta}} := \frac{1}{n} \sum_{i=1}^{n} \hat{\boldsymbol{\theta}}_i = \frac{1}{n} \sum_{i=1}^{n} \hat{g}(y_i, \boldsymbol{x}_i, z_i, \boldsymbol{w}_n), \tag{A5}$$

which serves as an estimator of the inverse function $G(\cdot)$.

EFI defines an energy function

$$U_n(\boldsymbol{Y}_n, \boldsymbol{X}_n, \boldsymbol{Z}_n, \boldsymbol{w}_n) = \eta_\theta \sum_{i=1}^{n} \|\hat{\boldsymbol{\theta}}_i - \bar{\boldsymbol{\theta}}\|^2 + \sum_{i=1}^{n} d(y_i, \boldsymbol{x}_i, z_i, \bar{\boldsymbol{\theta}}), \tag{A6}$$

where $\eta_\theta > 0$ is a regularization parameter, $\hat{\boldsymbol{\theta}}_i$'s and $\bar{\boldsymbol{\theta}}$ can be expressed as functions of $(\boldsymbol{Y}_n, \boldsymbol{X}_n, \boldsymbol{Z}_n, \boldsymbol{w}_n)$, and $d(\cdot)$ is a function measuring the difference between $y_i$ and $\tilde{y}_i$. The likelihood function is given by

$$\pi_\epsilon(\boldsymbol{Y}_n | \boldsymbol{X}_n, \boldsymbol{Z}_n, \boldsymbol{w}_n) \propto e^{-U_n(\boldsymbol{Y}_n, \boldsymbol{X}_n, \boldsymbol{Z}_n, \boldsymbol{w}_n)/\epsilon}, \tag{A7}$$

for some constant $\epsilon$ close to 0. As discussed in Liang et al. (2025), the choice of $\eta_\theta$ does not affect much on the performance of EFI as long as $\epsilon$ is sufficiently small. Subsequently, the posterior of $\boldsymbol{w}_n$ is given by

$$\pi_\epsilon(\boldsymbol{w}_n | \boldsymbol{X}_n, \boldsymbol{Y}_n, \boldsymbol{Z}_n) \propto \pi(\boldsymbol{w}_n) e^{-U_n(\boldsymbol{Y}_n, \boldsymbol{X}_n, \boldsymbol{Z}_n, \boldsymbol{w}_n)/\epsilon},$$

where $\pi(\boldsymbol{w}_n)$ denotes the prior of $\boldsymbol{w}_n$; and the predictive distribution of $\boldsymbol{Z}_n$ is given by

$$\pi_\epsilon(\boldsymbol{Z}_n | \boldsymbol{X}_n, \boldsymbol{Y}_n, \boldsymbol{w}_n) \propto \pi_0^{\otimes n}(\boldsymbol{Z}_n) e^{-U_n(\boldsymbol{Y}_n, \boldsymbol{X}_n, \boldsymbol{Z}_n, \boldsymbol{w}_n)/\epsilon},$$

where $\pi_0^{\otimes n}(\boldsymbol{Z}_n) = \prod_{i=1}^{n} \pi_0(z_i)$ under the assumption that $z_i$'s are independently identically distributed (i.i.d.). In EFI, $\boldsymbol{w}_n$ is estimated through maximizing the posterior $\pi_\epsilon(\boldsymbol{w}_n | \boldsymbol{X}_n, \boldsymbol{Y}_n)$ given the observations $\{\boldsymbol{X}_n, \boldsymbol{Y}_n\}$. By the Bayesian version of Fisher's identity (Song et al., 2020), the gradient equation $\nabla_{\boldsymbol{w}_n} \log \pi_\epsilon(\boldsymbol{w}_n | \boldsymbol{X}_n, \boldsymbol{Y}_n) = 0$ can be re-expressed as

$$\nabla_{\boldsymbol{w}_n} \log \pi_\epsilon(\boldsymbol{w}_n | \boldsymbol{X}_n, \boldsymbol{Y}_n) = \int \Big[ \nabla_{\boldsymbol{w}_n} \log \pi_\epsilon(\boldsymbol{w}_n | \boldsymbol{X}_n, \boldsymbol{Y}_n, \boldsymbol{Z}_n)$$
$$\times \pi_\epsilon(\boldsymbol{Z}_n | \boldsymbol{X}_n, \boldsymbol{Y}_n, \boldsymbol{w}_n) \Big] d\boldsymbol{Z}_n = 0, \tag{A8}$$

which can be solved using an adaptive stochastic gradient MCMC algorithm (Liang et al., 2022; Deng et al., 2019). The algorithm works by iterating between the *latent variable imputation* and *parameter updating* steps, see Algorithm 1 for the pseudo-code. This algorithm is termed "adaptive" because the transition kernel in the latent variable imputation step changes with the working parameter estimate of $\boldsymbol{w}_n$. The parameter updating step can be implemented using mini-batch SGD, and the latent variable imputation step can be executed in parallel for each observation $(y_i, \boldsymbol{x}_i)$. Hence, the algorithm is scalable with respect to large datasets.

Under mild conditions for the adaptive SGLD algorithm, it can be shown that

$$\|\boldsymbol{w}_n^{(k)} - \boldsymbol{w}_n^*\| \xrightarrow{p} 0, \quad \text{as } k \to \infty, \tag{A9}$$

where $\boldsymbol{w}_n^*$ denotes a solution to equation (3) and $\xrightarrow{p}$ denotes convergence in probability, and that

$$\boldsymbol{Z}_n^{(k)} \xrightarrow{d} \pi_\epsilon(\boldsymbol{Z}_n | \boldsymbol{X}_n, \boldsymbol{Y}_n, \boldsymbol{w}_n^*), \quad \text{as } k \to \infty, \tag{A10}$$

in 2-Wasserstein distance, where $\overset{d}{\rightsquigarrow}$ denotes weak convergence. To study the limit of (6) as $\epsilon$ decays to 0, i.e.,

$$p_n^*(z|Y_n, X_n, w_n^*) = \lim_{\epsilon \downarrow 0} \pi_\epsilon(Z_n|X_n, Y_n, w_n^*),$$

where $p_n^*(z|Y_n, X_n, w_n^*)$ is referred to as the extended fiducial density (EFD) of $Z_n$, Liang et al. (2025) impose specific conditions on the structure of the $w$-network, including that the $w$-network is sparse and that the output layer width (i.e., the dimension of $\theta$) is either fixed or grows very slowly with the sample size $n$. Under these assumptions, they prove the consistency of $w_n^*$ based on the sparse deep learning theory developed in Sun et al. (2022). This consistency further implies that

$$G^*(Y_n, X_n, Z_n) = \frac{1}{n} \sum_{i=1}^n \hat{g}(y_i, x_i, z_i, w_n^*), \tag{A11}$$

serves as a consistent estimator for the inverse function/mapping $\theta = G(Y_n, X_n, Z_n)$. Refer to Appendix A2 for the analytic expression of $p_n^*(z|Y_n, X_n, w_n^*)$.

Let $\mathcal{Z}_n = \{z \in \mathbb{R}^n : U_n(Y_n, X_n, Z_n, w_n^*) = 0\}$ denote the zero-energy set. Under some regularity conditions on the energy function, Liang et al. (2025) proved that $\mathcal{Z}_n$ is invariant to the choice of $G(\cdot)$. Let $\Theta := \{\theta \in \mathbb{R}^p : \theta = G^*(Y_n, X_n, z), z \in \mathcal{Z}_n\}$ denote the parameter space of the target model, which represents the set of all possible values of $\theta$ that $G^*(\cdot)$ takes when $z$ runs over $\mathcal{Z}_n$. Then, for any function $b(\theta)$ of interest, its EFD $\mu_n^*(\cdot|Y_n, X_n)$ associated with $G^*(\cdot)$ is given by

$$\mu_n^*(B|Y_n, X_n) = \int_{\mathcal{Z}_n(B)} dP_n^*(z|Y_n, X_n, w_n^*), \tag{A12}$$

for any measurable set $B \subset \Theta$, where $\mathcal{Z}_n(B) = \{z \in \mathcal{Z}_n : b(G^*(Y_n, X_n, z)) \in B\}$, and $P_n^*(z|X_n, Y_n, w_n^*)$ denote the cumulative distribution function (CDF) corresponding to $p_n^*(z|X_n, Y_n, w_n^*)$. The EFD provides an uncertainty measure for $b(\theta)$. Practically, it can be constructed based on the samples $\{b(\bar{\theta}_1), b(\bar{\theta}_2), \dots, b(\bar{\theta}_M)\}$, where $\{\bar{\theta}_1, \bar{\theta}_2, \dots, \bar{\theta}_M\}$ denotes the fiducial $\bar{\theta}$-samples collected at step (iv) of Algorithm 1.

Finally, we note that for a neural network model, its parameters are only unique up to certain loss-invariant transformations, such as reordering hidden neurons within the same hidden layer or simultaneously altering the sign or scale of certain connection weights (Sun et al., 2022). Therefore, for the $w$-network, the consistency of $w_n^*$ refers to its consistency with respect to one of the equivalent solutions to (3), while mathematically $w_n^*$ can still be treated as unique.

### A1.2   Self-Diagnosis in EFI

Given the flexibility of DNN models, reliable diagnostics are crucial in deep learning to ensure model robustness and accuracy while identifying potential issues during training. Unlike dropout and Bayesian methods, which lack self-diagnostic capabilities, EFI includes a built-in mechanism for self-diagnosis. Specifically, this can be achieved through (i) analyzing the QQ-plot of the imputed random errors, and (ii) verifying that the energy function $U_n$ converges to zero.

According to Lemma 3.1, the imputed random errors $\hat{Z}_n$ should follow the same distribution as the true random errors $Z_n$. Since the theoretical distribution of $Z_n$ is known, the convergence of $\hat{Z}_n$ can be assessed using QQ-plot as shown in Figure 2(b). When $Z_n$ has been correctly imputed, the energy function must converge to zero to ensure the consistency of the inverse function estimator. In practice, we can check whether $U_n(Y_n, X_n, Z_n, w_n) = o(\epsilon)$ as $\epsilon \to 0$. The validity of inference for the model uncertainty can thus be ensured if both diagnostic tests are satisfied. This diagnostic method is entirely data-driven, offering a simple way for validating the EFI results.

If the diagnostic tests are not satisfied, the hyperparameters of EFI can be adjusted to ensure both tests are met for valid inference. These adjustments may include modifying the width of the neck layer, the size of the $w_n^{(1)}$-network, the size of the data-modeling network, as well as tuning the learning rates and iteration numbers used in Algorithm 1.

### A2   Extended Fiducial Density Function of $Z_n$

Let $\mathcal{Z}_n = \{z \in \mathbb{R}^n : U_n(Y_n, X_n, Z_n, w_n^*) = 0\}$ denote the zero-energy set, and let $P_n^*(z|X_n, Y_n, w_n^*)$ denote the cumulative distribution function (CDF) corresponding to $p_n^*(z|X_n, Y_n, w_n^*)$. Under

some regularity conditions on the energy function, Liang et al. (2025) proved that $\mathcal{Z}_n$ is invariant to the choice of $G(\cdot)$. Furthermore, they studied the convergence of $\lim_{\epsilon \downarrow 0} \pi_\epsilon(z|\boldsymbol{X}_n, \boldsymbol{Y}_n, \boldsymbol{w}_n^*)$ in two cases: $\Pi_n(\mathcal{Z}_n) > 0$ and $\Pi_n(\mathcal{Z}_n) = 0$, where $\Pi_n(\cdot)$ denotes the probability measure corresponding to the density function $\pi_0^{\otimes}(z)$ on $\mathbb{R}^n$. Specifically,

(a) $(\Pi_n(\mathcal{Z}_n) > 0)$: In this case, $p_n^*(z|\boldsymbol{X}_n, \boldsymbol{Y}_n, \boldsymbol{w}_n^*)$ is given by

$$\frac{dP_n^*(z|\boldsymbol{X}_n, \boldsymbol{Y}_n, \boldsymbol{w}_n^*)}{dz} = \frac{1}{\Pi_n(\mathcal{Z}_n)} \pi_0^{\otimes n}(z), \quad z \in \mathcal{Z}_n, \tag{A13}$$

which is invariant to the choices of the inverse function $G(\cdot)$ and energy function $U_n(\cdot)$. For example, the logistic regression belongs to this case as shown in Liang et al. (2025).

(b) $(\Pi_n(\mathcal{Z}_n) = 0)$: In this case, $\mathcal{Z}_n$ forms a manifold in $\mathbb{R}^n$ with the highest dimension $p$, and $p_n^*(z|\boldsymbol{Y}_n, \boldsymbol{X}_n, \boldsymbol{w}_n^*)$ concentrates on the highest dimensional manifold and is given by

$$\frac{dP_n^*(z|\boldsymbol{X}_n, \boldsymbol{Y}_n, \boldsymbol{w}_n^*)}{d\nu}(z) = \frac{\pi_0^{\otimes n}(z) \left(\det(\nabla_{\boldsymbol{t}}^2 U_n(z))\right)^{-1/2}}{\int_{\mathcal{Z}_n} \pi_0^{\otimes n}(z) \left(\det(\nabla_{\boldsymbol{t}}^2 U_n(z))\right)^{-1/2} d\nu}, \quad z \in \mathcal{Z}_n, \tag{A14}$$

where $\nu$ is the sum of intrinsic measures on the $p$-dimensional manifold in $\mathcal{Z}_n$, and $\boldsymbol{t} \in \mathbb{R}^{n-p}$ denotes the coefficients of the normalized smooth normal vectors in the tubular neighborhood decomposition (Milnor and Stasheff, 1974) of $z$.

By Theorem 3.2 and Lemma 4.2 in Liang et al. (2025), if the target model is noise-additive and $d(\cdot)$ is specified as in (A4), then $P_n^*(z|\boldsymbol{X}_n, \boldsymbol{Y}_n, \boldsymbol{w}_n^*)$ in (A14) can be reduced to

$$\frac{dP_n^*(z|\boldsymbol{X}_n, \boldsymbol{Y}_n, \boldsymbol{w}_n^*)}{d\nu} = \frac{\pi_0^{\otimes n}(z)}{\int_{\mathcal{Z}_n} \pi_0^{\otimes n}(z) d\nu}. \tag{A15}$$

That is, under the consistency of $\boldsymbol{w}_n^*$, $p_n^*(z|\boldsymbol{X}_n, \boldsymbol{Y}_n, \boldsymbol{w}_n^*)$ is reduced to a truncated density function of $\pi_0^{\otimes n}(z)$ on the manifold $\mathcal{Z}_n$, while $\mathcal{Z}_n$ itself is also invariant to the choice of the inverse function. In other words, for noise-additive models, the EFD of $\boldsymbol{Z}_n$ is asymptotically invariant to the inverse function we learned given its consistency.

## A3    Narrow-Neck $w$-Networks

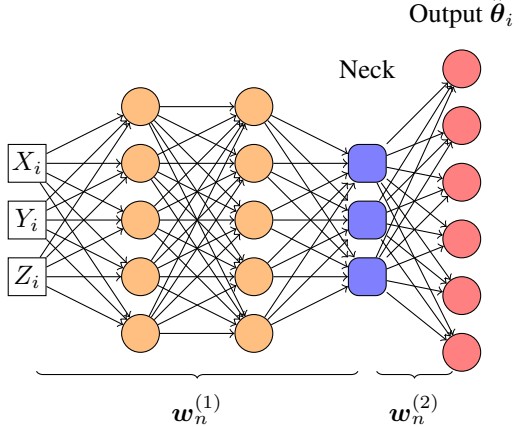

Figure A2: A conceptual structure of narrow neck $w$-networks.

## A4    Proof of Theorem 3.2

### A4.1    Asymptotic Equivalence between StoNets and DNNs

We first provide a brief review of the theory regarding asymptotic equivalence between the StoNet and DNN models, which was originally established in Liang et al. (2022).

Consider a StoNet model:

$$\begin{aligned}
\boldsymbol{Y}_1 &= \boldsymbol{b}_1 + \boldsymbol{w}_1 \boldsymbol{X} + \boldsymbol{e}_1, \\
\boldsymbol{Y}_i &= \boldsymbol{b}_i + \boldsymbol{w}_i \Psi(\boldsymbol{Y}_{i-1}) + \boldsymbol{e}_i, \quad i = 2, 3, \ldots, h, \\
\boldsymbol{Y} &= \boldsymbol{b}_{h+1} + \boldsymbol{w}_{h+1} \Psi(\boldsymbol{Y}_h) + \boldsymbol{e}_{h+1},
\end{aligned} \tag{A16}$$

where $\boldsymbol{X} \in \mathbb{R}^p$ and $\boldsymbol{Y} \in \mathbb{R}^{d_{h+1}}$ represent the input and response variables, respectively; $\boldsymbol{Y}_i \in \mathbb{R}^{d_i}$ are latent variables; $\boldsymbol{e}_i \in \mathbb{R}^{d_i}$ are introduced noise variables; $\boldsymbol{b}_i \in \mathbb{R}^{d_i}$ and $\boldsymbol{w}_i \in \mathbb{R}^{d_i \times d_{i-1}}$ are model parameters, $d_0 = p$, and $\Psi(\boldsymbol{Y}_{i-1}) = (\psi(\boldsymbol{Y}_{i-1,1}), \psi(\boldsymbol{Y}_{i-1,2}), \ldots, \psi(\boldsymbol{Y}_{i-1,d_{i-1}}))^T$ is an element-wise activation function for $i = 1, \ldots, h + 1$. The StoNet defines a latent variable model that reformulates the DNN as a composition of many simple regressions. In this context, we assume that $\boldsymbol{e}_i \sim N(0, \sigma_i^2 I_{d_i})$ for $i = 1, 2, \ldots, h, h + 1$, though other distributions can also be considered for $\boldsymbol{e}_i$'s (Sun and Liang, 2022).

The DNN model corresponding to (A16) is given as follows:

$$\begin{aligned}
\tilde{\boldsymbol{Y}}_1 &= \boldsymbol{b}_1 + \boldsymbol{w}_1 \boldsymbol{X}, \\
\tilde{\boldsymbol{Y}}_i &= \boldsymbol{b}_i + \boldsymbol{w}_i \Psi(\tilde{\boldsymbol{Y}}_{i-1}), \quad i = 2, 3, \ldots, h, \\
\boldsymbol{Y} &= \boldsymbol{b}_{h+1} + \boldsymbol{w}_{h+1} \Psi(\tilde{\boldsymbol{Y}}_h) + \boldsymbol{e}_{h+1}.
\end{aligned} \tag{A17}$$

Let $\boldsymbol{\vartheta} = \{\boldsymbol{b}_1, \boldsymbol{w}_1, \ldots, \boldsymbol{b}_{h+1}, \boldsymbol{w}_{h+1}\}$ denote the parameter set of the DNN model. Let $\pi_{\mathrm{DNN}}(\boldsymbol{Y}|\boldsymbol{X}, \boldsymbol{\vartheta})$ denote the likelihood function of the DNN model (A17), and let $\pi(\boldsymbol{Y}, \boldsymbol{Y}_{\mathrm{mis}}|\boldsymbol{X}, \boldsymbol{\vartheta})$ denote the likelihood function of the StoNet (A16). Let $Q^*(\boldsymbol{\vartheta}) = \mathbb{E}(\log \pi_{\mathrm{DNN}}(\boldsymbol{Y}|\boldsymbol{X}, \boldsymbol{\vartheta}))$, where the expectation is taken with respect to the joint distribution $\pi(\boldsymbol{X}, \boldsymbol{Y})$, Liang et al. (2022) made the following assumption regarding the network structure, activation function, and the variance of the latent variables of StoNet:

**Assumption A4.1.** (i) Parameter space $\tilde{\Theta}$ (of $\boldsymbol{\vartheta}$) is compact; (ii) For any $\boldsymbol{\vartheta} \in \tilde{\Theta}$, $\mathbb{E}(\log \pi(\boldsymbol{Y}, \boldsymbol{Y}_{\mathrm{mis}}|\boldsymbol{X}, \boldsymbol{\vartheta}))^2 < \infty$ ; (iii) The activation function $\psi(\cdot)$ is $c$-Lipschitz continuous; (iv) The network's widths $d_l$'s and depth $h$ are allowed to increase with $n$; (v) The noise introduced in StoNet satisfies the following condition: $\sigma_1 \leq \sigma_2 \leq \cdots \leq \sigma_{h+1}$, and $d_{h+1}(\prod_{i=k+1}^{h} d_i^2) d_k \sigma_k^2 \prec \frac{\sigma_{h+1}^2}{h}$ for any $k \in \{1, 2, \ldots, h\}$.

**Assumption A4.2.** (i) $Q^*(\boldsymbol{\vartheta})$ is continuous in $\boldsymbol{\vartheta}$ and uniquely maximized at $\boldsymbol{\vartheta}^*$; (ii) for any $\epsilon > 0$, $\sup_{\boldsymbol{\vartheta} \in \Theta \backslash B(\epsilon)} Q^*(\boldsymbol{\vartheta})$ exists, where $B(\epsilon) = \{\boldsymbol{\vartheta} : \|\boldsymbol{\vartheta} - \boldsymbol{\vartheta}^*\| < \epsilon\}$, and $\delta = Q^*(\boldsymbol{\vartheta}^*) - \sup_{\boldsymbol{\vartheta} \in \Theta \backslash B(\epsilon)} Q^*(\boldsymbol{\vartheta}) > 0$.

Under Assumptions A4.1 and A4.2, Liang et al. (2022) proved the following lemma.

**Lemma A4.3.** *(Liang et al., 2022) Suppose that Assumptions A4.1-A4.2 hold, and $\pi(\boldsymbol{Y}, \boldsymbol{Y}_{\mathrm{mis}}|\boldsymbol{X}, \boldsymbol{\vartheta})$ is continuous in $\boldsymbol{\vartheta}$. Then*

$$(i) \quad \sup_{\boldsymbol{\vartheta} \in \Theta} \left| \frac{1}{n} \sum_{i=1}^{n} \log \pi(\boldsymbol{Y}^{(i)}, \boldsymbol{Y}_{mis}^{(i)}|\boldsymbol{X}^{(i)}, \boldsymbol{\vartheta}) - \frac{1}{n} \sum_{i=1}^{n} \log \pi_{\mathrm{DNN}}(\boldsymbol{Y}^{(i)}|\boldsymbol{X}^{(i)}, \boldsymbol{\vartheta}) \right| \xrightarrow{p} 0, \tag{A18}$$

$$(ii) \quad \|\hat{\boldsymbol{\vartheta}}_n - \boldsymbol{\vartheta}^*\| \xrightarrow{P} 0, \quad as \ n \to \infty,$$

*where $\boldsymbol{\vartheta}^* = \arg\max_{\boldsymbol{\vartheta} \in \tilde{\Theta}} \mathbb{E}(\log \pi_{\mathrm{DNN}}(\boldsymbol{Y}|\boldsymbol{X}, \boldsymbol{\vartheta}))$ denotes the true parameters of the DNN model as specified in (A17), and $\hat{\boldsymbol{\vartheta}}_n = \arg\max_{\boldsymbol{\vartheta} \in \tilde{\Theta}} \{\frac{1}{n} \sum_{i=1}^{n} \log \pi(\boldsymbol{Y}^{(i)}, \boldsymbol{Y}_{mis}^{(i)}|\boldsymbol{X}^{(i)}, \boldsymbol{\vartheta})\}$ denotes the maximum likelihood estimator of the StoNet model (A16) with the pseudo-complete data.*

Lemma A4.3 implies that the StoNet and DNN are asymptotically equivalent as the training sample size $n$ becomes large, and it forms the basis for the bridging the StoNet and the DNN. The asymptotic equivalence can be elaborated from two perspectives. First, suppose the DNN model (A17) is true. Lemma A4.3 implies that when $n$ becomes large, the weights of the DNN can be learned by training a StoNet of the same structure with $\sigma_i^2$'s satisfying Assumption A4.1-(v). On the other hand, suppose that the StoNet (A16) is true, and then Lemma A4.3 implies that *for any StoNet satisfying Assumptions A4.1 & A4.2, the weights $\boldsymbol{\vartheta}$ can be learned by training a DNN of the same structure when the training sample size is large.*

## A4.2 Justification for Condition (12)-(ii)

To justify this condition, we first introduce the following lemma:

**Lemma A4.4.** *Consider a random matrix $\mathbb{M} \in \mathbb{R}^{n \times d}$ with $n \geq d$. Suppose that the eigenvalues of $\mathbb{M}^T \mathbb{M}$ are upper bounded, i.e., $\lambda_{\max}(\mathbb{M}^T \mathbb{M}) \leq \kappa_{\max}$ for some constant $\kappa_{\max} > 0$. Let $\Psi(\mathbb{M})$ denote an elementwise transformation of $\mathbb{M}$. Then $\lambda_{\max}\left((\Psi(\mathbb{M}))^T (\Psi(\mathbb{M}))\right) \leq \kappa_{\max}$ for the tanh, sigmoid and ReLU transformations.*

*Proof.* For ReLU, the result follows from Lemma 5 of Dittmer et al. (2018). For *tanh* and *sigmoid*, since they are Lipschitz continuous with a Lipschitz constant of 1, Lemma 5 of Dittmer et al. (2018) also applies. $\qquad\square$

Since the connection weights take values in a compact space $\tilde{\boldsymbol{\Theta}}$, there exists a constant $0 < \tau_{\max} < \infty$ such that

$$\lambda_{\max}(\boldsymbol{w}_l \boldsymbol{w}_l^T) \leq \tau_{\max},$$

for any $l = 1, 2, \ldots, h+1$, where $\boldsymbol{w}_l \in \mathbb{R}^{d_l \times d_{l-1}}$ is the weight matrix of the DNN at layer $l$.

Let $\mathbb{M}_l \in \mathbb{R}^{n \times d_l}$ denote the output of hidden layer $l \in \{1, 2, \ldots, h\}$ of the StoNet; that is,

$$
\begin{aligned}
\mathbb{M}_l &= \Psi(\Psi(\mathbb{M}_{l-1}) \boldsymbol{w}_l^T), \quad l = 1, 2, \ldots, h-1, \\
\mathbb{M}_h &= \Psi(\Psi(\mathbb{M}_{h-1}) \boldsymbol{w}_h^T + \mathbb{V}).
\end{aligned}
\tag{A19}
$$

Consider the case that the activation functions are bounded, such as *sigmoid* and *tanh*. Then the matrix $\mathbb{E}(\mathbb{M}_h^T \mathbb{M}_h)$ has the eigenvalues upper bounded by $n\kappa_{\max}$ for some constant $0 < \kappa_{\max} < \infty$.

For the case that the activation functions are unbounded, such as *ReLU* or *leaky ReLu*, we can employ the layer-normalization method in training. In this case, by Lemma A4.4, the matrix $\Psi(\mathbb{M}_{h-1})^T \Psi(\mathbb{M}_{h-1})$ has the eigenvalues upper bounded by $n\kappa_{\max}$, provided that the eigenvalues of the matrix $\mathbb{M}_{h-1}^T \mathbb{M}_{h-1}$ is bounded by $n\kappa_{\max}$ after layer-normalization.

Let $\tilde{\mathbb{M}}_h = \Psi(\mathbb{M}_{h-1}) \boldsymbol{w}_h^T$. Then, by an extension of Ostrowski's theorem, see Theorem 3.2 of Higham and Cheng (1998), the eigenvalues of the matrix $\tilde{\mathbb{M}}_h^T \tilde{\mathbb{M}}_h = \boldsymbol{w}_h (\Psi(\mathbb{M}_{h-1}))^T \Psi(\mathbb{M}_{h-1}) \boldsymbol{w}_h^T$ is bounded by

$$\lambda_{\max}(\tilde{\mathbb{M}}_h^T \tilde{\mathbb{M}}_h) \leq n\kappa_{\max}\tau_{\max}.$$

Also, we have $\frac{1}{n}\lambda_{\max}(\mathbb{E}(\mathbb{V}^T \mathbb{V})) = \sigma_v^2$ under the assumption (*). Then, with the use of Lemma A4.4 and the Cauchy-Schwarz inequality, we have

$$\frac{1}{n}\lambda_{\max}(\mathbb{E}(\mathbb{M}_h^T \mathbb{M}_h)) \leq \kappa_{\max}\tau_{\max} + \sigma_v^2 + 2\sigma_v\sqrt{\kappa_{\max}\tau_{\max}},$$

as the sample size $n$ becomes large. This concludes the proof for condition (12)-(ii).

## A4.3 Proof of Theorem 3.2

*Proof.* First, we note that $\sigma_{\hat{\theta}}^2$, the variance of each component of $\hat{\boldsymbol{\theta}}_i (\in \mathbb{R}^p)$, is of the order $O(\epsilon/\eta)$ under the setting of the energy function (A6). By setting $\sigma_u^2 \prec \frac{\epsilon}{\eta h d_h p}$, it is easy to verify that the $\boldsymbol{w}$-network satisfies Assumption A4.1. Therefore, under the additional regularity conditions given in Assumption A4.2, the proposed stochastic $\boldsymbol{w}$-network is asymptotically equivalent to the original $\boldsymbol{w}$-network. Furthermore, the stochastic $\boldsymbol{w}$-network can be trained by training the original $\boldsymbol{w}$-network using Algorithm 1.

Next, we prove that the inverse mapping learned through training the stochastic $\boldsymbol{w}$-network is consistent. For Lemma 3.1, Liang et al. (2025) first established the convergence of the weights to $\boldsymbol{w}_n^*$, and subsequently established the weak convergence of the latent variables. Therefore, due to the Markov chain nature of Algorithm 1, it suffices to prove that the inverse mapping produced by the StoNet is consistent, assuming the latent variables have been correctly imputed (i.e., the true values of $\boldsymbol{Z}_n$ are known). Once the inverse mapping's consistency is established, the latent variables will be correctly imputed in the next iteration, owing to the algorithm's equation-solving nature, which is ensured by setting $\epsilon \to 0$ upon convergence to the zero-energy region. This ensures that algorithm remains in its equilibrium, enabling accurate inference of model uncertainty.

Under the StoNet setting, estimation of $\boldsymbol{w}_n^{(2)}$ involves solving $p$ low-dimensional regressions. In particular, given $\mathbb{M} = (\boldsymbol{m}_1, \boldsymbol{m}_2, \ldots, \boldsymbol{m}_n)^T \in \mathbb{R}^{n \times d_l}$, solving each of the regressions contributes a parameter estimation error:

$$\mathbb{E}\|\hat{\boldsymbol{\xi}}_j^m - \boldsymbol{\xi}_j^*\|^2 = \frac{\sigma_{\hat{\theta}}^2}{n}\text{Tr}([\mathbb{E}(\boldsymbol{m}_i \boldsymbol{m}_i^T)]^{-1}) \leq \frac{d_h \sigma_{\hat{\theta}}^2}{n \rho_{\min}} = O(\frac{d_h \epsilon}{n \eta}), \tag{A20}$$

where $\boldsymbol{\xi}_i^*$ denotes the $i$th column of $\boldsymbol{w}_n^{*(2)}$, i.e., $\boldsymbol{w}_n^{*(2)} := (\boldsymbol{\xi}_1^*, \ldots, \boldsymbol{\xi}_p^*)^T \in \mathbb{R}^{p \times d_h}$; $\hat{\boldsymbol{\xi}}_j^m = (\mathbb{M}^T\mathbb{M})^{-1}\mathbb{M}^T\hat{\boldsymbol{\theta}}^{(j)}$ is the OLS estimator for the regression coefficients; and $\hat{\boldsymbol{\theta}}^{(j)} = (\hat{\theta}_1^{(j)}, \ldots, \hat{\theta}_n^{(j)})^T$.

Let $\hat{\boldsymbol{w}}^{m(2)} = (\hat{\boldsymbol{\xi}}_1^m, \ldots, \hat{\boldsymbol{\xi}}_p^m)^T \in \mathbb{R}^{p \times d_h}$. By a fundamental property of linear regression, the mean prediction $\hat{\boldsymbol{\theta}}_i^* = \hat{\boldsymbol{w}}^{m(2)}\boldsymbol{m}_i$ is consistent with respect to $\boldsymbol{\theta}^* = \mathbb{E}(\hat{\boldsymbol{\theta}}_i)$, provided that $d_h \prec n$, the $\boldsymbol{m}_i$'s extract all $\hat{\boldsymbol{\theta}}$-relevant information from the input variables, and the $\boldsymbol{w}$-network has sufficient capacity to establish the linear relationship $\hat{\boldsymbol{\theta}}_i \sim \boldsymbol{m}_i$ for $i = 1, 2, \ldots, n$. The equality $\boldsymbol{\theta}^* = \mathbb{E}(\hat{\boldsymbol{\theta}}_i)$ is guaranteed by the setting of $\epsilon \to 0$ and by the construction of the energy function, which approaches to zero if and only if the empirical mean $\frac{1}{n}\sum_{i=1}^n \hat{\boldsymbol{\theta}}_i$ converges to $\boldsymbol{\theta}^*$ and the variance of $\hat{\boldsymbol{\theta}}_i$ approaches to zero. Furthermore, as shown in Liang et al. (2022), the stochastic layer effectively provides a sufficient dimension reduction for the input variables.

Finally, we prove that the inverse mapping obtained by Algorithm 1 during the training of the non-stochastic $\boldsymbol{w}$-network is also consistent. Let $\tilde{\boldsymbol{\theta}}_i^* = \boldsymbol{w}^{*(2)}\Psi(\boldsymbol{\mu}_i)$. Let $\tilde{\theta}_i^{*(j)}$ and $\hat{\theta}_i^{*(j)}$ denote the $j$th element of $\tilde{\boldsymbol{\theta}}_i^*$ and $\hat{\boldsymbol{\theta}}_i^*$, respectively. From equation (*), we have

$$\begin{aligned}
\mathbb{E}|\hat{\theta}_i^{*(j)} - \tilde{\theta}_i^{*(j)}| &= \mathbb{E}|(\hat{\boldsymbol{\xi}}_j^m)^T\boldsymbol{m}_i - (\boldsymbol{\xi}_j^*)^T\Psi(\boldsymbol{\mu}_i)| \\
&\leq \mathbb{E}|(\hat{\boldsymbol{\xi}}_j^m - \boldsymbol{\xi}_j^*)^T\boldsymbol{m}_i| + \mathbb{E}|(\boldsymbol{\xi}_j^*)^T(\boldsymbol{m}_i - \Psi(\boldsymbol{\mu}_i))| \\
&\leq (\mathbb{E}\|\hat{\boldsymbol{\xi}}_j^m - \boldsymbol{\xi}_j^*\|^2)^{1/2}(\mathbb{E}\|\boldsymbol{m}_i\|^2)^{1/2} + (\mathbb{E}\|\boldsymbol{\xi}_j^*\|^2)^{1/2}(\mathbb{E}\|\boldsymbol{m}_i - \Psi(\boldsymbol{\mu}_i)\|^2)^{1/2} \\
&\leq (\mathbb{E}\|\hat{\boldsymbol{\xi}}_j^m - \boldsymbol{\xi}_j^*\|^2)^{1/2}(\text{Tr}(\mathbb{E}(\boldsymbol{m}_i \boldsymbol{m}_i^T))^{1/2} + c(\mathbb{E}\|\boldsymbol{\xi}_j^*\|^2)^{1/2}(\text{Tr}(\sigma_v^2 I_{d_h}))^{1/2} \\
&\lesssim \sqrt{\frac{d_h \epsilon}{n \eta}}\sqrt{d_h \rho_{\max}} + c d_{\tilde{\Theta}} d_h \sigma_v,
\end{aligned} \tag{A21}$$

where $d_{\tilde{\Theta}}$ denotes the radius of the parameter space $\tilde{\Theta}$ (centered at 0), the second inequality follows from Cauchy-Schwarz inequality, the third inequality follows from the Taylor expansion for $\Psi(\boldsymbol{\mu}_i + \boldsymbol{v}_i)$ (at the point $\boldsymbol{\mu}_i$), and the last inequality follows from (A20), condition (12)-(ii), and the boundedness of $\boldsymbol{\xi}_j^*$ as stated in Assumption A4.1-(i).

Substituting $\sigma_v \prec \sqrt{\frac{\epsilon}{\eta h d_h p}}$ and ignoring some constant factors in (A21), we obtain

$$\mathbb{E}\|\hat{\boldsymbol{\theta}}_i^* - \tilde{\boldsymbol{\theta}}_i^*\|_1 \leq p\mathbb{E}|\hat{\theta}_i^{*(j)} - \tilde{\theta}_i^{*(j)}| \lesssim p d_h\sqrt{\frac{\epsilon}{n}} + \sqrt{\frac{\epsilon p d_h}{h}},$$

where $\|\cdot\|_1$ denotes the $l_1$-norm of a vector.

By setting $\epsilon \prec \min\{\frac{n}{p^2 d_h^2}, \frac{h}{p d_h}\}$, we have $\mathbb{E}\|\hat{\boldsymbol{\theta}}_i^* - \tilde{\boldsymbol{\theta}}_i^*\|_1 = o(1)$, which implies $\tilde{\boldsymbol{\theta}}_i^*$ is also consistent with respect to $\boldsymbol{\theta}^* = \mathbb{E}(\hat{\boldsymbol{\theta}}_i)$. Consequently, the inverse mapping $\frac{1}{n}\sum_{i=1}^n \tilde{\boldsymbol{\theta}}_i^*$ produced by Algorithm 1 in training the non-stochastic $\boldsymbol{w}$-network is also consistent with respect to $\boldsymbol{\theta}^*$. This concludes the proof of the theorem. $\qquad\square$

## A5 Additional Numerical Results

Regarding uncertainty quantification, we note that there are two types of uncertainties:

1. **Aleatoric uncertainty:** This refers to the *irreducible noise* inherent in the data-generating process. It can be modeled as

$$y_i = f(x_i \mid \theta) + \epsilon_i,$$

where $\epsilon_i \sim \mathcal{N}(0, \sigma^2)$. Estimating the unknown variance $\sigma^2$ corresponds to quantifying the aleatoric uncertainty (system random error). This is precisely what we addressed in the last experiment included in our previous rebuttal, where $\sigma^2 = 0.1$ was treated as unknown.

2. **Epistemic uncertainty:** This refers to the *reducible estimation error* due to limited data or incomplete knowledge of the true model (see, for example, model comparison in Section A5.8). In classical statistics, confidence intervals quantify epistemic uncertainty: as the dataset size increases, epistemic uncertainty—and thus the width of the confidence interval—decreases.

In the following section, we demonstrate through different examples that EFI is able to accurately quantity both types of uncertainties. We use the coverage rate as the key metric to quantify epistemic uncertainty. The coverage rate can reach the nominal level only when the parameter estimates are unbiased and the uncertainty estimation is accurate.

### A5.1 1-D Poisson Equation

Figure A3 and Figure A4 provide typical trajectories learned for the 1-D Poisson model (13) using the methods: PINN, Dropout, and Bayesian PINN. For the ablation study, we vary the noise standard deviation from $0.01$ to $0.1$ to further assess the validity and accuracy of the proposed method in uncertainty quantification. The results, presented in Tables A1, A2, and A3, show that under different noise levels, the EFI algorithm consistently achieves a 95% coverage rate for the 95% confidence intervals.

To further investigate the relationship between confidence intervals and epistemic uncertainty, we conducted an additional experiment using two different sample sizes: 20 and 80. The results, presented in Table A4, clearly show that as the sample size increases, the width of the confidence interval decreases while the coverage rate remains consistent. This demonstrates that the confidence interval effectively captures the reducible nature of epistemic uncertainty.

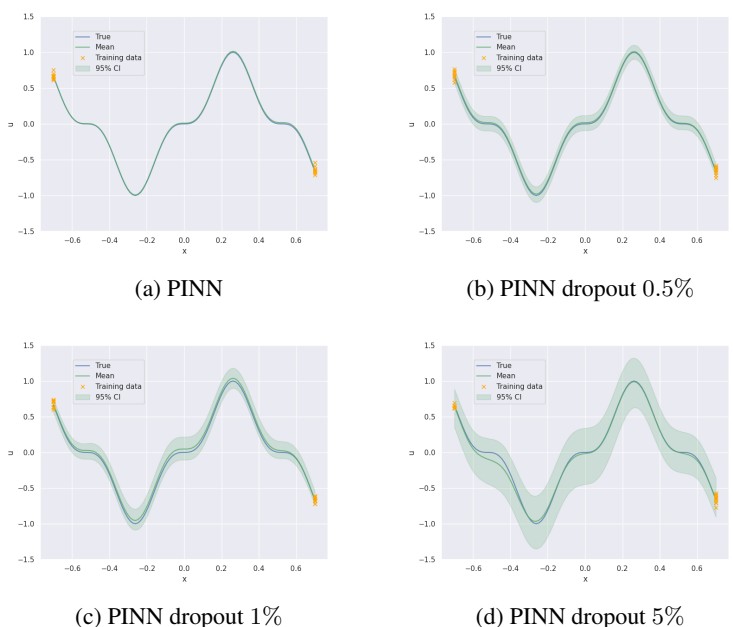

(a) PINN

(b) PINN dropout $0.5\%$

(c) PINN dropout $1\%$

(d) PINN dropout $5\%$

Figure A3: 1-D Poisson model (13): (a) Trajectory learned by PINN (without uncertainties); (b)-(d) Trajectories learned by Dropout with different dropout rates.

### A5.2 1-D Poisson Equation with $f$-measurement error

We revisit the same 1-D Poisson model as defined in (13), but now incorporate measurement errors in both $u$ and $f$. Specifically, we consider $4$ sensors for $u$ and $40$ sensors for $f$, with each sensor

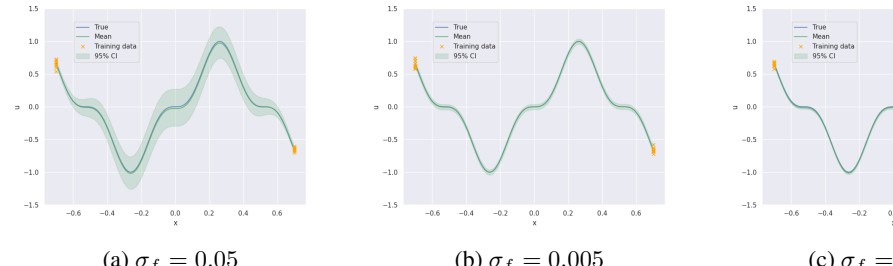

| (a) $\sigma_f = 0.05$ | (b) $\sigma_f = 0.005$ | (c) $\sigma_f = 0.0005$ |

Figure A4: 1-D Poisson model (13): Typical trajectories learned by Bayesian PINN with different $\sigma_f$ values.

Table A1: Comparison of different methods for 1D-Poisson with $\sigma = 0.01$

| Method | MSE | Coverage Rate | CI-Width |
| --- | --- | --- | --- |
| PINN (no dropout) | 0.000008 (0.000001) | 0.2808 (0.028721) | 0.002079 (0.000026) |
| Dropout (0.5%) | 0.000160 (0.000037) | 1.0000 (0.000000) | 0.199798 (0.004135) |
| Dropout (1%) | 0.000676 (0.000082) | 1.0000 (0.000000) | 0.276534 (0.006642) |
| Dropout (5%) | 0.009080 (0.001516) | 0.9639 (0.011218) | 0.593541 (0.025812) |
| Bayesian ($\sigma_f = 0.05$) | 0.000181 (0.000025) | 0.9984 (0.000400) | 0.254861 (0.001554) |
| Bayesian ($\sigma_f = 0.005$) | 0.000007 (0.000001) | 0.9915 (0.003220) | 0.028386 (0.000201) |
| Bayesian ($\sigma_f = 0.0005$) | 0.000007 (0.000001) | 0.9544 (0.013751) | 0.010437 (0.000046) |
| **EFI** | **0.000007 (0.000001)** | **0.9423 (0.015229)** | **0.010759 (0.000117)** |

recording 10 replicate measurements. Measurement errors are modeled as $z_i^u \sim N(0, 0.05^2)$ and $z_i^f \sim N(0, 0.05^2)$, and we simulate 100 independent datasets under this setting. The experimental results are summarized in Table A5.

For the Bayesian PINN (B-PINN) method (Yang et al., 2021), we set $\sigma_u = \sigma_f = 0.05$, ensuring the likelihood function is correctly specified. However, as shown in Table A5, B-PINN produces excessively wide confidence intervals, resulting in an inflated and inaccurate coverage rate. This highlights another issue inherent to Bayesian DNNs, as noted in Liang et al. (2018b); Sun et al. (2022): their performance can be significantly affected by the choice of prior in small-$n$-large-$p$ settings. Here, $p$ refers to the number of parameters in the DNN used to approximate the solution $u(\boldsymbol{x})$. Similarly, the dropout method continues to produce overly wide confidence intervals and inflated coverage rates.

In contrast, EFI achieves a coverage rate of 94.88% with the smallest confidence interval width. Notably, this experiment involves noise in both $u$ and $f$ observations. To evaluate the imputed errors, the QQ-plot of $\hat{z}_i^u$ and $\hat{z}_i^f$ across 100 experiments is shown in Figure A5. The Q-Q plot confirms that the distribution of imputed errors closely follows its theoretical distribution, supporting the validity of the EFI approach in handling measurement noise from different sources.

Table A2: Comparison of different methods for 1D-Poisson with $\sigma = 0.025$

| Method | MSE | Coverage Rate | CI-Width |
| --- | --- | --- | --- |
| PINN (no dropout) | 0.000046 (0.000006) | 0.1529 (0.019175) | 0.002110 (0.000025) |
| Dropout (0.5%) | 0.000157 (0.000028) | 1.0000 (0.000000) | 0.195658 (0.003892) |
| Dropout (1%) | 0.000666 (0.000084) | 1.0000 (0.000000) | 0.267936 (0.005751) |
| Dropout (5%) | 0.004573 (0.000654) | 0.9979 (0.001157) | 0.643227 (0.031514) |
| Bayesian ($\sigma_f = 0.05$) | 0.000161 (0.000019) | 0.9992 (0.000307) | 0.257184 (0.001612) |
| Bayesian ($\sigma_f = 0.005$) | 0.000060 (0.000013) | 0.9630 (0.012690) | 0.037341 (0.000418) |
| Bayesian ($\sigma_f = 0.0005$) | 0.000090 (0.000014) | 0.8504 (0.023270) | 0.024676 (0.000366) |
| **EFI** | **0.000048 (0.000006)** | **0.9577 (0.014487)** | **0.027845 (0.000115)** |

Table A3: Comparison of different methods for 1D-Poisson with $\sigma = 0.1$

| Method | MSE | Coverage Rate | CI-Width |
|---|---|---|---|
| PINN (no dropout) | 0.000581 (0.000066) | 0.0320 (0.006518) | 0.002088 (0.000024) |
| Dropout (0.5%) | 0.000935 (0.000106) | 0.9905 (0.002706) | 0.196226 (0.003748) |
| Dropout (1%) | 0.001413 (0.000175) | 0.9976 (0.001929) | 0.274188 (0.007653) |
| Dropout (5%) | 0.006840 (0.001027) | 0.9955 (0.002439) | 0.658607 (0.031770) |
| Bayesian ($\sigma_f = 0.05$) | 0.000966 (0.000106) | 0.9936 (0.002068) | 0.285660 (0.001405) |
| Bayesian ($\sigma_f = 0.005$) | 0.000674 (0.000080) | 0.9582 (0.015014) | 0.104743 (0.000325) |
| Bayesian ($\sigma_f = 0.0005$) | 0.004944 (0.000788) | 0.3827 (0.030245) | 0.066296 (0.002515) |
| **EFI** | **0.000624 (0.000058)** | **0.9493 (0.013526)** | **0.099543 (0.000775)** |

Table A4: Numerical results for 1D-Poisson with $\sigma = 0.1$, where the confidence interval width shrinks as sample size increases. The results are computed based on 100 independent simulations.

| Method | $n_b$ (sample size) | MSE | Coverage Rate | CI-Width |
|---|---|---|---|---|
| EFI | 20 | 0.000624 (0.000058) | 0.9493 (0.013526) | 0.099543 (0.000775) |
| EFI | 80 | 0.000173 (0.000017) | 0.9501 (0.013459) | 0.054256 (0.000723) |

Table A5: Comparison of different methods for the 1-D Poisson model (13) (with $f$-measurement error), averaged over 100 runs.

| Method | hidden layers | MSE | Coverage Rate | CI-Width |
|---|---|---|---|---|
| PINN | [50, 50] | 0.000271 (0.000019) | 0.0921 (0.008076) | 0.003625 (0.000085) |
| Dropout (0.5%) | [50, 50] | 0.000310 (0.000022) | 1.0000 (0.000000) | 0.240585 (0.002282) |
| Dropout (1.0%) | [50, 50] | 0.000530 (0.000046) | 1.0000 (0.000000) | 0.357782 (0.005141) |
| Dropout (5.0%) | [50, 50] | 0.003527 (0.000193) | 1.0000 (0.000000) | 0.669799 (0.006158) |
| Bayesian | [50, 50] | 0.000233 (0.000017) | 0.9960 (0.002035) | 0.086291 (0.000068) |
| **EFI** | **[50, 50]** | **0.000238 (0.000017)** | **0.9488 (0.009419)** | **0.060269 (0.000365)** |

### A5.3 Computating Time

Table A6 reports the wall-clock time for the Poisson-1D experiment with different algorithms. As shown by the table, EFI is slower than PINN (with dropout) but much faster than B-PINN (with the HMC sampler). Importantly, we have demonstrated that EFI is the only existing method capable of correctly constructing confidence intervals in a statistically rigorous manner. For larger networks, we can adopt transfer learning techniques by constructing EFI hyper-networks only for the last few layers of the $\boldsymbol{\theta}$-network, which would significantly reduce the computational cost.

### A5.4 Non-linear Poisson Equation

We extend our study to a non-linear Poisson equation given by:

$$\beta \frac{\partial^2 u}{\partial x^2} + k \tanh(u) = f, \quad x \in \Omega, \tag{A22}$$

where $\Omega = [-0.7, 0.7]$, $\beta = 0.01$, $k = 0.7$, $u = \sin^3(6x)$, and $f$ can be derived from (A22). For this scenario, we use 4 sensors located at $x \in \{-0.7, -0.47, 0.47, 0.7\}$ to provide noisy observations of the solution $u$. Additionally, we employ 40 sensors, equally spaced within $[-0.7, 0.7]$, to measure $f$. Both $u$ and $f$ measurements are assumed to contain noise. In the simulation, measurement errors are modeled as $z_i^u \sim N(0, 0.05^2)$ for $i = 1, 2, \ldots, 40$, with each solution sensor providing 10 replicate measurements, and $z_i^f \sim N(0, 0.05^2)$ for $i = 1, 2, \ldots, 400$.

The experimental results are summarized in Table A7. The findings exhibit a similar pattern to those in Table A5: The Bayesian and dropout methods yield inflated coverage rates and overly wide confidence intervals, whereas the EFI method achieves an accurate coverage rate and the narrowest confidence interval. For a fair comparison, we exclude cases where B-PINN converged to incorrect

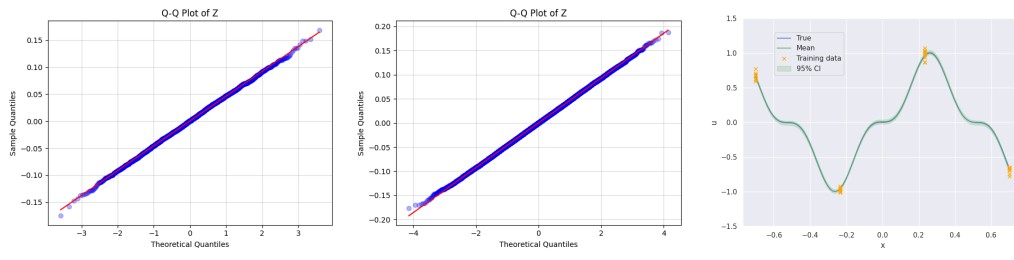

(a) QQ-plot of imputed errors in $u$  (b) QQ-plot of imputed errors in $f$  (c) Confidence interval

Figure A5: EFI-PINN diagnostic for the 1-D Poisson model (13) (with $f$-measurement error)

Table A6: Wall-clock time for PINN, B-PINN and EFI-PINN

| Algorithm | Hypernetwork | Epoch ($\times 10^3$) | Wall-clock time (s) | Time per epoch (ms) |
| --- | --- | --- | --- | --- |
| PINN (dropout) | - | 200 | 133 | 0.665 |
| B-PINN | - | 100 | 1330 | 13.300 |
| EFI | [16,16,16] | 200 | 391 | 1.955 |
| EFI | [16,16,4] | 200 | 385 | 1.925 |

solutions, as these represent instances of failure in the optimization process, see Figure A6 for an instance.

Table A7: Comparison of different methods for the nonlinear 1-D Poisson model (A22) (with $f$-measurement error), averaged over 100 runs.

| Method | hidden layers | MSE | Coverage Rate | CI-Width |
| --- | --- | --- | --- | --- |
| PINN | [50, 50] | 0.000507 (0.000044) | 0.0947 (0.007564) | 0.005154 (0.000565) |
| Dropout (0.5%) | [50, 50] | 0.001050 (0.000123) | 0.9962 (0.002068) | 0.246367 (0.002182) |
| Dropout (1%) | [50, 50] | 0.002807 (0.000309) | 0.9861 (0.003829) | 0.358088 (0.005178) |
| Dropout (5%) | [50, 50] | 0.007010 (0.000394) | 0.9956 (0.001566) | 0.543565 (0.003096) |
| Bayesian (unstable removed) | [50, 50] | 0.000376 (0.000033) | 0.9938 (0.002618) | 0.104673 (0.000394) |
| **EFI** | **[50, 50]** | **0.000385 (0.000039)** | **0.9483 (0.009191)** | **0.099880 (0.002853)** |

### A5.5 Non-linear Poisson Inverse Problem

In this section, we consider the same non-linear Poisson equation as in (A22), but with $k = 0.7$ treated as an unknown parameter to be estimated. For this setup, we utilize 8 sensors, evenly distributed across $\Omega = [-0.7, 0.7]$ to measure $u$, with measurement noise modeled as $z_i^u \sim N(0, 0.05^2)$. Additionally, 200 sensors are employed to measure $f$, and these measurements are assumed to be noise-free.

To apply EFI framework to inverse problem, we extend the output of the $w$-network by adding an additional dimension dedicated to estimating $k$, as depicted in Figure A7. This modification enables the EFI framework to simultaneously estimate the solution $u$ and the parameter $k$, along with their respective uncertainties. The results are presented in Table A8, demonstrating the capability of EFI to provide accurate uncertainty quantification for both $u$ and $k$. In contrast, B-PINN consistently produces excessively large confidence intervals for both the solution u and the parameter k. Notably, the confidence interval for $k$ estimated by B-PINN is approximately twice as wide as that produced by EFI, indicating a significant overestimation of uncertainty. This highlights the superior precision and robustness of the EFI framework in inverse problems.

For this problem, we also tested EFI with a larger data modeling network, consisting of two hidden layers and each hidden layer consisting of 100 hidden units. As expected, EFI produced results similar to those obtained with a much smaller data modeling network. As noted earlier, EFI can accurately quantify model uncertainty as long as the random errors are correctly imputed and the

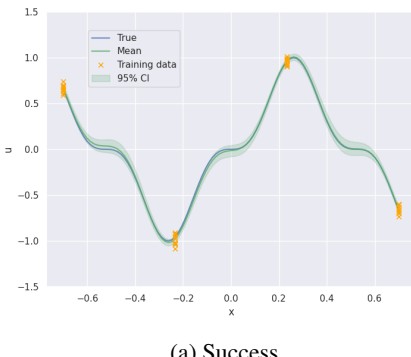

(a) Success

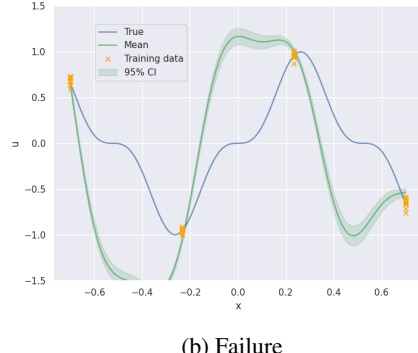

(b) Failure

Figure A6: Successful and failed optimization results of Bayesian PINN for the nonlinear 1-D Poisson model (A22) (with $f$-measurement error)

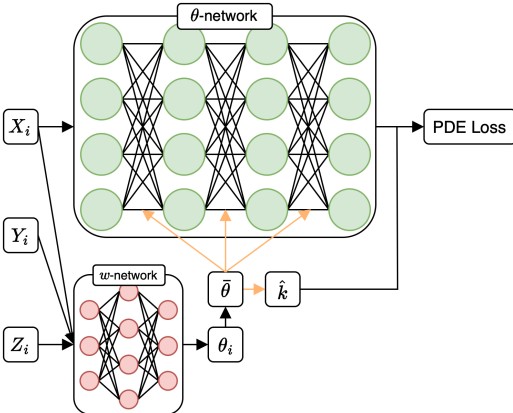

Figure A7: Diagram of EFI for inverse problems, where the orange links indicates the contribution of $\bar{\theta}$ to both $\theta$-network and $k$ estimation.

inverse function is consistently estimated. This capability is independent of the specific configurations of the $w$-network and the data modeling network, highlighting EFI's flexibility and robustness.

Table A8: Comparison of different methods for the nonlinear 1-D Poisson model (A22) with parameter estimation, averaged over 100 runs.

| Method | hidden layers | MSE($\times 10^{-4}$) | Coverage Rate | CI-Width | k-Mean | k-Coverage Rate | k-CI-Width |
|---|---|---|---|---|---|---|---|
| PINN | [50, 50] | 1.79 (0.13) | 0.1464 (0.0097) | 0.0045 (0.0001) | 0.6998 (0.0006) | 0.00 (0.0000) | 7.9e-5 (5e-6) |
| Dropout (0.5%) | [50, 50] | 1.84 (0.10) | 1.0000 (0.0000) | 0.2393 (0.0035) | 0.6912 (0.0007) | 0.09 (0.0288) | 0.0045 (0.0002) |
| Dropout (1.0%) | [50, 50] | 2.72 (0.24) | 1.0000 (0.0000) | 0.3032 (0.0071) | 0.6874 (0.0008) | 0.11 (0.0314) | 0.0091 (0.0007) |
| Dropout (5.0%) | [50, 50] | 27.08 (2.12) | 1.0000 (0.0000) | 0.5363 (0.0131) | 0.6493 (0.0023) | 0.06 (0.0239) | 0.0252 (0.0014) |
| Bayesian | [50, 50] | 1.31 (0.08) | 0.9752 (0.0055) | 0.0517 (3e-5) | 0.6994 (0.0005) | 1.00 (0.0000) | 0.0411 (0.0002) |
| **EFI** | **[50, 50]** | **1.03 (0.08)** | **0.9473 (0.0099)** | **0.0396 (0.0002)** | **0.6985 (0.0004)** | **0.94 (0.0239)** | **0.0179 (0.0002)** |
| **EFI** | **[100, 100]** | **0.98 (0.07)** | **0.9560 (0.0099)** | **0.0395 (0.0002)** | **0.6995 (0.0004)** | **0.96 (0.0197)** | **0.0168 (0.0002)** |

### A5.6 Poisson equation with unknown noise standard deviation

We now consider the case where the noise standard deviation is treated as an unknown parameter. In this setting, the variability in the observations due to the inherent randomness of the data-generating process—often referred to as systematic error—can be interpreted as aleatoric uncertainty.

As shown in Table A9, the EFI algorithm successfully recovers accurate estimates for both the solution $u$ and the noise standard deviation $\sigma$, accompanied by well-calibrated confidence intervals.

Table A9: Numerical results for 1D-Poisson with unknown $\sigma = 0.1$ and sample size $n_b = 60$, where the number in the parentheses represents the standard error of the estimator.

| Method | MSE | Coverage Rate | CI-Width | $\sigma$-mean | $\sigma$-CR | $\sigma$-CI-Width |
|---|---|---|---|---|---|---|
| EFI | 0.000220 (0.000025) | 0.9559 (0.014349) | 0.061347 (0.000848) | 0.097269 (0.001158) | 0.9400 (0.023868) | 0.056768 (0.001284) |

## A5.7 Black-Scholes Model

As a practical example, we consider the classical option pricing model in finance — the Black-Scholes model (Black and Scholes, 1973):

$$\frac{\partial V}{\partial t} + \frac{1}{2}\sigma^2 S^2 \frac{\partial^2 V}{\partial S^2} + rS\frac{\partial}{\partial S} - rV = 0,$$
$$C(0, t) = 0 \text{ for all } t,$$
$$C(S, t) \to S - K \text{ as } S \to \infty, \quad (A23)$$
$$C(S, T) = \max\{S - K, 0\},$$

which describes the price $V(S, t)$ of an option. Here, $S$ is the price of the underlying asset (e.g., a stock), $t$ is time, $\sigma$ represents the volatility of the asset, $r$ is the risk-free interest rate, $K$ is the strike price, and $T$ is the expiration time of the option. The boundary conditions reflect specific financial constraints.

This model has been widely used to calculate the price of European call and put options. Specifically, the analytic solution for the call option price $C(S_t, t)$ is given by

$$C(S_t, t) = \Phi(d_+)S_t - \Phi(d_-)Ke^{-r(T-t)},$$
$$d_+ = \frac{1}{\sigma\sqrt{T-t}}\left[\ln\left(\frac{S_t}{K}\right) + \left(r + \frac{\sigma^2}{2}\right)(T-t)\right], \quad (A24)$$
$$d_- = d_+ - \sigma\sqrt{T-t},$$

where $\Phi(\cdot)$ denotes the standard normal cumulative distribution function. However, the uncertainty of the model has not yet been well studied in the literature. Accurately quantifying model uncertainty can significantly benefit decision-making, providing investors with a scientific foundation for making safer and more informed choices.

In this simulation experiment, we set $T = 1$, $\sigma = 0.5$, $r = 0.05$ and $K = 1$. The domain is defined on $\Omega = [0, T] \times [0, S_{\max}]$, where $S_{\max} = 2$. We assume the availability of 5 sensors at $t = 0$ for the price levels $S \in \{0.2, 0.4, 0.6, 0.8, 1.0\}$, with each sensor providing 10 replicate measurements. Measurement errors are modeled as $z_i^u \sim N(0, 0.05^2)$ for $i = 1, \ldots, 50$, representing noisy observations. For the boundaries at $\{S = 0\}$ and $\{t = T\}$, we use 50 sensors with noise-free measurements. For physical domain, we randomly pick 800 points from $\Omega$ to satisfy the Black-Scholes equation. The results of the simulation are presented in Table A10, where the metrics are evaluated at $t = 0$ and $t = 0.5$. At $t = 0$, the evaluation reflects the model's performance using noisy observed data, while at $t = 0.5$, the solutions are extended from the boundaries using the Black-Scholes equation. This setup highlights the model's ability to handle noisy observations and accurately propagate solutions over time through the governing equation. EFI demonstrates superior performance by providing not only the most accurate solutions, as evidenced by the lowest MSE, but also the most reliable confidence intervals.

Table A10: Comparison of different methods for the Black-Scholes Model, averaged over 100 runs: 'CR' refers to the coverage rate with a nominal value of 95%.

| Method | hidden layers | MSE($t=0$)($\times 10^{-4}$) | CR($t=0$) | CI-Width($t=0$) | MSE($t=0.5$)($\times 10^{-4}$) | CR($t=0.5$) | CI-Width($t=0.5$) |
|---|---|---|---|---|---|---|---|
| PINN | [50, 50] | 3.08 (0.44) | 0.1410 (0.0102) | 0.0046 (0.0002) | 15.73 (2.26) | 0.2427 (0.0192) | 0.0080 (0.0006) |
| Dropout (0.5%) | [50, 50] | 1.37 (0.20) | 0.5897 (0.0216) | 0.0190 (0.0002) | 2.19 (0.37) | 0.6303 (0.0252) | 0.0197 (0.0004) |
| Dropout (1.0%) | [50, 50] | 4.30 (2.59) | 0.6743 (0.0219) | 0.0244 (0.0005) | 1.83 (0.26) | 0.6983 (0.0255) | 0.0234 (0.0004) |
| Dropout (5.0%) | [50, 50] | 1.71 (0.70) | 0.9137 (0.0122) | 0.0538 (0.0004) | 1.70 (0.21) | 0.9387 (0.0110) | 0.0510 (0.0002) |
| Bayesian ($\sigma_f = 0.05$) | [50, 50] | 1.59 (0.41) | 0.9637 (0.0175) | 0.0516 (0.0010) | 12.61 (7.14) | 0.9413 (0.0187) | 0.0658 (0.0015) |
| Bayesian ($\sigma_f = 0.005$) | [50, 50] | 10.75 (1.46) | 0.5437 (0.0255) | 0.0388 (0.0007) | 39.39 (8.82) | 0.4807 (0.0225) | 0.0426 (0.0010) |
| **EFI** | **[50, 50]** | **0.38 (0.05)** | **0.9440 (0.0133)** | **0.0158 (0.0001)** | **0.17 (0.02)** | **0.9600 (0.0082)** | **0.0123 (0.0001)** |

To further illustrate these findings, we visualize the prediction surface in Figure A8. The figure reveals that B-PINN extends the solution poorly toward the edge at $S = 2$, where no data points are available,

relying solely on physical laws for extrapolation. In contrast, EFI provides a smoother and more accurate extension. The dropout method performs reasonably well for this example with a dropout rate of 5%; however, its confidence interval remains significantly wider than that of EFI. As previously noted, determining an appropriate dropout rate is not feasible without additional information. Figure A9 highlights EFI's ability to correctly quantify uncertainties. Near the boundary at $S = 0$, where boundary information is available, the confidence interval is appropriately narrow. As the stock price $S$ increases, and boundary information becomes scarce, the confidence interval widens, reflecting the growing uncertainty. In comparison, dropout and Bayesian methods fail to capture this behavior accurately. They produce overly broad or inconsistent intervals, particularly near the boundaries and regions with limited data, underscoring their limitations in handling uncertainty quantification for this problem.

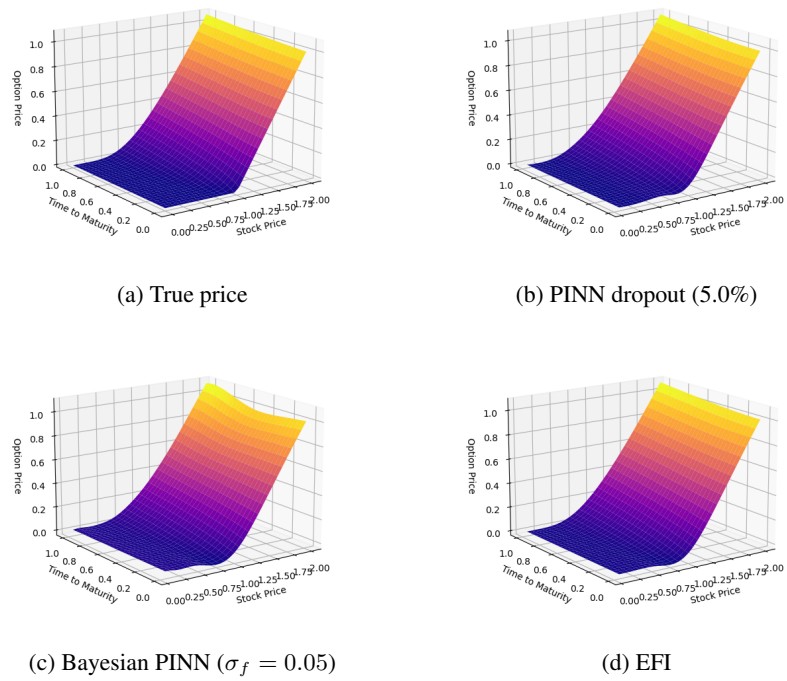

(a) True price                                  (b) PINN dropout (5.0%)

(c) Bayesian PINN ($\sigma_f = 0.05$)           (d) EFI

Figure A8: European Call Option Price.

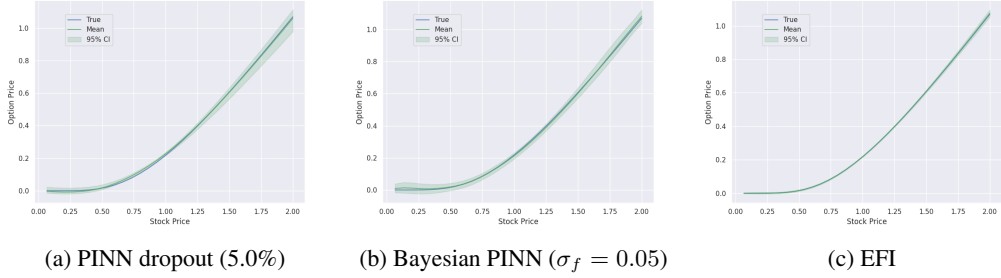

(a) PINN dropout (5.0%)    (b) Bayesian PINN ($\sigma_f = 0.05$)    (c) EFI

Figure A9: European Call Option Price at $t = 0$.

## A5.8   Real Data: Montroll Growth Model

Consider the Montroll growth model:

$$\frac{dp}{dt}(t) = k \cdot p(t) \cdot \left( 1 - \left( \frac{p(t)}{C} \right)^{\theta} \right), \tag{A25}$$

where $k$, $C$, and $\theta$ are unknown parameters. We applied this model to published data on the growth of Chinese hamster V79 fibroblast tumor cells (Marusic et al., 1994), which also appears in Rodrigues (2024). The dataset comprises 45 measurements of tumor volumes ($10^9$ vm$^3$) collected over a 60-day period. Table A11 shows the parameter estimates obtained using PINN, and Figure A10(a) shows the learned growth curve.

Table A11: Parameter estimates obtained with PINN for the model (A25).

| Parameter | Estimated Value |
|:---:|:---:|
| $k$ | 0.8311 |
| $C$ | 7.3327 |
| $\theta$ | 0.1694 |

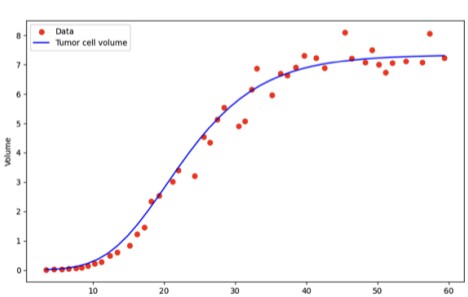

(a) Montroll growth curve learned with PINN

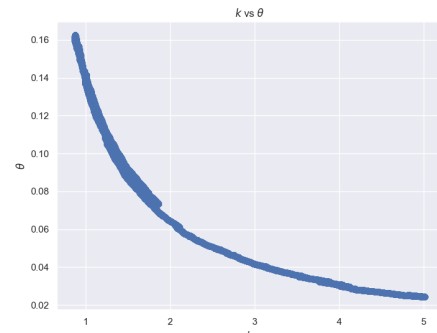

(b) Relationship between $k$ and $\theta$

Figure A10: Montroll growth Model for Chinese hamster V79 fibroblast tumor cells.

However, the standard PINN method does not provide confidence intervals for $p(t)$ or for the parameters $k$, $C$, and $\theta$. To apply EFI-PINN to this dataset, we assume a heteroscedastic noise structure given by $\sigma_t = \sqrt{t} \cdot \sigma$, where $\sigma$ is treated as an additional unknown parameter. Thus, we estimate four parameters in total under the EFI framework. Using EFI-PINN, we detected that $k$ and $\theta$ are nearly non-identifiable, see Figure A10(b), which shows their joint distribution by plotting their samples throughout training.

To address this non-identifiability issue, we fix $\theta = 0.1$ and reduce the model to:

$$\frac{dp}{dt}(t) = k \cdot p(t) \cdot \left( 1 - \left( \frac{p(t)}{C} \right)^{0.1} \right).$$

The corresponding confidence intervals for $p(t)$ are shown in the left plot of Figure 3. The confidence intervals of $k$, $C$, and $\sigma$ are given in Table A12.

Table A12: Parameter estimates (with 95% confidence intervals) obtained by EFI-PINN for the Montroll growth model.

| Parameter | 95% Confidence Interval | Mean |
|:---:|:---:|:---:|
| $\theta$ | – | 0.1 |
| $k$ | (1.25, 1.40) | 1.25 |
| $C$ | (7.30, 7.69) | 7.44 |
| $\sigma$ | (0.27, 0.47) | 0.36 |

The Montroll experiment highlights the strength of EFI in quantifying uncertainty for all parameters of interest. Moreover, it demonstrates EFI's ability to detect model identifiability issues, underscoring its utility in the statistical inference of scientific models. Additionally, EFI produces an estimate of $\sigma$, which enables the quantification of predictive uncertainty.

## A5.9 Real Data: FKPP Model and Porous-FKPP Model

Consider the Fisher–Kolmogorov–Petrovsky–Piskunov (FKPP) model and the porous FKPP (P-FKPP) model, which are governed by the following reaction-diffusion equations:

$$\frac{\partial u}{\partial t} = D\frac{\partial^2 u}{\partial x^2} + ru\left(1 - \frac{u}{K}\right), \tag{A26}$$

$$\frac{\partial u}{\partial t} = D\frac{\partial}{\partial x}\left[\left(\frac{u}{K}\right)^m \frac{\partial u}{\partial x}\right] + ru\left(1 - \frac{u}{K}\right), \tag{A27}$$

where $D$, $r$, and $m$ are unknown parameters, and $K$ denotes the carrying capacity. This equation has been used to model a wide range of growth and transport of biological processes. We applied it to scratch assay data (Jin and Cai, 2006). The biological experiments were conducted under varying initial cell densities — specifically, 10,000, 12,000, 14,000, 16,000, 18,000, and 20,000 cells per well. Cell densities were recorded at 37 equally-spaced spatial positions across five equally-spaced time points — specifically, 0.0 days, 0.5 days, 1.0 days, 1.5 days, and 2.0 days. See also Lagergren et al. (2020) for additional descriptions of the dataset. In addition to the dataset, we partitioned the space-time domain $[0, 2] \times [0, 2]$, where the first interval corresponds to the spatial domain and the second to the temporal domain, into a $50 \times 10$ grid for computing the PDE loss (i.e., the $f$-term in equation (10)). The fitting curves of the EFI algorithm for different models and initial density values are shown in Figures A11 and A12. Notably, beyond point estimation, EFI also constructs confidence intervals. Additionally, we note that the right plot of Figure 3 was generated using a parameter setting different from that listed in Table A20. In this setting, fewer sample points were allocated for computing the PDE loss, which resulted in smaller fitting errors but larger deviations from the assumed PDE model. Essentially, the two settings correspond to different datasets, as the number of sample points used to evaluate the energy function differs between them.

In Table A13, the root mean squared errors (RMSEs) are computed between the predicted solution $u$ and the observed data, reflecting a combination of epistemic and aleatoric uncertainty. Based on the model formulations in (A26) and (A27), the P-FKPP model is more flexible and is therefore expected to exhibit reduced epistemic uncertainty, leading to smaller RMSE values. Consistent with this expectation, Table A13 shows that the P-FKPP model achieves lower RMSEs compared to the standard FKPP model.

Regarding parameter uncertainty, we note that EFI is able to quantify the uncertainty associated each parameter. However, due to the transformation applied to the first term of (A27), the values of $D$ are no longer on the same scale across the two models, whereas the values of $r$ remain comparable in scale. The results are reported in Table A13.

**Summary**   Through both simulation and real data experiments, we have demonstrated that the proposed EFI algorithm effectively quantifies uncertainties associated with the model and the data-generating process, resulting in accurate estimation of both epistemic and aleatoric uncertainties.

Table A13: RMSE and estimated parameters with 95% confidence intervals for FKPP and Porous-FKPP models.

| Model | Initial Cell Density | RMSE | $D$ | Interval | $R$ | Interval | $M$ | Interval |
|---|---|---|---|---|---|---|---|---|
| FKPP | 10000 | 58.04 | 0.00936 | (0.00754, 0.01195) | 0.829 | (0.797, 0.877) | – | – |
| FKPP | 12000 | 82.09 | 0.00378 | (0.00281, 0.00461) | 0.632 | (0.603, 0.658) | – | – |
| FKPP | 14000 | 82.93 | 0.02929 | (0.02739, 0.03268) | 0.534 | (0.505, 0.585) | – | – |
| FKPP | 16000 | 99.14 | 0.02636 | (0.02503, 0.02789) | 0.608 | (0.585, 0.633) | – | – |
| FKPP | 18000 | 115.27 | 0.03784 | (0.03541, 0.04032) | 0.549 | (0.524, 0.575) | – | – |
| FKPP | 20000 | 136.67 | 0.05471 | (0.05007, 0.05817) | 0.492 | (0.458, 0.520) | – | – |
| P-FKPP | 10000 | 46.90 | 1167.67 | (72.48, 2719.97) | 0.846 | (0.832, 0.856) | 1.335 | (1.037, 1.490) |
| P-FKPP | 12000 | 67.88 | 1825.54 | (32.84, 4416.53) | 0.674 | (0.649, 0.696) | 1.433 | (1.033, 1.603) |
| P-FKPP | 14000 | 73.59 | 289.37 | (79.08, 552.13) | 0.625 | (0.600, 0.649) | 1.096 | (0.951, 1.199) |
| P-FKPP | 16000 | 70.83 | 57.36 | (19.21, 97.22) | 0.628 | (0.608, 0.650) | 0.920 | (0.804, 0.999) |
| P-FKPP | 18000 | 96.50 | 21.58 | (9.07, 35.78) | 0.563 | (0.536, 0.587) | 0.780 | (0.683, 0.863) |
| P-FKPP | 20000 | 123.34 | 1.472 | (1.058, 2.020) | 0.496 | (0.464, 0.530) | 0.408 | (0.370, 0.452) |

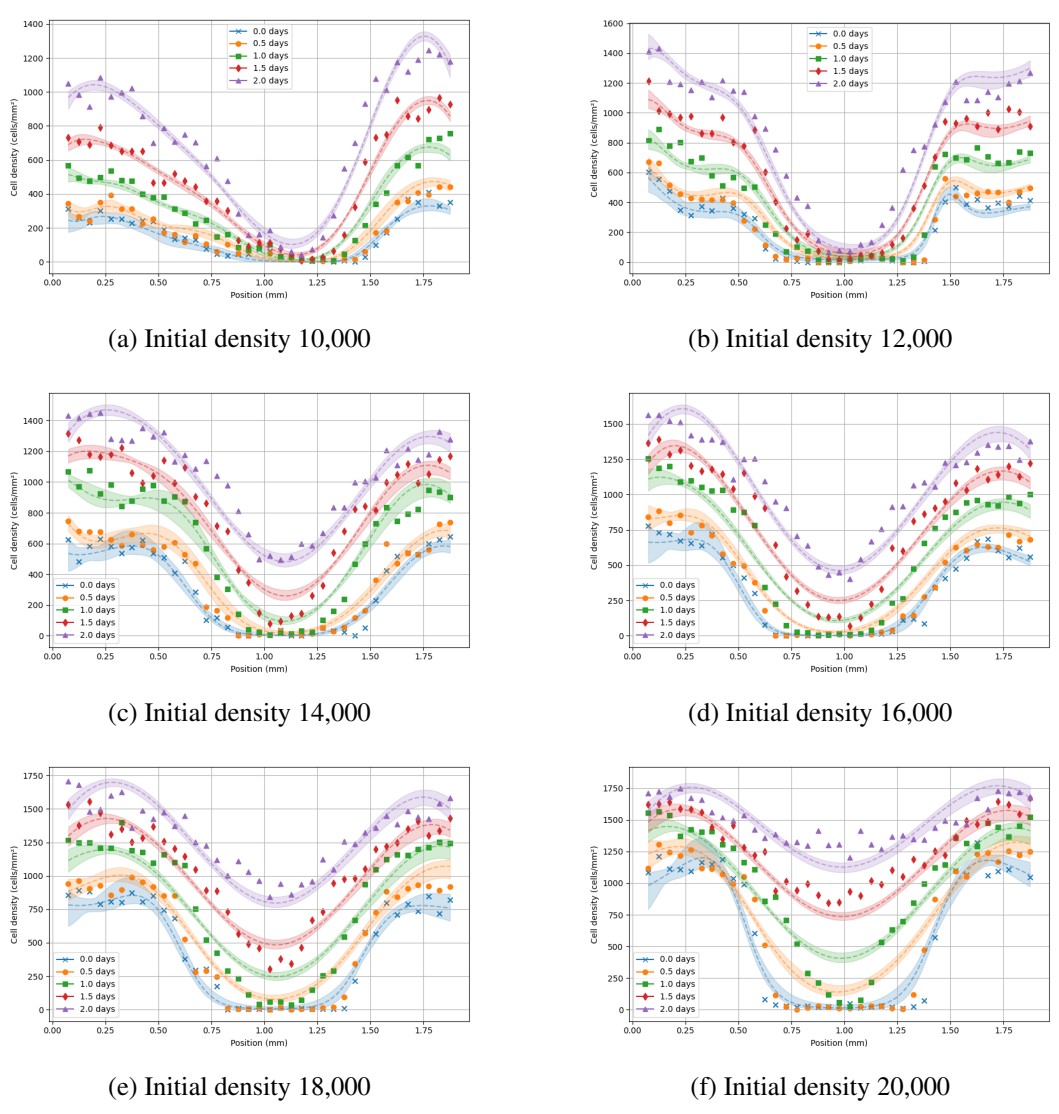

(a) Initial density 10,000

(b) Initial density 12,000

(c) Initial density 14,000

(d) Initial density 16,000

(e) Initial density 18,000

(f) Initial density 20,000

Figure A11: FKPP model

## A6 Experimental Settings

In all the simulation experiments, we begin by generating data from a specific physics model. Using the simulated data, we iteratively run the algorithm to estimate the model parameters. To enhance convergence of Algorithm 1, some algorithmic parameters (such as learning rate, SGD momentum, $\lambda = 1/\epsilon$, and etc.) are adjusted during the initial iterations, referred to as the annealing period. To tune different parameters, we use three different annealing schemes, including linear, exponential and polynomial. Their specific forms are in Table A14. For the nonlinear Poisson inverse problem, both the DNN parameters $\vartheta$ and the unknown parameter $k$ are estimated; while for all other problems, only the DNN parameters $\vartheta$ are estimated. At the end of the simulation, the samples collected in the burn-in period are discarded, and the samples collected in the remaining iterations are used for inference. The burn-in period is set to be at least as long as the annealing period across all experiments.

In our simulations, to ensure the weights of the $w$-network remain within a compact space as required in Assumption A4.1-(i), we impose a Gaussian prior, N(0,100), on each connection weight. However, due to the large variance, this prior has minimal impact on the algorithm's performance, serving primarily to ensure its stability.

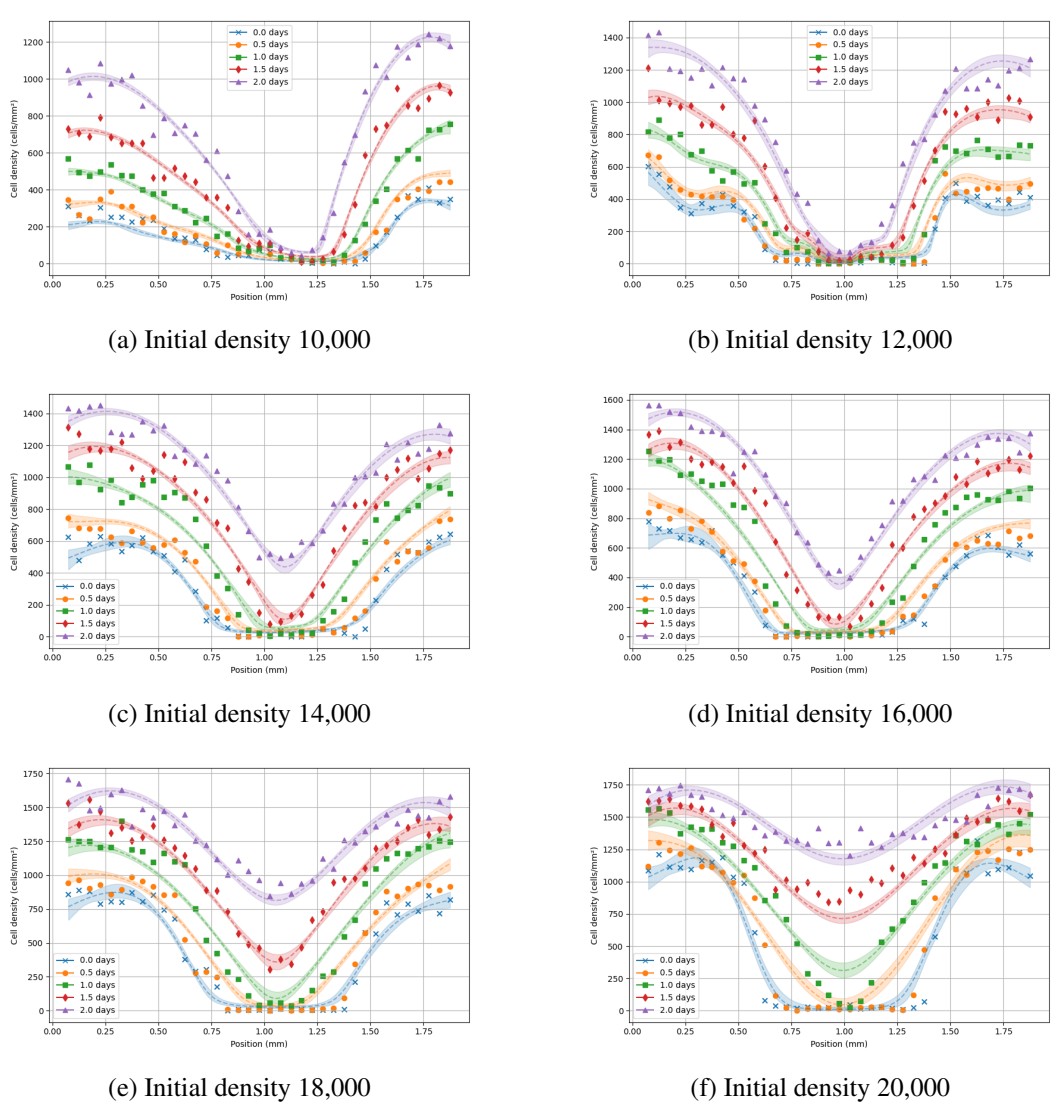

(a) Initial density 10,000

(b) Initial density 12,000

(c) Initial density 14,000

(d) Initial density 16,000

(e) Initial density 18,000

(f) Initial density 20,000

Figure A12: Porous-FKPP model

Table A14: Notations

| Notation | Meaning |
|---|---|
| epochs | total number of sampling/optimization iterations |
| burn-in period | the proportion of total iterations allocated to the burn-in process |
| annealing period | the proportion of total iterations allocated for parameter adjustment to enhance convergence |
| linear_x_y (with progress $\rho \in [0, 1]$) | $x + (y - x)\rho$ |
| exp_x_y (with progress $\rho \in [0, 1]$) | $x^{1-\rho} y^\rho$ |
| poly_x_c_k (with progress $\rho \in [0, 1]$) | $\frac{x}{1+(c\rho)^k}$ |

Table A15: Parameter settings for 1D-Poisson

| Parameter Name | PINN | Dropout | Bayesian PINN | EFI |
|---|---|---|---|---|
| $t_{\text{start}}$ | -0.7 | -0.7 | -0.7 | -0.7 |
| $t_{\text{end}}$ | 0.7 | 0.7 | 0.7 | 0.7 |
| noise sd in $u$ | 0.05 | 0.05 | 0.05 | 0.05 |
| noise sd in $f$ | 0.0 | 0.0 | 0.0 | 0.0 |
| # of solution sensors | 2 | 2 | 2 | 2 |
| # of solution replicates | 10 | 10 | 10 | 10 |
| # of differential sensors | 200 | 200 | 200 | 200 |
| # of differential replicates | 1 | 1 | 1 | 1 |
| epochs | 500000 | 500000 | 100000 | 200000 |
| burn-in period | 0.5 | 0.5 | 0.4 | 0.1 |
| hidden layers | $[50, 50]$ | $[50, 50]$ | $[50, 50]$ | $[50, 50]$ |
| activation function | tanh | tanh | tanh | tanh |
| learning rate | 3e-4 | 3e-4 | 1e-4/1e-4/linear_1e-4_3e-6 | poly_5e-6_100.0_0.55 |
| $\eta_f$ | 1.0 | 1.0 | / | 1.0 |
| dropout rate | / | 0.5%/1%/5% | / | / |
| $L$ | / | / | 6 | / |
| $\sigma_f$ | / | / | 0.05/exp_0.05_0.005/exp_0.05_0.0005 | / |
| $\sigma_u$ | / | / | 0.05 | / |
| annealing period | / | / | 0.3 | 0.1 |
| sgd momentum | / | / | / | linear_0.9_0.0 |
| sgld learning rate | / | / | / | poly_5e-6_10.0_0.55 |
| $\lambda$ | / | / | / | linear_50.0_500.0 |
| $\eta_\theta$ | / | / | / | 1.0 |
| encoder hidden layers | / | / | / | $[16, 16, 16]$ |
| encoder activation function | / | / | / | leaky relu |

Table A16: Parameter settings for 1D-Poisson (with $f$ error)

| Parameter Name | PINN | Dropout | Bayesian PINN | EFI |
|---|---|---|---|---|
| $t_{\text{start}}$ | -0.7 | -0.7 | -0.7 | -0.7 |
| $t_{\text{end}}$ | 0.7 | 0.7 | 0.7 | 0.7 |
| noise sd in $u$ | 0.05 | 0.05 | 0.05 | 0.05 |
| noise sd in $f$ | 0.05 | 0.05 | 0.05 | 0.05 |
| # of solution sensors | 4 | 4 | 4 | 4 |
| # of solution replicates | 10 | 10 | 10 | 10 |
| # of differential sensors | 40 | 40 | 40 | 40 |
| # of differential replicates | 10 | 10 | 10 | 10 |
| epochs | 100000 | 100000 | 50000 | 200000 |
| burn-in period | 0.5 | 0.5 | 0.4 | 0.1 |
| hidden layers | $[50, 50]$ | $[50, 50]$ | $[50, 50]$ | $[50, 50]$ |
| activation function | tanh | tanh | tanh | tanh |
| learning rate | 3e-4 | 3e-4 | 1e-4 | poly_2.5e-6_50.0_0.55 |
| $\eta_f$ | 1.0 | 1.0 | / | 1.0 |
| dropout rate | / | 0.5%/1%/5% | / | / |
| $L$ | / | / | 6 | / |
| $\sigma_f$ | / | / | 0.05 | / |
| $\sigma_u$ | / | / | 0.05 | / |
| annealing period | / | / | / | 0.1 |
| sgd momentum | / | / | / | linear_0.9_0.0 |
| sgld learning rate | / | / | / | poly_5e-6_100.0_0.55 |
| $\lambda$ | / | / | / | linear_50.0_1000.0 |
| $\eta_\theta$ | / | / | / | 1.0 |
| encoder hidden layers | / | / | / | $[64, 64, 16]$ |
| encoder activation function | / | / | / | leaky relu |

Table A17: Parameter settings for nonlinear 1D-Poisson (with $f$ error)

| Parameter Name | PINN | Dropout | Bayesian PINN | EFI |
|---|---|---|---|---|
| $t_{\text{start}}$ | -0.7 | -0.7 | -0.7 | -0.7 |
| $t_{\text{end}}$ | 0.7 | 0.7 | 0.7 | 0.7 |
| noise sd in $u$ | 0.05 | 0.05 | 0.05 | 0.05 |
| noise sd in $f$ | 0.05 | 0.05 | 0.05 | 0.05 |
| # of solution sensors | 4 | 4 | 4 | 4 |
| # of solution replicates | 10 | 10 | 10 | 10 |
| # of differential sensors | 40 | 40 | 40 | 40 |
| # of differential replicates | 10 | 10 | 10 | 10 |
| $k$ | 0.7 | 0.7 | 0.7 | 0.7 |
| epochs | 100000 | 100000 | 100000 | 200000 |
| burn-in period | 0.5 | 0.5 | 0.4 | 0.1 |
| hidden layers | $[50, 50]$ | $[50, 50]$ | $[50, 50]$ | $[50, 50]$ |
| activation function | tanh | tanh | tanh | tanh |
| learning rate | 3e-4 | 3e-4 | 1e-4 | poly_5e-6_100.0_0.55 |
| $\eta_f$ | 1.0 | 1.0 | / | 1.0 |
| dropout rate | / | 0.5%/1%/5% | / | / |
| $L$ | / | / | 6 | / |
| $\sigma_f$ | / | / | exp_0.2_0.05 | / |
| $\sigma_u$ | / | / | 0.05 | / |
| annealing period | / | / | 0.3 | 0.1 |
| sgd momentum | / | / | / | linear_0.9_0.0 |
| sgld learning rate | / | / | / | poly_5e-6_100.0_0.55 |
| $\lambda$ | / | / | / | exp_50.0_1000.0 |
| $\eta_\theta$ | / | / | / | 1.0 |
| encoder hidden layers | / | / | / | $[64, 64, 16]$ |
| encoder activation function | / | / | / | leaky relu |

Table A18: Parameter settings for nonlinear 1D-Poisson with parameter estimation

| Parameter Name | PINN | Dropout | Bayesian PINN | EFI |
|---|---|---|---|---|
| $t_{\text{start}}$ | -0.7 | -0.7 | -0.7 | -0.7 |
| $t_{\text{end}}$ | 0.7 | 0.7 | 0.7 | 0.7 |
| noise sd in $u$ | 0.05 | 0.05 | 0.05 | 0.05 |
| noise sd in $f$ | 0.0 | 0.0 | 0.0 | 0.0 |
| # of solution sensors | 8 | 8 | 8 | 8 |
| # of solution replicates | 10 | 10 | 10 | 10 |
| # of differential sensors | 200 | 200 | 200 | 200 |
| # of differential replicates | 1 | 1 | 1 | 1 |
| $k$ | 0.7 | 0.7 | 0.7 | 0.7 |
| epochs | 50000 | 500000 | 50000 | 350000 |
| burn-in period | 0.5 | 0.5 | 0.4 | 0.1 |
| hidden layers | $[50, 50]$ | $[50, 50]$ | $[50, 50]$ | $[50, 50]$ |
| activation function | tanh | tanh | tanh | tanh |
| learning rate | 3e-4 | 3e-4 | 1e-4 | poly_5e-6_100.0_0.55 |
| $\eta_f$ | 1.0 | 1.0 | / | 1.0 |
| dropout rate | / | 0.5%/1%/5% | / | / |
| $L$ | / | / | 10 | / |
| $\sigma_f$ | / | / | 0.05 | / |
| $\sigma_u$ | / | / | 0.05 | / |
| annealing period | / | / | / | 0.1 |
| sgd momentum | / | / | / | linear_0.9_0.0 |
| sgld learning rate | / | / | / | poly_5e-6_100.0_0.55 |
| $\lambda$ | / | / | / | linear_50.0_1000.0 |
| $\eta_\theta$ | / | / | / | 1.0 |
| encoder hidden layers | / | / | / | $[128, 128, 12]$ |
| encoder activation function | / | / | / | leaky relu |

Table A19: Parameter settings for Black-Scholes Model

| Parameter Name | PINN | Dropout | Bayesian PINN | EFI |
|---|---|---|---|---|
| $S$ range | $[0.0, 2.0]$ | $[0.0, 2.0]$ | $[0.0, 2.0]$ | $[0.0, 2.0]$ |
| $t$ range | $[0.0, 1.0]$ | $[0.0, 1.0]$ | $[0.0, 1.0]$ | $[0.0, 1.0]$ |
| $\sigma$ | 0.5 | 0.5 | 0.5 | 0.5 |
| $r$ | 0.05 | 0.05 | 0.05 | 0.05 |
| $K$ | 1.0 | 1.0 | 1.0 | 1.0 |
| noise sd | 0.05 | 0.05 | 0.05 | 0.05 |
| # of price sensors | 5 | 5 | 5 | 5 |
| # of price replicates | 10 | 10 | 10 | 10 |
| # of boundary samples | 50 | 50 | 50 | 50 |
| # of differential samples | 800 | 800 | 800 | 800 |
| epochs | 200000 | 200000 | 100000 | 300000 |
| burn-in period | 0.5 | 0.5 | 0.4 | 0.1 |
| hidden layers | $[50, 50]$ | $[50, 50]$ | $[50, 50]$ | $[50, 50]$ |
| activation function | softplus($\beta = 5$) | softplus($\beta = 5$) | softplus($\beta = 5$) | softplus($\beta = 10$) |
| learning rate | 3e-4 | 3e-4 | 1e-4/linear_1e-4_1e-5 | poly_5e-6_100.0_0.55 |
| $\eta_f$ | 1.0 | 1.0 | / | 1.0 |
| dropout rate | / | 0.5%/1%/5% | / | / |
| $L$ | / | / | 6 | / |
| $\sigma_f$ | / | / | 0.05/exp_0.05_0.005 | / |
| $\sigma_u$ | / | / | 0.05 | / |
| pretrain epochs | / | / | 5000 | / |
| annealing period | / | / | 0.3 | 0.1 |
| sgd momentum | / | / | / | linear_0.9_0.0 |
| sgld learning rate | / | / | / | poly_5e-6_100.0_0.55 |
| sgld alpha | / | / | / | 1.0 |
| $\lambda$ | / | / | / | linear_50.0_1000.0 |
| $\eta_\theta$ | / | / | / | 1.0 |
| encoder hidden layers | / | / | / | $[64, 64, 16]$ |
| encoder activation function | / | / | / | leaky relu |

Table A20: Parameter settings for real data

| Parameter Name | Montroll growth | FKPP | P-FKPP |
|---|---|---|---|
| epochs | 200000 | 200000 | 200000 |
| burn-in period | 0.1 | 0.1 | 0.1 |
| hidden layers | $[50, 50]$ | $[50, 50]$ | $[50, 50]$ |
| activation function | softplus($\beta = 10$) | tanh | tanh |
| learning rate | poly_1e-6_10.0_0.55 | poly_1e-6_10.0_0.55 | poly_1e-6_10.0_0.55 |
| $\eta_f$ | 1.0 | 1.0 | 1.0 |
| sgd momentum | 0.9 | 0.9 | 0.9 |
| sgld learning rate | poly_1e-4_10.0_0.95 | poly_1e-6_10.0_0.95 | poly_1e-6_10.0_0.95 |
| sgld alpha | 1.0 | 1.0 | 1.0 |
| $\lambda$ | log_50.0_500.0 | log_50.0_500.0 | log_50.0_500.0 |
| $\eta_\theta$ | 1.0 | 1.0 | 1.0 |
| encoder hidden layers | $[32, 32, 16]$ | $[64, 64, 16]$ | $[64, 64, 16]$ |
| encoder activation function | leaky relu | leaky relu | leaky relu |

