# OpenReview forum: "Uncertainty Quantification for Physics-Informed Neural Networks with Extended Fiducial Inference"
_NeurIPS.cc/2025/Conference — NeurIPS 2025 poster_

### Official Review · Reviewer_4Xoo · 2025-06-30

**Clarity:** 3
**Significance:** 3
**Originality:** 2
**Rating:** 4
**Confidence:** 2

**Summary:**

This paper introduces a novel approach to uncertainty quantification for Physics-Informed Neural Networks (PINNs), based on the recently proposed Extended Fiducial Inference (EFI) framework by Liang et al. (2025). A key advantage of EFI is that it circumvents the potentially misleading specification of a prior distribution, which is often a concern in Bayesian frameworks—particularly in approaches like MC Dropout where even the dropout rate can implicitly act as a prior. The proposed algorithm is empirically validated through several experiments.

**Questions:**

N/A

**Ethical Concerns:**

["NO or VERY MINOR ethics concerns only"]

**Final Justification:**

After reading the response from the authors and the author's response to other reviewers, I feel my score 4 borderline accept is a fair assessment. Many claims in this paper are very much based on a previous paper on Extended Fiducial Inference (EFI) framework by Liang et al. (2025), for which I would encourage the authors to give a clearer explanation of the related necessary background knowledge on EFI. In this current format, as a reviewer, I have huge difficulty understanding and evaluating the real contribution and potential impact of this paper. This is my major negative perspective of this paper. The rest of the weakness stated in my reviews, like the validity and presentation of the assumption, can be addressed should the authors spend more time polishing the paper. I look forward to see another updated version of this paper.

**Limitations:**

Yes

**Quality:**

3

**Strengths And Weaknesses:**

###Do we still require a prior distribution in EFI?###

While I acknowledge that I am not an expert in fiducial inference, I find the concept of EFI quite confusing. Based on Appendix A1, it seems that a prior $\pi(w_n)$ is still required. If so, how does EFI avoid the pitfalls associated with overly informative priors that could overshadow the observed data? A more transparent discussion of this point would be helpful.

###Why is $\delta_g = 0$?###

In the paragraph following Lemma 3.1, the authors claim that with imputation at each step for every observation, we have $\delta_g = 0$. However, this is not obvious to me. This assumption is crucial since it ensures the Wasserstein-2 distance bound vanishes as both $T$ and $k$ tend to infinity. I request a more detailed and rigorous explanation for why $\delta_g = 0$ under the described setup.

##Eigenvalue assumption###

The paper assumes a uniform lower bound on the minimal eigenvalue of $E[m_i m_i^\top]$. This is a strong assumption, and I would appreciate a simple toy example or practical justification showing when and why this condition would hold in real-world scenarios.

###Interpretation of Theorem 3.2###

Theorem 3.2 states an asymptotic consistency result on the estimation of the inverse function $G$ with finite samples. However, the practical implications of this result are unclear. Can the authors provide a more quantitative version—perhaps a sample or iteration complexity bound that shows how fast the method converges to within an error $\delta$?

============================

Honestly, I find the paper difficult to review—not necessarily due to its quality, but because it builds on EFI, a relatively new and non-standard inference framework that is challenging to digest within the limited time available for conference reviews. Furthermore, I am not deeply familiar with the PINN literature, which made the model specification in Section 3.2 particularly hard to follow.

Given these factors, I strongly encourage the authors to consider submitting this paper to a journal venue, where the review process would allow for a more careful and thorough evaluation. For now, I will defer to the Area Chair for a final decision.

---

> ### Author Rebuttal · Authors · 2025-07-31
>
> **[W1.] Do we still require a prior distribution in EFI?**
>
> In Appendix~A1, we reviewed the original EFI theory, which requires imposing a sparse prior on $w$ to guarantee posterior consistency for the hypernetwork. Instead of adopting a Bayesian approach on the hypernetwork, we designed a narrow-neck network architecture to eliminate the need for prior assumptions on $w$.
>
> **[W2.] Why is $\delta_g = 0$?**
>
> Thanks for pointing this out. As mentioned in Theorem 4.2 of Liang et al. (2025), since we used the full dataset to calculate the gradient, the stochastic gradient reduces to the true gradient. Hence, $\delta_g = 0$.
>
> [1] Liang, F., Kim, S., and Sun, Y. Exended fiducial inference: Toward an automated process of statistical inference. Journal of the Royal Statistical Society Series B, 87:98–131, 2025.
>
>
> **[W3.] Eigenvalue assumption**
>
> In real-world implementations, when the penalty coefficient $\eta_\theta$ in Equation~(10) is large, the estimated parameters $\hat{\theta}\_i $ tend to be almost identical across all $i$. Since the $\hat{\theta}\_i$'s are highly correlated, the same is true for the corresponding $m_i$'s. As a result, $\mathbf{E}[m_i m_i^T]$ is nearly a rank-1 matrix, with one dominant eigenvalue and all others close to zero. On the other hand, if $\eta\_\theta$ is not sufficiently large, more variation is allowed across the $\hat{\theta}\_i$'s, and the lower bound for the minimum eigenvalue of $\mathbf{E}[m_i m_i^T]$ will be slightly larger. In theory, as $\eta_\theta \to \infty$, the bottleneck size could be reduced to $1$. In practice, however, $\eta_\theta$ does not approach infinity, so we choose a slightly larger bottleneck size to promote more robust convergence of the algorithm while still satisfying the eigenvalue assumption.
>
> **[W4.] Interpretation of Theorem 3.2**
>
> The asymptotic consistency of the inverse function essentially ensures a one-to-one correspondence between $ Z_n $ and $\theta $; that is, different realizations of the random error $Z_n$ lead to different estimates of $ \theta $, thereby ensuring that the uncertainty in $Z_n $ is properly propagated to $ \theta $ and  the uncertainty in $\theta$ is accurately quantified.

---

> ### Comment · Reviewer_4Xoo · 2025-08-01
> **Thank you for the response**
>
> To be honest, I am not able to verify the correctness of the authors' response. This may be due to the simplicity or lack of clarity in the author's rebuttal provided, or it may reflect my own limited familiarity with the relevant literature. Given that the paper builds heavily on Liang et al. (2025), which has only recently been accepted to JRSSB, it is understandably difficult for most readers to properly assess the current paper.
>
> 1. It seems that a sparse prior distribution is still needed for EFI framework. Even after reading the paper again, I still do not understand why a narrow-neck neural network eliminates the need for a prior.
>
> 2. it relies heavily on Liang et al. (2025), which I am not able to verify its correctness.
>
> 3. I can understand the first half of the response. However, for the second half, why the lower bound for the minimum eigenvalue of $\mathbf{E}[m_i m_i^T]$ increases, when more variation is allowed across the $\hat{\theta}_i$?
>
> 4. I do not understand author's response. I am asking about the non-asymptotic convergence of the inverse function with finite samples; while the author's response tells me how the uncertainty in $Z_n $ is properly propagated to $ \theta $. I am completely lost here.

---

> ### Author Response · Authors · 2025-08-01
> **Reply to follow-up questions**
>
> Thank you for your prompt and valuable comments. Your advice will be incorporated into our revision.
>
> **[Q1]**
>
> We apologize for any confusion. In the original EFI framework by Liang et al. (2025), a sparse prior is imposed on
> the $w$-network to achieve consistency for the inverse function estimator. As stated on page 2, one of the goals of the present paper is to develop a new EFI framework by employing a narrow-neck
> $w$-network, such that the sparse prior on the $w$-network can be eliminated, while still achieving consistency for the inverse function estimator. We establish this consistency in Theorem 3.2.
>
> In the proof, we introduce an auxiliary stochastic neural network (StoNet), which is constructed by randomizing the feeding values of the hidden units at the neck-layer by adding a small random error (see Equation (12) of the present paper).
> This StoNet was originally developed in Liang et al. (2022) and
> Sun & Liang (2022), and it
> has been shown to be able to bridge statistical theory from linear models to deep neural networks (Sun & Liang, 2025).
> Specifically, StoNet is asymptotically equivalent to the conventional
> DNN as the size (standard deviation) of the added random errors
> approaches 0 (see Liang et al. (2022)} for the proof).
> Building on the properties of the StoNet, we show in the present paper
> that the narrow-neck structure ensures the inverse function
> can be consistently estimated.
>
> This result is extremely important for EFI, as it completely eliminates the need for priors, aligning with the core goal of fiducial inference --- inferring the uncertainty of model parameters based solely on observations.
>
> [1]  Liang, F., Kim, S., and Sun, Y. Exended fiducial inference: Toward an automated process of statistical inference. Journal of the Royal Statistical Society Series B, 87:98–131, 2025.
> [2]  Liang, S., Sun, Y., and Liang, F. Nonlinear sufficient dimension reduction with a stochastic neural network. NeurIPS 2022, 2022.
> [3]   Sun, Y. and Liang, F. A kernel-expanded stochastic neural network. Journal of the Royal Statistical Society Series B, 84(2):547–578, 2022.
> [4]   Sun, Y. and Liang, F. Uncertainty quantification for large-scale deep neural networks via post-stonet modeling. Statistica Sinica, pp. in press, 2025. URL https://www3.stat.sinica.edu.tw/ss_newpaper/ SS-2024-0294_na.pdf.
>
> **[Q2]**
>
> The concept of EFI presented in the paper is from Liang et al. (2025) and a more recent work Kim and Liang (2025); both work involves solid theoretical studies. However, as mentioned above, the present work represents an important development of EFI. It completely eliminates the need for priors, aligning with the core goal of fiducial inference — inferring the uncertainty of model parameters based solely on observations.
>
> [5] Kim, S. and Liang, F. Extended fiducial inference for individual treatment effects via deep neural networks. Statistics and Computing, 35, 2025. URL https://api.semanticscholar.org/CorpusID:278326914.
>
> **[Q3]**
>
> Thank you for the thoughtful question. In a nutshell,
> the output of the $w$-network and the output of the last hidden layer together forms a multivariate regression (see equation (11) of the paper) under the framework of the stochastic neural network:
> $$\hat{\theta}_i=\Xi^T m_i +e_i,  \quad i=1,2,\ldots,n $$
> where $\hat{\theta}_i=(\hat{\theta}_i^{(1)}, \ldots, \hat{\theta}_i^{(p)}))^T \in \mathbb{R}^{p}$, $m_i \in \mathbb{R}^{d_h}$, $\Xi=(\xi_1,\ldots,\xi_p) \in \mathbb{R}^{d_h\times p}$, and $e_i \in \mathbb{R}^p$.
> However, this regression differs from conventional regression, where $\hat{\theta}_i$ can be treated as known (as $\epsilon \to 0$), while $m_i$ is random (imputed conditioned on the input
> and output of the network).
> In the limit,  as the standard deviation of $e_i$ approaches 0,
> we have
> $$(\hat{\theta}_1, \ldots,\hat{\theta}_n)^T=(m_1,\ldots,m_n)^T \Xi, $$
> or, in matrix form,
> $$\hat{\Theta}= M \Xi, $$
> Therefore, $rank(\hat{\Theta}) \leq rank(M)$.
>
> In our experiments, we observe that if $\eta_{\theta}$ is small, which increases the variation of $\hat{\Theta}$, the smallest eigenvalues of $M$ increases.
> In fact, what we intend to convey is that if $\eta_{\theta}$ is small, we need to adjust the neck-layer to be slightly wider. By appropriately choosing $\eta_{\theta}$ and the neck-layer width $d_h$, the eigenvalue lower bound condition can be satisfied.
>
> **[Q4]**
>
> We apologize for the misunderstanding — we initially thought you were requesting an explanation of the ``practical implications of this result.''
>
> Regarding the convergence rate of the inverse function estimator (i.e., non-asymptotic convergence with finite samples), we note that this has been implicitly established in our proof of Theorem 3.2. As indicated by equations (A20) and (A21) (in the supplementary material), the convergence rate is $O(1/\sqrt{n})$, which is an ideal rate.

---

> > ### Comment · Reviewer_4Xoo · 2025-08-09
> > **Thank you for the response**
> >
> > I would like to thank the authors for their response, which have addressed some of my concerns. However, given the content of the paper and the clarify of the introduction on EFI, I maintain my score of 4 which I think is a fair assessment.

---

### Official Review · Reviewer_Uzpp · 2025-07-01

**Clarity:** 3
**Significance:** 3
**Originality:** 3
**Rating:** 4
**Confidence:** 3

**Summary:**

The paper proposes Extended Fiducial Inference (EFI)  to attach calibrated uncertainty estimates to physics-informed neural networks (PINNs).  EFI learns a slim “narrow-neck” hyper-network that maps observed data plus latent noise to the PINN’s weights; by Monte-Carlo sampling the latent noise one obtains predictive intervals whose width automatically adjusts to data quality. The authors evaluate their method on artificial and more real-world physics problems.

**Questions:**

Eq. (1)

the uncertainty of z comes from things that can't be measured (noise, partial observability). This can be called aleatoric uncertainty.

However, the uncertainty over \theta  will shrink as more data becomes available, it can be called epistemic.

But see https://iclr.cc/virtual/2025/poster/31334 for a more critical take on this.

It can be argued  that fusing the uncertainty from z into w will create epistemic and aleatoric uncertainty.

Example:
let y = \theta_1*x*z  + \theta_2*x*(1-z) and z from Bernoulli({0,1})
\theta and z are latent. As we collect more data in the form of (x,y) pairs ideally the posterior over
 \theta will shrink its posterior as more data is available (epistemic). Uncertainties over \theta can best be understood by uncertainty over the slope of the bimodal function.


Eq. (2) will lead to \theta containing both the "slope uncertainty" as well as the variation that originates from the bimodality. It is thus a mix of epistemic and aleatoric uncertainty.

What kind of uncertainty is being captured once EFI pushes the observation noise through G?
Can we still distinguish whether high uncertainty over \theta originates from lack of data or from a high noise level? (i.e. are epistemic and aleatoric components recoverable?)

**Ethical Concerns:**

["NO or VERY MINOR ethics concerns only"]

**Final Justification:**

my initial review still stands. i believe the papers contribution license acceptance, but I'm still cautious regarding practicality and how principled epistemic and aleatoric uncertainty modeling is.

**Quality:**

3

**Strengths And Weaknesses:**

Strenghts:

- (+) The paper is written very well. Especially Section 1 & 2 are excellent.

- (+) The paper is adressing an important area of research (uncertainty modeling) and the EFI method is an underexplored subfield.

- (+) The experiments make sense given the research question and are evaluated in a clean way. (Some design choices like showing confidence bands, coverage can be questionable: for instance for uncertainty modeling a standard method is to calculate the avg. log-likelihood, ECE etc.).
- (+/-)  Carried over from EFI: Compared to variational inferences, the uncertainty modeling approach has few free hyper-parameters and does not need explicit prior modeling. However, some typical drawback of MCMC carry over: (algorithm 1: burn-in periods, difficulty to check convergence, sampling process usually slower than energy-loss minimization.  perhaps wall-clock comparison of the algorithm compared to dropout, ensembling and the likes?)

- (-)  It is unclear whether epistemic uncertainty is truly captured by the approach (or if its both epistemic and aleatoric which can't be disentangled, see questions below). There is a  lack of discussion on the epistemic and aleatoric uncertainty concept in Section 2. Also the approach appears to share similarity with the BNN+LV approach (see e.g. Decomposition of Uncertainty in Bayesian Deep Learning for Efficient and Risk-sensitive Learning). In Appendix A5 these concepts are discussed, but very limited y = f(x) + \epsilon (normal, uniform noise) is quite limited noise assumption.
Will the method still work if the noise is heteroscedastic?
what about the most general case, where if y = f(x,z) and z is from some distribution?
Can you refute the following statement: "EFI conflates epistemic & aleatoric uncertainty"?

---

> ### Author Rebuttal · Authors · 2025-07-31
>
> **[W1.] Carried over from EFI: Compared to variational inferences, the uncertainty modeling approach has few free hyper-parameters and does not need explicit prior modeling. However, some typical drawback of MCMC carry over: (algorithm 1: burn-in periods, difficulty to check convergence, sampling process usually slower than energy-loss minimization. perhaps wall-clock comparison of the algorithm compared to dropout, ensembling and the likes?)**
>
> In Table 1, we report the wall time for the Poisson-1D experiment using different algorithms. As shown, EFI is slower than PINN (with dropout) but much faster than B-PINN with the HMC sampler. Importantly, we have demonstrated that EFI is the only existing method capable of correctly constructing confidence intervals in a statistically rigorous manner. Since we employ a narrow-neck hypernetwork structure, the total number of parameters depends linearly on the size of the neck layer. Therefore, EFI is scalable with respect to the size of the $\theta$-network. For even larger networks, we can adopt transfer learning techniques by constructing EFI hypernetworks only for the last few layers of the $\theta$-network, which would significantly reduce the computational cost.
>
> It is worth noting that the EFI training process is based on the Adaptive SGMCMC algorithm, which alternates between one step of SGD and one step of SGLD. Since both SGD and SGLD are stochastic gradient-based algorithms, EFI has the same order of sample complexity as standard optimization-based approaches. However, because EFI aims to construct accurate confidence intervals, it generally requires more training time compared to purely optimization-focused methods.
>
> **Table:** Wall time for PINN, B-PINN and EFI-PINN
> (Device: MacBook Pro M4 Pro CPU)
>
> | Algorithm       | Hypernetwork   | Epoch (×10³) | Wall time (s) | Time per epoch (ms) |
> |-----------------|----------------|--------------|---------------|---------------------|
> | PINN (dropout)  | -              | 200          | 133           | 0.665               |
> | B-PINN          | -              | 100          | 1330          | 13.300              |
> | EFI             | [16,16,16]     | 200          | 391           | 1.955               |
> | EFI             | [16,16,4]      | 200          | 385           | 1.925               |
>
>
> **[W2.] It is unclear whether epistemic uncertainty is truly captured by the approach (or if it's both epistemic and aleatoric which can't be disentangled, see questions below). There is a lack of discussion on the epistemic and aleatoric uncertainty concept in Section 2. Also the approach appears to share similarity with the BNN+LV approach (see e.g. Decomposition of Uncertainty in Bayesian Deep Learning for Efficient and Risk-sensitive Learning). In Appendix A5 these concepts are discussed, but very limited $y = f(x) + \epsilon$ (normal, uniform noise) is quite limited noise assumption. Will the method still work if the noise is heteroscedastic? what about the most general case, where if $y = f(x,z)$ and $z$ is from some distribution? Can you refute the following statement: "EFI conflates epistemic \& aleatoric uncertainty"?**
>
> As explained below (see our replies to Questions), EFI integrates epistemic \& aleatoric uncertainty in a mathematically rigorous way. The method still works if the noise is heteroscedastic. In this case, we can set $\epsilon$ to a very small value, enabling the noise to be recovered. Actually, this has been tested in Liang et al. (2025), where the example in Section 5.4 demonstrates that EFI works with heteroscedatic noise (or outliers).
>
> Yes, the method also works in the most general case, where $y=f(x,z)$. Refer to Liang et al. (2025) for the theoretical development on this case, where the random error does not appear in the model in an additive form.
>
> [1] Liang, F., Kim, S., and Sun, Y. Exended fiducial inference: Toward an automated process of statistical inference. Journal of the Royal Statistical Society Series B, 87:98–131, 2025.
>
> **[Questions]**
>
> Thank you very much for the thoughtful comments. EFI aims to
> learn an inverse mapping for the model parameters, i.e., $\theta=G(Y_n,X_n,Z_n)$, through which the uncertainty of $\theta$ can be derived from the uncertainty of $Z_n$ (in the form of variable transformation as shown in equation (8) of the paper).
>
> To make equation (8) (of the paper) more interpretable in the context of uncertainty quantification, we
> can rewrite it in a less formal manner as follows:
> $$
> \mu_n^*(\theta|Y_n,X_n)= \int \pi(\theta|Y_n,X_n,Z_n) \pi(Z_n|Y_n,X_n) dZ_n,
> $$
> where $\pi(Z_n|Y_n,X_n)$ provides a measure for the
> aleatoric uncertainty, while $\pi(\theta|Y_n,X_n,Z_n)$ provides
> a measure for the epistemic uncertainty (whose variation can decrease to 0
> as $n \to \infty$) and depends on the aleatoric uncertainty. That is, the uncertainty measured by EFI has integrated both types of uncertainty.
> We agree with Kirchhof et al. (2025) that distinguishing between these two types of uncertainty is not always feasible and one
> can depend on the other.
>
> To better explain this concept, let's consider a linear regression example:
> $$
> y=x\beta+ Z,
> $$
> where $Z$ follows the standard Gaussian distribution.
>  As shown in Liang et al. (2025), an
> inverse function for $\beta$ is given by
> $$
> G(Y_n,X_n,Z_n)=(X_n^TX_n)^{-1} X_n^T(Y_n-Z_n).
> $$
> Integrating with respect to the fiducial density of $Z_n$,  the fiducial
> density of $\beta$ is given by
> $$
> N((X_n^T X_n)^{-1} X_n^T Y_n, (X_n'X_n)^{-1}).
> $$
> This theoretical result has been numerically verified in Liang et al. (2025) through the implementation of EFI.
>
> Next, let's consider the binary response example you provided.
> Given a set of $Z$-values,
> directly solving the data-generating equation leads to
> $$
> \theta|y_i, x, z_i=\begin{cases} \frac{y_i}{x} (:=\theta_{1}), & \mbox{if $z_i=1$} \newline
>    \frac{y_i}{x} (:=\theta_{2}), & \mbox{if $z_i=0$}, \\
>    \end{cases}
> $$
> for $i=1,2,\ldots,n$.
>
> Let $(z_i^{(1)}: i=1,2,\ldots,n)$, $\ldots$, $(z_i^{(m)}: i=1,2,\ldots,n)$
> denote $m$ imputations from $\pi(Z_n|Y_n,X_n)$ in EFI, which measures the aleatoric uncertainty.
> By empirically integrating out $Z_n$ as what does in EFI, we obtain
> $$
> \theta=\begin{cases} \theta_1, & \mbox{with prob $\hat{p}$} \newline
>         \theta_2, & \mbox{with prob $1-\hat{p}$}, \\
>      \end{cases}
> $$
>
> where $\hat{p}$ is given by
> $$ \hat{p}= \frac{1}{m} \sum_{k=1}^m \frac{ \\# \\{i: z_i^{(k)}=1\\} }{n}, $$
> whose variance decreases to 0 as $n\to \infty$.
> Again, the uncertainty of $\theta$ has integrated both
> the aleatoric and epistemic uncertainty. EFI can be implemented for the example by replacing its latent variable sampling algorithm with a Metropolis-based algorithm (dealing with discrete values of Z).

---

> ### Author Response · Authors · 2025-08-05
>
> Thank you very much for your encouraging comments.
>
> **Error heterogeneity:** We refer to Section 5.4 of Liang et al. (2025) and Figure 5 therein. Although the example is primarily designed for outlier detection, it also serves as an effective illustration of parameter estimation under heterogeneous error structures. The results demonstrate that the proposed method is robust to heterogeneous errors.
>
> **Computational efficiency:** We acknowledge that the method is slower than variational inference (VI), primarily due to its sampling-based nature, as the method draws samples of $\theta$ from its fiducial distribution. However, the trade-off is our method's ability for uncertainty quantification. Moreover, our method is still scalable with respect to big data problems.
>
> **Self-diagnostic capability:** The method possesses a built-in self-diagnostic mechanism, as it must converge to a zero-energy solution to ensure valid statistical inference. This feature enhances usability and robustness, particularly in accommodating variations in hyper-network parameters, which are typically non-identifiable following the general property of deep neural networks.
> The zero-energy convergence feature also partially address the issue of **multi-modality** (that you raised), which ensures the convergence to a solution to the data-generating equation.
>
> **Conceptual contribution and practical potential:** We agree with your assessment that the paper offers a novel conceptual framework for uncertainty quantification. We also believe that the proposed method has strong practical potential. Notably, it can be extended to large-scale models, such as ResNets and CNNs, via transfer learning, as discussed at the end of the paper.  Compared to existing approaches, the proposed method not only applies to a wide range of problems but also provides a rigorous framework for uncertainty quantification.
>
> [1] Liang, F., Kim, S., and Sun, Y. Exended fiducial inference: Toward an automated process of statistical inference. Journal of the Royal Statistical Society Series B, 87:98–131, 2025.

---

### Official Review · Reviewer_ZTUj · 2025-07-02

**Clarity:** 2
**Significance:** 2
**Originality:** 3
**Rating:** 4
**Confidence:** 2

**Summary:**

In this paper, the authors propose a more accurate uncertainty quantification method for Physics-Informed Neural Networks (PINNs). Traditional Bayesian approaches require specifying a prior distribution to estimate the posterior, which can be challenging and subjective. Similar limitations apply to dropout-based methods. To address this, the authors leverage Extended Fiducial Inference (EFI)—a method that avoids the need for prior specification. However, the original EFI framework has known limitations in handling high-dimensional model parameters and large sample sizes. To overcome these challenges, the authors develop a new EFI-based framework tailored to PINNs, with accompanying theoretical guarantees. The proposed method is evaluated on both synthetic toy problems and real-world datasets.

**Questions:**

I am not very familiar with EFI. In line 91, a distribution for the random error is required—does that function similarly to a prior? I don’t think specifying a prior is a major drawback of Bayesian methods. In fact, incorporating informative priors, such as knowledge-based priors, can improve performance (see [1]). Even if the true prior is unknown, using a Gaussian prior with small variance generally does not hurt performance (see [2]).

More discussion is needed around general uncertainty quantification methods. There are various approaches such as MCMC methods, variational inference, and ensemble-based methods.

I couldn’t find a direct evaluation of uncertainty quality. Instead of relying only on MSE to assess regression accuracy, how can we compare the predicted uncertainty against ground-truth uncertainty?

The real-world evaluation is not strong. UQ is widely applied in computer vision and has been evaluated on various image classification tasks. The proposed method could also be tested on more challenging real-world tasks to better demonstrate its value. Can the method apply on classification tasks?

[1] Bayesian Neural Networks with Domain Knowledge Priors
[2] What Are Bayesian Neural Network Posteriors Really Like?

**Ethical Concerns:**

["NO or VERY MINOR ethics concerns only"]

**Final Justification:**

The rebuttal solved some of my concerns. I will raise the score but lower my confidence.

**Limitations:**

yes.

**Paper Formatting Concerns:**

No.

**Quality:**

2

**Strengths And Weaknesses:**

Strengths:

The paper is well-written and clearly organized, making it easy to follow the methodology and key contributions.

Enabling uncertainty quantification (UQ) for PINNs is a valuable and timely direction, with growing importance in scientific machine learning and real-world applications.

Weaknesses:

The paper lacks a thorough discussion of related work in general uncertainty quantification, particularly outside the context of PINNs.

The evaluation of the proposed UQ method is limited—there is no direct assessment of uncertainty quality.

There is a lack of real-world experiments to demonstrate the practical utility and robustness of the proposed approach.

---

> ### Author Rebuttal · Authors · 2025-07-31
>
> **[W1.] The paper lacks a thorough discussion of related work in general uncertainty quantification, particularly outside the context of PINNs.**
>
> In recent years, a variety of methods have been developed for uncertainty quantification in machine learning models, such as conformal prediction (Vovk et al., 2005), deep ensembles (Lakshminarayanan et al., 2016), stochastic deep learning (Sun & Liang, 2022; Liang et al., 2022). However, these methods primarily focus on prediction uncertainty and are either ineffective or inapplicable for quantifying uncertainty in model parameters. In contrast, EFI is applicable to both prediction uncertainty and model parameter uncertainty, and it further provides theoretical guarantees for the validity of the resulting prediction and confidence intervals.
>
> [1] Vovk, V., Gammerman, A., and Shafer, G. Algorithmic Learning in a Random World. Springer, 2005.
> [2] Lakshminarayanan, B., Pritzel, A., and Blundell, C. Simple and scalable predictive uncertainty estimation using deep ensembles. In Neural Information Processing Systems, 2016. URL https://api. semanticscholar.org/CorpusID:6294674.
> [3] Sun, Y. and Liang, F. A kernel-expanded stochastic neural network. Journal of the Royal Statistical Society Series B, 84(2):547–578, 2022.
> [4] Liang, S., Sun, Y., and Liang, F. Nonlinear sufficient dimension reduction with a stochastic neural network. NeurIPS 2022, 2022.
>
>
>
> **[W2.] The evaluation of the proposed UQ method is limited—there is no direct assessment of uncertainty quality.**
>
> Below we provide an example, where the length of intervals provides a direct assessment of uncertainty quality. See Q2.
>
> **[W3.] There is a lack of real-world experiments to demonstrate the practical utility and robustness of the proposed approach.**
>
> We are not sure what you are asking for. In the paper, we have provided some real data examples in the context of PINNs. Regarding real-world  applications of the original EFI method, we refer to Kim & Liang (2025), where EFI was applied to infer the uncertainty of individual treatment effects (with real data examples). Notably, as shown in Kim & Liang (2025), EFI eliminates the need for the central limit theorem in uncertainty inference for complex models, as well as the requirement for specifying the exact form of the target model.  Additionally, for potential applications of EFI on classification problems, see Q3.
>
> [5] Kim, S. and Liang, F. Extended fiducial inference for individual treatment effects via deep neural networks. Statistics and Computing, 35, 2025. URL https://api.semanticscholar.org/CorpusID:278326914.
>
>
> **[Q1.] In line 91, a distribution for the random error is required—does that function similarly to a prior? I don’t think specifying a prior is a major drawback of Bayesian methods. In fact, incorporating informative priors, such as knowledge-based priors, can improve performance. Even if the true prior is unknown, using a Gaussian prior with small variance generally does not hurt performance.**
>
> A distributional assumption for the random error is essential for any likelihood-based approach; without it, the likelihood function cannot be constructed. In Bayesian methods, a prior is specified only for the model parameters. While a well-specified prior—particularly one based on domain knowledge—can enhance statistical analysis, it can also introduce challenges. In small-n-large-p settings, such as Bayesian deep neural networks, an improperly specified prior may overwhelm the information from the data, leading to biased inference. To mitigate this issue, a theoretical justification for posterior consistency is generally required.
>
> In the context of Bayesian PINNS, the PDEs themselves act as a form of prior, constraining the neural network parameters to lie on a sub-manifold. For instance, consider the 1-D Poisson example in Section 4.1. In Bayesian PINNs, a Gaussian prior has to be introduced to constraint the network parameters 2 to satisfy the required PDE. However, the resulting uncertainty heavily depends on the variance $\sigma_f^2$ (see Table 1 in manuscript). This indicates the dependence of statistical inference (for the model) on the prior hyperparameters, which is undesirable for a scientific method. In contrast, EFI offers a mathematically rigorous framework for incorporating PDE constraints into neural network training, enabling accurate uncertainty quantification for both predictions and model parameters—without relying on prior distributions.
>
> **[Q2.] More discussion is needed around general uncertainty quantification methods. There are various approaches such as MCMC methods, variational inference, and ensemble-based methods. I couldn’t find a direct evaluation of uncertainty quality. Instead of relying only on MSE to assess regression accuracy, how can we compare the predicted uncertainty against ground-truth uncertainty?**
>
> See W1 for discussions on general uncertainty quantification methods. We appreciate the feedback regarding the evaluation of our uncertainty quantification (UQ) method. In the context of Physics-Informed Neural Networks (PINNs), the ground-truth uncertainty is inherently unknown, making direct comparison between predicted and true uncertainty challenging. To address this, we adopt a statistical approach by assessing the reliability of predicted uncertainties through confidence intervals and their coverage rates. This is a well-established practice in statistics when ground-truth uncertainty is unavailable.
>
> To further validate our method, we conducted experiments on simpler problems (e.g., linear regression) where analytical uncertainty estimates (e.g., from Ordinary Least Squares, OLS) are available and serve as the ground truth. While applying our proposed EFI algorithm to such problems may seem excessive, it serves as a sanity check. By comparing EFI-derived confidence intervals with those from OLS, we demonstrate that EFI produces well-calibrtaed uncertainty estimates (nearly same as the ground truth). Below is an example of this verification:
>
> Consider a simple linear regression model:
> $$y_i = \beta_0  + \beta_1 x_i + \epsilon_i,\quad i=1,\dots, n$$
> where we set $\beta_0=-1$ and $\beta_1=2$.
> Linear regression can be reinterpreted within the framework of PINNs as follows:
> $$ y_i^u = u_{\theta}(x_i^u) + z_i^u, \quad i=1,2,\dots,n_u, $$
> $$ 0 = \frac{d^2}{dx^2} u_{\theta}(x_i^f), \quad i=1,2,\dots,n_f.$$
>
> Here, $ u_\theta $ is a DNN with no pre-imposed structure, while the assumption of **linearity** is enforced through the differential equation constraint. This formulation seamlessly integrates linear regression into the PINN framework. For EFI, we can derive $\beta_1$ estimates through automatic differentiation, where $\beta_1 = \frac{d}{dx}u_\theta(x)|_{x=0}$. In table below, we show that EFI obtained almost the same confidence interval as OLS.
>
> **Table:** Confidence intervals for OLS and EFI
>
> | Algorithm | β₀ (95% CI)  | β₁ (95% CI) |
> |-|-|-|
> | OLS | (-1.12, -0.90)  | (1.88, 2.24) |
> | EFI  | (-1.12, -0.93)  | (1.87, 2.23) |
>
> **[Q3.] UQ is widely applied in computer vision and has been evaluated on various image classification tasks. The proposed method could also be tested on more challenging real-world tasks to better demonstrate its value. Can the method apply on classification tasks?**
>
> Although this paper focuses on UQ for PINNs, EFI framework can be extended to classification and logistic regression scenario. Consider a logistic regression:
> $$P(y=1|x, \theta) = \frac{1}{1 + \exp\{-x^T\theta\}}$$
> We can adopt EFI framework by modifying the energy function as follows:
> $$\sum_{i=1}^n \rho((z_i-x_i^T \theta)(2y_i-1)),$$
> where $\rho(\cdot)$ is the ReLU function, and
> $z_1,z_2,\ldots,z_n \stackrel{iid}{\sim} Logistic(0,1)$ with the CDF given by $F(z)=1/(1+e^{-z})$. In this experiment, the samples $x_i$ are generated from two 2D Gaussian distributions with means
> $\mu_1 = (2, 2)$ and $\mu_0 = (-2, -2)$, and a covariance matrix
> $\Sigma = 2.56 I_2$. Each class contains 500 samples. The true parameter is set to
> $\theta = (1, 1)$. We are able to achieve 95% coverage rate on the parameter $\theta$. We can also incorporate DNN structure to adapt nonlinear decision boundaries:
> $$\sum_{i=1}^n \rho((z_i-h(x_i;\theta))(2y_i-1)),$$
> where $h(\cdot;\theta)$ is an arbitrary DNN parameterized by $\theta$. Furthermore, we can extend EFI framework to multiclass classification problem. We define:
> $$
> p_{ij} = \frac{e^{-x_i^T \theta_j}}{\sum_{l=1}^{k} e^{-x_i^T \theta_l}},
> $$
> Then the fitting error term for multi-class logistic regression can be defined as follows:
> $$\sum_{i=1}^n \left[
> \rho(u_i- \sum_{l=1}^{m_i} p_{ij}) +
> \rho( \sum_{l=1}^{m_i-1} p_{ij} -u_i)
> \right],  $$
> where $u_i \sim Unif(0,1)$ denotes a random number, $m_i\in \{1,2,\ldots,k\}$ denotes the true class of the training sample $x_i$,
> $\theta_{j}$ denotes the parameters corresponding to class $j$, and
>  $x_i^T \theta_j$ is referred to as the log-odds of sample $x_i$ for class $j$.

---

> > ### Comment · Reviewer_ZTUj · 2025-08-05
> >
> > The EFI part is still somewhat confusing to me, as I'm not very familiar with this topic and it’s relatively new to me. As a result, it's difficult for me to fully assess the correctness of the EFI section and the associated theorems. That said, if everything in that part is indeed correct, I believe this is a solid paper, and I will raise my score to a 4.
> >
> > I do have a few follow-up questions. I’m still unclear on how confidence intervals are being used to demonstrate the quality of UQ. Additionally, I have some concerns about whether this method can be applied to more challenging real-world scenarios—for example, image data. Or maybe it is not important for PINN.

---

> ### Author Response · Authors · 2025-08-05
>
> Thank you very much for your encouraging comments and for kindly raising the score to 4.
>
> **Confidence Interval.** Also, thank you for your thoughtful question regarding the use of confidence intervals  in evaluating the quality of uncertainty quantification (UQ). In our approach, the following two metrics are primarily considered:
>
> **— Coverage rate:** For a nominal 95% confidence interval, approximately 95% of the true values should fall within the interval across test samples. This calibration metric directly reflects how well the uncertainty estimates align with their underlying true values.
>
> **— Interval width:** It reflects the size of the uncertainty.
> Narrower intervals are desirable if they maintain correct coverage, as they indicate sharper confidence intervals — that is, the model’s ability to provide informative and precise rather than overly conservative uncertainty estimates.
>
> In summary, confidence intervals serve as a diagnostic tool for assessing UQ quality by evaluating their **coverage** and **width**, which together characterize the reliability and informativeness of the model’s uncertainty estimates.
>
> **Practical Applications:**  We believe that the proposed method has strong practical potential. Notably, it can be extended to large-scale models, such as ResNets and CNNs (with applications to image data), via transfer learning, as discussed at the end of the paper.

---

### Official Review · Reviewer_qVGg · 2025-07-03

**Clarity:** 3
**Significance:** 3
**Originality:** 3
**Rating:** 5
**Confidence:** 2

**Summary:**

This paper proposes using extended fiducial inference (EFI) for uncertainty quantification for physics-informed neural networks (PINNs). This method does not rely on assuming a parameter prior, which can be very challenging for neural networks. This is unlike commonly applied methods such as Monte Carlo Dropout (MCDropout) and Hamiltonian Monte Carlo (HMC) methods. These methods are compared on simulated Poisson distribution data, and the results indicate that the confidence intervals produced by EFI outperform MCDropout and HMC. EFI is also evaluated on for two PINNs.

**Questions:**

MCDropout [1] uses both a Gaussian parameter prior and dropout probability hyperparameters. While poor priors limiting the effectiveness of Bayesian neural network methods was stated as a reason to use EFI, the prior variance hyperparmeter was not explored in the simulation experiment for McDropout. Also, the dropout probability hyperparameter can be learned for MCDropout using Concrete Dropout [2], although setting the prior variance hyperparameter could still be an issue for Concrete Dropout. Showing that EFI performs better than Concrete Dropout with varying prior variances would better support the claim that, by not depending on poor priors, EFI could be preferred over Bayesian neural network methods. Additionally, comparing the results of the BNN methods to those of EFI for the evaluated PINNs would further support the authors claims. Other methods, although not as popular as MCDropout or HMC, that could be compared to are Bayesian methods that, like EFI, use a double neural network approach [3,4,5]. Comparing to one of those would strengthen the paper.

[1] Gal, Yarin, and Zoubin Ghahramani. "Dropout as a bayesian approximation: Representing model uncertainty in deep learning." International Conference on Machine Learning. PMLR, 2016.

[2] Gal, Yarin, Jiri Hron, and Alex Kendall. "Concrete dropout." Advances in Neural Information Processing Systems 30 (2017).

[3] Louizos, Christos, and Max Welling. "Multiplicative normalizing flows for variational bayesian neural networks." International Conference on Machine Learning. PMLR, 2017.

[4] Krueger, David  et al. "Bayesian Hypernetworks." https://arxiv.org/pdf/1711.01297.

[5] Pawlowski, Nick et al. "Implicit Weight Uncertainty in Neural Networks." https://arxiv.org/pdf/1711.01297.

**Ethical Concerns:**

["NO or VERY MINOR ethics concerns only"]

**Final Justification:**

The paper appears to cover a novel approach to uncertainty estimation in deep neural networks. For the task of estimating confidence intervals of the predicted mean of the output variable in PINNs, the approach appears to perform better than common Bayesian neural networks approaches. However, the method uses a double neural network approach and double neural network Bayesian methods are not evaluated. From the current results and discussions with the authors, I do not expect those Bayesian methods to outperform the proposed approach. I think that this would be the case due to Bayesian methods learning parameter variances related to epistemic uncertainty, while EFI learns parameter variances by focusing on prediction error. With the inclusion of a discussion on comparing the parameter uncertainty of EFI with that of Bayesian approaches, the paper would better position the method within the literature. With the addition of such a section, I would be willing to raise my score to a 5. Although, my confidence remains a 2, due to my unfamiliarity with fiducial inference.

**Limitations:**

Yes

**Quality:**

3

**Strengths And Weaknesses:**

Quality: The proposed EFI method appears to be technically sound, although I am not an expert on fiducial inference methods. However, the evaluations should be significantly improved to support the claims of the authors.

Clarity: The paper seems to clearly presents the proposed methods and how they differ from previously proposed approaches.

Significance: High-quality uncertainty quantification methods for PINNs are not only of interest to the machine learning community but also the other scientific community, such as physics and biology.

Originality: The proposed method is an alternative to Bayesian neural networks, the current primary approach for achieving uncertainty quantification. In cases where parameter priors are not known, the proposed method could significantly improve upon those approaches in a novel way.

---

> ### Author Rebuttal · Authors · 2025-07-31
>
> ## Q1: Concrete Dropout.
> Consider Concrete Dropout by Gal et al. (2017). Unlike the traditional dropout method using fixed dropout rate, Concrete Dropout treats the dropout rate as a hyperparameter and learns it simultaneously when training the model.
>
> **Table 1** Metrics for 1D-Poisson with Concrete Dropout
>
> | Method              | MSE                  | Coverage Rate        | CI-Width              |
> |---------------------|----------------------|----------------------|-----------------------|
> | PINN (no dropout)   | 0.000121 (0.000012)  | 0.0757 (0.011795)    | 0.002139 (0.000024)   |
> | Concrete Dropout    | 0.000184 (0.000020)  | 0.1185 (0.018002)    | 0.003706 (0.000073)   |
> | EFI                 | 0.000148 (0.000016)  | 0.9517 (0.014186)    | 0.050437 (0.000462)   |
>
>
> We applied Concrete Dropout with the code provided on https://github.com/yaringal/ConcreteDropout. **Table 1** shows the result of 1-D Poisson model with Concrete Dropout. All necessary algorithm parameters in Concrete Dropout we applied were set as the code provided.
>
> Compared to our proposed EFI algorithm, Concrete Dropout achieves a comparable MSE, indicating similar model estimation accuracy. However, it performs significantly worse in terms of uncertainty quantification, with a coverage rate of only **11.85%**, far below the nominal 95%. This substantial under-coverage is due to an underestimation of predictive uncertainty, as evidenced by the markedly narrower confidence interval width.
>
> From another perspective, given that the performance metrics of Concrete Dropout are relatively similar to those of the PINN algorithm without dropout, we suspect that it effectively learns a dropout rate that is quite close to zero.
>
> More importantly, regardless of whether the learned dropout rate is small or large, the method lacks theoretical justification or principled guidance to ensure that the resulting predictive uncertainty aligns with the true underlying uncertainty. Furthermore, it is not applicable for quantifying uncertainty in model parameters. In contrast, EFI provides a unified framework for quantifying both predictive uncertainty and parameter uncertainty.
>
> ## Q2: Other Bayesian methods
> We also attempted to implement the method proposed in **Implicit Weight Uncertainty in Neural Networks** in the context of PINNs. However, to the best of our knowledge, there is no existing literature that reports such an implementation. Our attempt failed to converge to the desired PDE solution. Due to time constraints, we were unable to thoroughly tune the hyperparameters, although we have tried very hard. In contrast, the existing Bayesian approach, B-PINN, as demonstrated in our paper, has already shown that Bayesian methods fail to construct faithful confidence intervals for PINNs.

---

> > ### Comment · Reviewer_qVGg · 2025-08-09
> >
> > Looking back at the Concrete Dropout results, it appears that the regularization strength for the entropy in the regularization term (dropout_regularizer) may be too low, causing the method to collapse to something close to the no dropout model. A reasonable setting for this would be 1 divided by the number of training examples. Would it be possible to rerun the Concrete Dropout experiments?

---

> ### Comment · Reviewer_qVGg · 2025-08-04
>
> Thank you for adding the concrete dropout results. Upon further reflection, I have an additional question. For the trained Bayesian neural networks, was homoscedastic regression without a learned prediction variance, homoscedastic regression with a learned prediction variance, or heteroscedastic regression used? If homoscedastic regression without a learned prediction variance was used, as seems to be the case from Lines 287-297, then the confidence interval results for the Bayesian neural networks take into account only epistemic uncertainty and not aleatoric uncertainty, which is based on the prediction error. This makes me question the strength of the Bayesian neural network results.

---

> > ### Author Response · Authors · 2025-08-04
> >
> > Thank you for your encouraging comments on concrete dropout results. Regarding Bayesian neural network results, we would clarify that they were obtained by following the approach in Yang et al. (2021). In their implementation, $\sigma_u$ is known noise variance, and the confidence intervals are constructed from Bayesian samples $\\{\theta_i\\}$. The objective is to estimate the underlying true function $u(x)$, for which the confidence interval was constructed based on $\\{u_{\theta_i}(x)\\}$. Therefore, the confidence interval takes into account the epistemic uncertainty only. If one is interested in the prediction interval at $x$ (i.e., for a future observation), then the prediction variance (aleatoric uncertainty) should be taken into account.
> >
> > [1] Yang, L., Meng, X., and Karniadakis, G. E. B-pinns: Bayesian physics-informed neural networks for forward and inverse pde problems with noisy data. Journal of Computational Physics, 425:109913, 2021. ISSN 00219991. doi: https://doi.org/10.1016/j.jcp.2020.109913. URL https://www.sciencedirect.com/science/ article/pii/S0021999120306872.

---

> ### Comment · Reviewer_qVGg · 2025-08-06
>
> Please correct me if I am wrong. EFI seeks to learn a distribution for theta based on the prediction errors. Looking at the variability of predictions for thetas sampled from an approximation of the posterior using a Bayesian method would give you information on how much a prediction for an input is constrained by the training data, as can be seen in Figure 3 of Yang et al. (2021). It may not inherently be linked to prediction error. However, aleatoric uncertainty is learned by taking prediction error into account. I understand that this is different from how the confidence interval would be constructed for EFI, but it is, to my knowledge, the way that a Bayesian method would model prediction error. I'd like to see how well that performs. I acknowledge the use of confidence intervals by Yang et al. (2021), but I do not think it is the correct tool for reasoning about prediction error for Bayesian models.

---

> ### Author Response · Authors · 2025-08-06
>
> Thank you for your thoughtful comments and for highlighting the important distinction between EFI and Bayesian methods. Your observations regarding confidence interval construction and the role of prediction errors are well taken and largely correct.
>
> To clarify some of the subtleties in how prediction error is handled, consider the following simple model:
> $$
> Y_i = g(X_i, \beta) + \sigma Z_i, \quad i = 1, 2, \ldots, n,
> $$
> where $ \beta $ and $ \sigma $ are model parameters, $ Z_i \sim F $ represents random error from a distribution $F$, and $ n $ is the sample size. Let $ \theta = (\beta, \sigma) $ denote the full set of parameters. Suppose our goals are:
>
> (i) to infer the function $ g(x_0,\beta) $ at a given covariate $X=x_0$,
> (ii) to predict a new response value $ y_0 $ at a given covariate $X=x_0 $.
>
>
> In Bayesian inference, if we have obtained posterior samples $ \tilde{\theta}\_1, \ldots, \tilde{\theta}\_M $, then uncertainty of $g(x_0,\beta)$ is assessed via the variability in the samples $g(x_0,\tilde{\beta}\_1), \dots, g(x_0,\tilde{\beta}\_M)$ --- capturing **epistemic uncertainty**. For prediction (goal ii), **aleatoric uncertainty** is incorporated by generating
> $$
> \tilde{y}\_{0,k}:=g(x_0, \tilde{\beta}\_k) + \tilde{\sigma}\_k \tilde{Z}\_k, \quad k = 1, \ldots, M,
> $$
> where $ \tilde{Z}\_k \sim  F $ are independent draws simulating the irreducible noise component (also independent of $Z_1,\ldots,Z_n$). Then prediction interval of $y_0$ can be constructed using $\tilde{y}\_{0,k}$, $k=1,2,\ldots,M$.
>
> EFI, by contrast, draws samples from the **fiducial distribution** $\mu_n^*(\theta \mid D_n) $, based solely on the observed data $ D_n=(X_1,y_1,\ldots,X_n,Y_n) $, without requiring a prior. This makes the inference more faithful to observations. Once fiducial samples of $ \theta $ are obtained, the subsequent steps mirror those in Bayesian analysis: uncertainty in $ g(x_0,\beta) $ is quantified through the empirical distribution of $g(x_0,\tilde{\beta}_1), \ldots, g(x_0,\tilde{\beta}_M)$, and predictive intervals for $ y_0 $ are constructed based on additional simulated noise $ \tilde{Z}_k $, $k=1,2,\ldots,M$.
>
> Notably, inference of the underlying true function $ u_{\theta}(x) $ in the present paper corresponds to goal (i). It is also worth emphasizing that EFI adopts a fundamentally different approach to construct its fiducial distribution $ \mu_n^*(\theta \mid D_n) $, which serves a similar role to the posterior distribution in Bayesian methods. Consequently, it necessitates a distinct sampling algorithm, differing from conventional MCMC techniques typically used in Bayesian inference.

---

> > ### Comment · Reviewer_qVGg · 2025-08-07
> >
> > Thank you for the detailed response. The results indicate that, if the goal is to get a confidence interval for the predicted mean of the Gaussian prediction distribution by sampling from the learned parameter distribution (the fiducial distribution for EFI and the approximate posterior for Bayesian approaches), EFI drastically outperforms Bayesian methods. While Bayesian methods are not explicitly designed for this approach, EFI is designed for it. Consequently, the results demonstrate that if a confidence interval on the prediction mean is needed EFI can provide that. I'd like to see some discussion of the above mentioned goals and epistemic and aleatoric uncertainty added to the paper.

---

> > > ### Author Response · Authors · 2025-08-07
> > >
> > > Thank you for your encouraging comments on EFI. We will certainly take your suggestion to include a discussion on how each type of uncertainty can be addressed within the EFI framework.

---

> ### Author Response · Authors · 2025-08-09
>
> Thank you for the advice, we reran the 1D–Poisson experiments (20 noisy boundary data points and 200 exact PDE data points) using dropout regularizers ranging from 0.1 to 0.0025, as summarized in Table 1. The highest coverage rate observed was **21.77%**, which is substantially below the nominal 95% target. From the small MSE values, we see that Concrete Dropout provides accurate point estimates; however, the narrow CI widths and low coverage rates indicate that the interval estimates are not well-calibrated and tend to be overly narrow. As noted in [1], Concrete Dropout improves calibration compared to standard dropout, particularly through automatic tuning of dropout probabilities; however, it does not claim to fully resolve calibration challenges. In the context of PINNs, we suspect that the PDE regularizer — which requires computing gradients with respect to the network input — may interact with the dropout mechanism in a way that limits its effectiveness. This interaction may partly explain why the anticipated calibration benefits are not fully realized in our setting.
> **Table1** Results for 1D-Poisson with Concrete Dropout
>
> | Method           | Dropout Regularizer | CI Width           | Coverage Rate       | MSE               |
> |------------------|---------------------|--------------------|---------------------|-------------------|
> | Concrete Dropout | 0.0025              | 0.005964 (0.000226) | 0.2177 (0.026686)   | 0.000175 (0.000018) |
> | Concrete Dropout | 0.005               | 0.004071 (0.000191) | 0.1406 (0.021347)   | 0.000153 (0.000017) |
> | Concrete Dropout | 0.01                | 0.002486 (0.000118) | 0.0767 (0.012740)   | 0.000150 (0.000014) |
> | Concrete Dropout | 0.025               | 0.001689 (0.000031) | 0.0552 (0.008315)   | 0.000149 (0.000016) |
> | Concrete Dropout | 0.05                | 0.001732 (0.000031) | 0.0465 (0.006927)   | 0.000165 (0.000020) |
> | Concrete Dropout | 0.1                 | 0.001749 (0.000029) | 0.0618 (0.011809)   | 0.000140 (0.000017) |
>
>
> [1] Accurate Uncertainties for Deep Learning Using Calibrated Regression (https://proceedings.mlr.press/v80/kuleshov18a/kuleshov18a.pdf)

---

### Note · Authors · 2025-08-12

Throughout the rebuttal process, we observed that the reviewers’ primary concern lies in understanding the correctness of the relatively new approach, EFI, and why it can outperform well-established uncertainty quantification methods such as dropout and Bayes. Here we briefly clarify the high-level concept of EFI.

Consider the data-generating equation $$y_i=u(x_i;\theta) + z_i,\quad i=1,\dots,n,$$ where $\theta\in\mathbb{R}^p$ denotes the parameter and $z_i\sim F$ is random error. Since the total number of unknowns ($n+p$), including $\theta$ and $z_1,\ldots,z_n$, is greater than the number of equations ($n$), the solution of $\theta$ is undetermined, causing its uncertainty. EFI aims to quantify this uncertainty by learning a mapping from data to parameters, denoted by $$\theta=G(X,Y,Z),$$ through which the uncertainty embedded in the data (i.e., the noise variable $Z$) is propagated to $\theta$ via a variable transformation. EFI is trained using an adaptive SGMCMC algorithm, which ensures that the latent variable $Z$ is correctly imputed and the inverse mapping is consistently estimated. Consequently, both the uncertainty of $Z$ (i.e., aleatoric uncertainty) and the uncertainty of $\theta$ (i.e., epistemic uncertainty) can be accurately quantified. The theoretical foundation of EFI has been rigorously established in [1] and [2], both published in top statistical journals.

The present work introduces another innovation to the theory of EFI. Leveraging the narrow-neck structure, it completely eliminates the need for prior specification, thereby aligning the method with the principle of fiducial inference—drawing inferences about parameters solely from the observations. In [1] and [2],, EFI relied on a sparse $w$-network to ensure the consistency of the inverse mapping.

EFI solves the data-generating equation, providing an explicit mechanism for propagating uncertainty from observations to parameters. In contrast, many existing methods, e.g., dropout, are not designed from the perspective of solving data-generating equations, making their resulting uncertainty difficult to calibrate. Bayesian methods treat $\theta$ as a variable without solving the data-generating equation. For high-dimensional problems, such as PINNs, the choice of prior can significantly influence the results.

[1] Liang, F., Kim, S., and Sun, Y. (2025) Journal of the Royal Statistical Society Series B, 87:98–131.

[2] Kim, S. and Liang, F. (2025) Statistics and Computing, 35.

---

### Decision · Program_Chairs · 2025-09-17

**Decision:**

Accept (poster)

**Comment:**

The paper follows in the footsteps of many works aiming to tackle uncertainty quantification in machine learning. In contrast to many of the currently well known methods, the paper however explores an alternative based on extended fiducial inference (EFI). This uncertainty quantification method is claimed to surpass some of the caveats of traditional methods, including Monte-Carlo Dropout or Bayesian uncertainty estimates. The paper first introduces EFI, then provides theoretical arguments for its use, and finally empirically compares it on a simulated example and small real-world data. The main strength of the paper is definitely its novelty and uniqueness, demonstrating out of the box thinking and a well-written exposition of a potentially new direction. This simultaneously could be regarded as the main weaknesses, making the paper hard to follow for non-experts in EFI (which is likely to be the majority of present researchers in the ML field) and limiting the current empirical assessment and comparison. The paper also does not inspect the use of uncertainty estimates in a deeper way.

Overall, this is a very hard to judge paper, as has also been noted by various reviewers. Like the reviewers, the AC is not an expert in fiducial inference, and the confidence in judgement of the paper remains similarly low to all the reviewers. On the positive side, the reviewers have all agreed that the perspective is likely novel and the paper appears to be technically correct, to the degree they can judge it. The AC agrees with this perspective and did not find obvious technical flaws. The main points of initial concern mostly seem to resolve around a) the limited discussion of other uncertainty quantification methods in the literature, b) a limited empirical investigation and comparison on any real world scenarios (beyond the simple one presented), c) a general lack of clarity whether the method indeed achieves what it promises in terms of overcoming limitations by other uncertainty quantification strategies. Especially the latter was a big point in the rebuttal, where the authors have provided further evidence and discussions to support their claims. Ultimately reviewers were decently satisfied, although their confidence remains low, finally all agreeing that the paper has a lot of potential value, that remains hard to judge.

The AC shares the sentiment with the reviewers that there are many ways in which the paper could be improved. The AC recommends the authors to carefully revise the paper for its final version. In particular, given that the paper doesn't currently have an appendix, the AC strongly recommends to provide substantially more background in appendix sections, so that readers less familiar with such a new topic have an easier time to follow. In these appendix sections, the AC also strongly encourages the authors to add their additional experiments wrt hyper parameter choices, that were provided in the rebuttal.

Taking all of the above into account, the AC finally recommends to accept the paper. Despite general low confidence, all reviewers and the AC agree that the paper may have a lot of value, even if it remains hard to judge by today's standards. This is generally the fate of any papers that pursue out-of-the box and novel directions, and it would feel wrong to penalize the paper for pursuing non-standard directions. Given the assumed technical correctness of the paper, the AC recommends to accept the paper and let the community decide the use and impact of the paper as time progress.